# Whole-genome sequencing of chronic lymphocytic leukemia identifies subgroups with distinct biological and clinical features

Pauline Robbe [1,2,43], Kate E. Ridout[1,43], Dimitrios V. Vavoulis [1], Helene Dréau[1], Ben Kinnersley [3], Nicholas Denny [4], Daniel Chubb[3], Niamh Appleby [1], Anthony Cutts[1], Alex J. Cornish [3], Laura Lopez-Pascua[5], Ruth Clifford[6,7], Adam Burns[1], Basile Stamatopoulos[8], Maite Cabes[9], Reem Alsolami[10], Pavlos Antoniou[11], Melanie Oates[12], Doriane Cavalieri[13], Genomics England Research Consortium*, CLL pilot consortium, Jane Gibson [14], Anika V. Prabhu[2], Ron Schwessinger[4], Daisy Jennings[2], Terena James[11], Uma Maheswari[11], Martí Duran-Ferrer [15], Piero Carninci[2,16], Samantha J. L. Knight[17], Robert Månsson[18], Jim Hughes [4], James Davies[4], Mark Ross[11], David Bentley[11], Jonathan C. Strefford [19], Stephen Devereux [20,21], Andrew R. Pettitt[22,23], Peter Hillmen[24], Mark J. Caulfield[25,26], Richard S. Houlston [3], José I. Martín-Subero[16,27] & Anna Schuh [1] ✉

The value of genome-wide over targeted driver analyses for predicting clinical outcomes of cancer patients is debated. Here, we report the whole-genome sequencing of 485 chronic lymphocytic leukemia patients enrolled in clinical trials as part of the United Kingdom's 100,000 Genomes Project. We identify an extended catalog of recurrent coding and noncoding genetic mutations that represents a source for future studies and provide the most complete high-resolution map of structural variants, copy number changes and global genome features including telomere length, mutational signatures and genomic complexity. We demonstrate the relationship of these features with clinical outcome and show that integration of 186 distinct recurrent genomic alterations defines five genomic subgroups that associate with response to therapy, refining conventional outcome prediction. While requiring independent validation, our findings highlight the potential of whole-genome sequencing to inform future risk stratification in chronic lymphocytic leukemia.

Chronic lymphocytic leukemia (CLL), the most common adult hematological malignancy in Western countries, is characterized by diverse treatment outcomes even in the era of targeted agents. The full complement of genomic events contributing to this clinical diversity have yet to be determined. Thus far, only mutations in *TP53* influence clinical practice[1-7]. Other prognostic markers, including the immunoglobulin heavy chain variable (IGHV) region mutational status, and existing molecular classifications have limited predictive value in individual patients[7-10].

Previous sequencing studies of CLL have focused largely on mutations affecting protein-coding genes[7-13], and whole-genome sequencing (WGS) has been reported for only a small number of CLL patients,

mostly with low-risk disease[1-6]. Hence, the association between clinical parameters and genomic alterations has largely been restricted to driver coding mutations and copy number changes.

Here, to provide the largest and most comprehensive analysis of the entire genomic landscape of CLL and its relationship to clinical outcome, we performed WGS of 485 clinical trial patients recruited to the United Kingdom's 100,000 Genomes Project. The results of our study provide additional insights into coding and noncoding single nucleotide mutations. We then exploit WGS data to provide a detailed map of structural alterations and global features, including telomere length, mutational signatures and genomic complexity (GC). Finally, we integrate the different modes of genetic alterations to define five genomic subgroups (GSs) of CLL and relate these to clinical outcome. Our results provide a springboard to indepth functional validation of putative drivers and our integrated genome-wide approach could, after independent clinical validation, refine current clinical outcome prediction.

## Results

We performed WGS of tumor and matched normal samples from 485 patients with treatment-naïve CLL enrolled in clinical trials to a median depth of 109× and 36×, respectively (Supplementary Tables 1–3). A second tumor sample was available for a subset of 25 patients at relapse. In addition, RNA sequencing (RNA-seq; $n = 73$) and assay for transposase-accessible chromatin with high-throughput sequencing (ATAC-seq; $n = 24$) data were generated for a subset of CLL samples with recurrent noncoding mutations (Supplementary Table 4).

### Coding mutations and structural variants

We initially identified putative coding drivers by (1) screening for genes impacted by single nucleotide variants (SNVs) and small insertion/deletions (indels) and (2) integrating SNV/indels with copy number alterations (CNAs) (Fig. 1a; Methods). We identified 36 known and 22 putative driver genes (Fig. 1b and Supplementary Fig. 1), which were not found associated with CLL in the literature and also not prevalent above 1% in two landmark genomic studies in CLL[3,7]. These were classified as previously unknown putative drivers and included the immune checkpoint regulator IRF2BP2 (4.3%) (Supplementary Table 4).

We identified 74 regions of the genome that were recurrently affected by CNAs in at least four samples (Fig. 1c, Extended Data Fig. 1a and Supplementary Table 6). Using DNA microarray data, 85% of CNAs could be validated (Supplementary Table 7). In addition to 14 well-known CNAs, including del13q14.2, del11q22.3 and del17p13.1, we identified a further 60 regions, of which 27 were previously not recognized. The breakpoints of the remaining 33 CNAs could be refined to a smaller minimally overlapping region[14-19]. By combining SNVs/indels with CNAs (discovery method 2; Methods), we predicted the most likely target gene for nine known regions, including TP53/del17p13.1, and seven additional regions including PCM1/del8p, IRF2BP2/del1q42.2q42.3 and SMCHD1/del18p11.32-p11.31 (Fig. 1d, Extended Data Fig. 1b,c and Supplementary Table 8). We also found 66 additional genes affected by recurrent CNAs using more permissive

criteria (Methods). While these are potentially interesting, they were not considered to be putative CLL drivers and were not taken forward for downstream analyses (Supplementary Table 9).

A major advantage of WGS is the power to identify inversions and translocations. We identified 1,248 inversions (Extended Data Fig. 2a; Methods) with frequent breakpoints involving either the immunoglobulin light chain kappa (IGK) locus ($n = 65$, 13.4%), the immunoglobulin heavy chain (IGH) locus ($n = 65$, 13.4%) or chr13q14.2 ($n = 40$, 8.7%) (Extended Data Fig. 2b and Supplementary Tables 10 and 11). We detected 993 translocations, of which two occurred in more than ten samples and affected known genes with no previously documented role in CLL, including t(14;22) with breakpoints within WDHD1 ($n = 12$, 2.6%) and t(5;6) (CTNND2-ARHGAP18, $n = 11$, 2.4%) (Fig. 1e and Extended Data Fig. 2c).

The 22 potential coding driver genes were altered by truncating mutations or also affected by CNAs (Fig. 2a, Extended Data Fig. 3a–d, Supplementary Table 12 and Supplementary Figs. 2 and 3). Most mutations occurred in protein domains, and 62% of mutations were detectable in more than 50% of tumor cells (median cancer cell fraction (CCF) ≥0.5) and 89% in at least 20%. All previously unreported CNAs for which we could predict a target gene(s) were also clonal (median CCFs ≥0.8) (Fig. 2b and Extended Data Fig. 3e). Candidate driver mutations affected multiple biological pathways including the DNA damage/cell-cycle and RNA-ribosome processing (Fig. 2c).

Performing RNA-seq on representative CLL samples from 74 patients with known and potential coding mutations (for 40 of the 58 drivers, $n$ variants = 118, Supplementary Table 4; Methods), we validated the expression of 73% of variants at the RNA level (Extended Data Fig. 4a and Supplementary Table 13). As expected, most (29/43) mutations that were either not detectable or were seen at low expression levels were truncating mutations consistent with nonsense-mediated decay (Supplementary Table 13). Additionally, allelic skewing and/or a reduction of mutant transcript expression compared with the mean expression of wild-type (WT) transcripts across the cohort was shown, notably for specific mutations in SPEN, SETD2, TP53 and IRF2BP2 (Fig. 2d). When considering all mutations, significantly reduced gene expression was demonstrated for TP53, ATM and SETD2 (refs. [20,21]) (Extended Data Fig. 4b).

When we associated the 36 known and 22 putative drivers and regions of CNAs with other biological variables such as disease stage, TP53 alterations, IGHV mutation status (unmutated, u-IGHV; and hypermutated, m-IGHV) and stereotyped B cell receptor immunoglobulin subsets (BCR IG) including IGHV3-21 usage (Fig. 2e and Supplementary Table 14; Fisher's exact test, false discovery rate (FDR) < 0.05), we found that SETD2/del3p21.31, del9p21.3 and gains of chr17q21.31 were associated with relapsed/refractory (R/R) disease and TP53 disruption, whereas MED12 and DDX3X mutations were associated with u-IGHV CLL. BCR IG subset 2, representing about 3% of all CLL, and known to be associated with poor prognosis[22], was linked to the putative driver FAM50A. The IGHV3-21 rearrangement was also enriched for FAM50A and for ATM/del11q22, SF3B1 mutations and chr21q21.3-q22.3 gains.

**Fig. 1 | Identification of coding mutations and structural variants.**
**a**, Methodology used for the discovery of candidate coding drivers. Discovery method 1A selected genes with a FDR below significance threshold for two out of four algorithms. Discovery method 1B combined the P values of four algorithms using weighted Stouffer and weighted Harmonic mean. Genes with FDR below significance threshold for at least one result were selected. With discovery method 2, CNAs were used to define minimally affected regions (by copy number loss or gain). Then, genes included in these genomic regions were selected as candidate drivers if they presented at least five SNVs/indels impacting the coding sequence focality and recurrence scores greater than threshold and mechanism of action of gene in agreement with the type with CNAs (loss for TSG and gain for oncogenes). An additional list, not considered as candidate drivers, included genes fulfilling all requirements except the SNVs/indels count threshold.

(Permissive list; see Methods for more details). **b**, Number of SNVs/indels (left axis) and proportion in the cohort (right axis) of the 58 candidate drivers. Other CLL cohorts used as comparators are described in (Supplementary Table 5; Methods). **c**, The 76 regions recurrently affected by CNAs. The y axis is shown in $\log_{10}$ scale. Known CLL drivers are indicated in blue and putative driver genes identified as hotspots are indicated in yellow. **d**, Candidate drivers found by integrating both CNAs and SNVs/indels. The score represents combined focality and recurrence scores derived from MutComFocal, integrating SNVs/indels data with CNA data (Discovery method 2; Methods); NS, not significant. Known CLL drivers are indicated in blue and putative driver genes identified as hotspots are indicated in yellow. **e**, All translocation breakpoint pairs found in more than three samples (out of 495), including those occurring in coding and noncoding regions.

## Association of coding mutations with disease evolution

We examined the relationship between recurrent gene mutations and disease evolution in three different cohorts (Fig. 3a and Supplementary Table 4; Methods): (1) unpaired frontline-treated versus R/R (main cohort, unmatched, $n$ = 443 versus 30—excluding the 12 early CLL); (2) paired samples from the CLL and Richter's syndrome (RS) phases

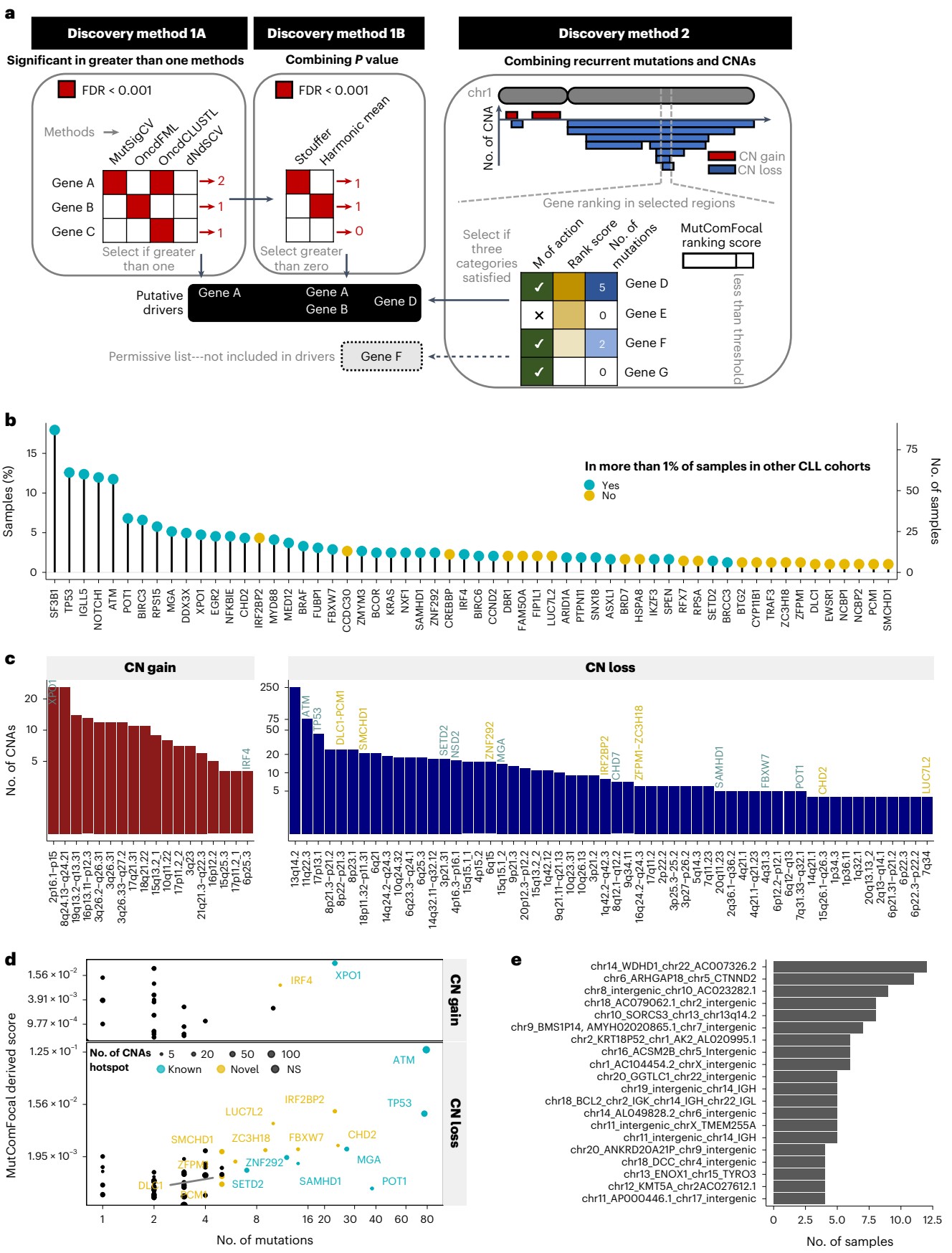

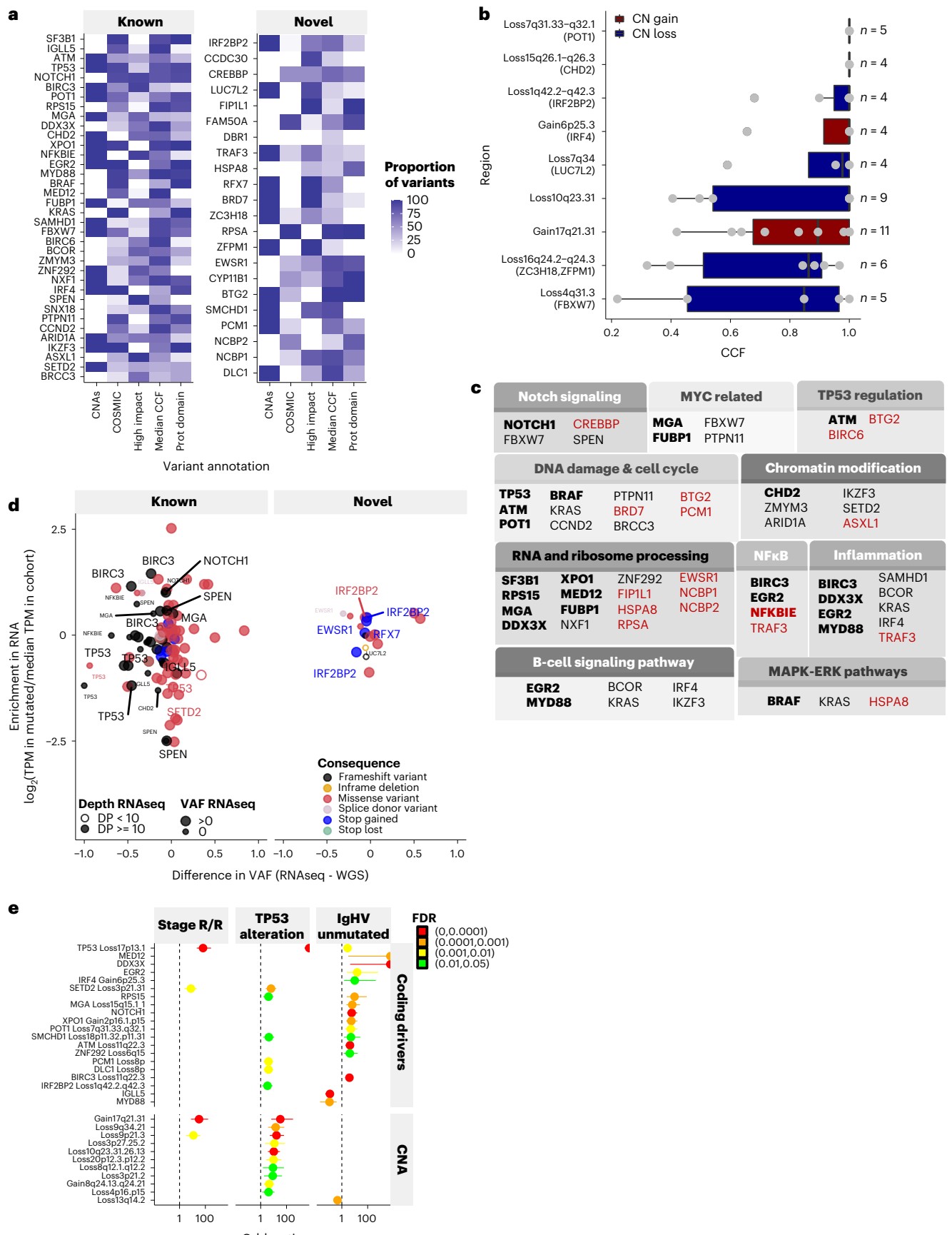

**Fig. 2 | Biological features of coding mutations and CNAs. a**, Annotations of genes. CNAs, presence/absence of CNAs affecting the gene; COSMIC, proportion of variants reported in the COSMIC database; High impact, proportion of nonsense variants, Median CCF, median cancer cell fraction of variants; Prot domain: proportion of variants occurring in a protein domain from the Prot2HG database[40]. **b**, Distribution of cancer cell fractions in selected recurrent regions of CNAs (all regions in Extended Data Fig. 3e). The boxplot shows the minimum and maximum values and the interquartile range. **c**, Candidate drivers classified in ten main pathways described in CLL[3,7]. Genes in bold are present in more than 3% and genes in red font are candidate drivers. Other drivers are absent because not involved in these ten main pathways. **d**, Detection of variants of interest (*n* = 118) by RNA-seq (with minimum depth of five) in selected 73 samples.

Difference of variant allele frequency (VAF) between RNA-seq methods and WGS methods shows allelic skew of variants. Ratio of expression in transcript per million (TPM) in sample with variant against all other samples reflects change in gene expression. Selected variants annotated with gene names, all data in Supplementary Table 13. DP, depth. **e**, Enrichment of genomic features in different CLL subgroups using two-sided Fisher's exact test (plot showing the median, minimum and maximum values). The groups were (1) stage: relapsed/refractory (R/R), versus frontline (*N* = 443 frontline versus 30 R/R), (2) *TP53*: altered versus WT (*N* = 420 WT versus 65 disrupted), (3) IGHV mutational status: unmutated versus hypermutated (*N* = 197 hypermutated versus 288 unmutated), where an enrichment for the former is indicated by an odds ratio greater than one. Adjusted *P* values (FDR) are shown.

of the same patient (previously published cohort[23], matched, *n* = 17) and (3) a second sample taken from a subset of the 485 patients at relapse who had already been profiled before frontline treatment: paired frontline-treated versus relapsed (main cohort, matched, *n* = 25/485).

Recurrent coding gene mutations were linked to disease evolution in all three cohorts. They presented higher mutation counts and frequency in the RS compared with the CLL phase (*P* = 2.1 ×10⁻²; Extended Data Fig. 4c,d) and higher CCFs at the more advanced stages with a median CCF > 0.8 (Fig. 3b and Extended Data Fig. 4e–g).

Restricting analysis to patients with information on long-term survival outcome (*n* = 243 / 485), 13 known or putative drivers and recurrent CNAs were significantly associated with progression-free survival (PFS) and 11 with overall survival (OS) (FDR < 0.05) (Fig. 3c and Supplementary Tables 15 and 16).

Out of the 22 putative drivers, 21 were also related to disease progression (Extended Data Fig. 4c–f), including two of the most commonly mutated ones. *IRF2BP2* (interferon regulatory factor 2 binding protein 2), located in the minimally deleted region of chr1q42.3 (Fig. 3d) was also affected by deleterious mutations and CNAs (Fig. 3e) (in total, *n* = 28/485, 5.8%) with high CCFs (Fig. 3f, left panel). Mutations showed evidence of clonal expansion in more advanced disease (Fig. 3f, right panel) and altered RNA expression (Fig. 2d). This gene contributes to the differentiation of immature B-cells and is associated with a familial form of common variable immunodeficiency disorder[24].

Similarly, *SMCHD1* (structural maintenance of chromosomes flexible hinge domain containing 1), previously reported as a candidate tumor suppressor in hematopoietic cancers[25] was affected by copy number losses (del18p11.32-p11.31) (Fig. 3g) and truncating SNVs/indels with high CCFs (Fig. 3h) (*n* = 24/485, 5.0%). *SMCHD1* mutations showed clonal expansion (Fig. 3i) and were associated with adverse OS (median = 48.2 months, *P* value < 1 × 10⁻⁴, log-rank test) (Fig. 3j).

### Noncoding putative driver mutations

To gain insight into the significance of noncoding mutations, we first identified CLL-specific regulatory elements (REs) by integrating ATAC-seq and H3K27ac profiles[26,27] as well as chromatin states[28] from

publicly available primary CLL (Fig. 4a; Methods). Out of the 29,224 promoters and 56,137 enhancers identified, 90% were present in CLL as a whole, whereas the remaining 10% were specific for IGHV subgroups and were used for the IGHV subtype-specific annotation (Methods). Mapping noncoding mutations to REs (Fig. 4b; Methods), we could identify 29 untranslated regions (UTRs), 25 enhancers (23 of them cataloged by the GeneHancer database[29]) and 72 promoters that had hotspot mutations or were recurrently mutated more frequently than expected (FDR < 0.1), defined as significantly mutated (Extended Data Fig. 5a and Supplementary Table 17).

Next, we defined the candidate target genes of these 126 mutated noncoding regulatory elements. Mutations within UTRs and promoters were annotated predominately according to proximity (Methods). For enhancers, we calculated the correlation between H3K27ac levels for each regulatory elements and the gene expression levels of surrounding genes located within the same topologically associated domain (TAD) of the B cell lymphoblastoid cell line GM12878[30] (Methods). In total, 29 regulatory elements had target genes known to be CLL drivers or cancer drivers in the COSMIC database (Fig. 4c); 89 were linked to other genes (Fig. 4d) and 8 to none (Extended Data Fig. 5a and Supplementary Table 17). Four mutated regulatory elements were specific for u-IGHV (Extended Data Fig. 5b) and none for m-IGHV. Overall, genes targeted by mutated regulatory elements were enriched for gene ontology terms linked to the immune system, lymphocyte activation and cell death (Fig. 4e and Supplementary Table 18).

Of the 29 mutated UTRs, 58% (*n* = 17) had a median CCF ≥ 0.5, and 83% had a CCF > 0.2, thus indicating their selection during CLL pathogenesis (Extended Data Fig. 5c). These included the 3′ UTR mutations of *NOTCH1* creating a splice site that leads to increased gene expression[3,31] (*n* = 16; FDR = 4.57 × 10⁻²). The NF-κB signaling gene *NFKBIZ* (*n* = 8, FDR = 2.38 × 10⁻²) was also found significantly mutated, confirming previous findings[6] and known to increase levels of mRNA and protein in lymphoma[32,33]. We observed clonal mutations in the 5′ UTR of *IGLL5* (*n* = 28; FDR < 2.2 × 10⁻¹⁶), previously found to be associated with reduced expression[4]. Previously unreported significantly mutated UTRs included the 5′ UTR of *BCL2* (*n* = 6; FDR = 1.01 ×10⁻⁶, Fig. 5a). We performed RNA-seq on samples carrying these mutations

**Fig. 3 | Associations of coding mutations and CNAs with disease progression. a**, Three cohorts used for studying the presence of variants during disease evolution. Unpaired samples are taken from different patients; cohort (1) were samples from treatment-naïve patients and R/R patients; cohort (2) were paired samples of CLL and RS phase of the same patient; cohort (3) were paired samples taken at two different timepoints before treatment and at relapse. **b**, Distribution of cancer cell fractions in the three cohorts studied for selected genes. For cohort (1), figures are not shown if no R/R sample carried a mutation. Other genes are presented in Extended Data Fig. 4e,f. Boxplots showing results for unpaired samples and connected datapoints show results for paired samples (corresponding variants are connected by a dotted line). An asterisk indicates a candidate driver. **c**, Genomic features linked to patients' PFS (left panel) and OS (right panel). Hazard ratio and FDR of each genomic feature tested against PFS using a Cox proportional-hazards model on the subset of patients for which clinical outcomes data were available (*n* = 243). Adjusted *P* values (FDR) are

shown in different colors. (See Supplementary Table 14 for the full detailed list of genomic features tested and Supplementary Table 15 and 16 for full results of the statistical tests). **d–f**, candidate driver *IRF2BP2* was recurrently affected by CN losses (**d**) and SNVs/indels, especially truncating ones (**e**), and was associated with increased CCF in variants for more advanced disease stage (**f**). Coloured rectangle in (**e**) represents protein domains. **g–j**, candidate driver *SMCHD1* was recurrently affected by CN losses (**g**) SNVs/indels, especially truncating ones (**h**), presented evidence of increased CCFs in more advanced CLL in cohort (2) (no data available in R/R of cohort (1)) (1), and associated with more adverse overall survival as shown in the Kaplan–Meier plot where shaded areas show the 95% confidence intervals and *P* values were derived from a log-rank test (**j**). Coloured rectangle in **e** represent protein domains. Boxplots show the minimum and maximum values and interquartile range and each individual variant is represented with an individual datapoint.

(Supplementary Table 4; Methods) demonstrating that 5′ UTR mutations were associated with BCL2 overexpression ($P = 4.3 \times 10^{-2}$; Fig. 5b), which is noteworthy given that BCL2 inhibitors are used therapeutically in CLL[34].

A high clonality (>0.5) was also observed when considering the 97 significantly mutated promoters and enhancers; 72% had a median CCF >0.5 and 97% of a CCF >0.2 (Supplementary Fig. 4a). Six discrete regions spanning 117 kb contained 50 variants and were annotated in

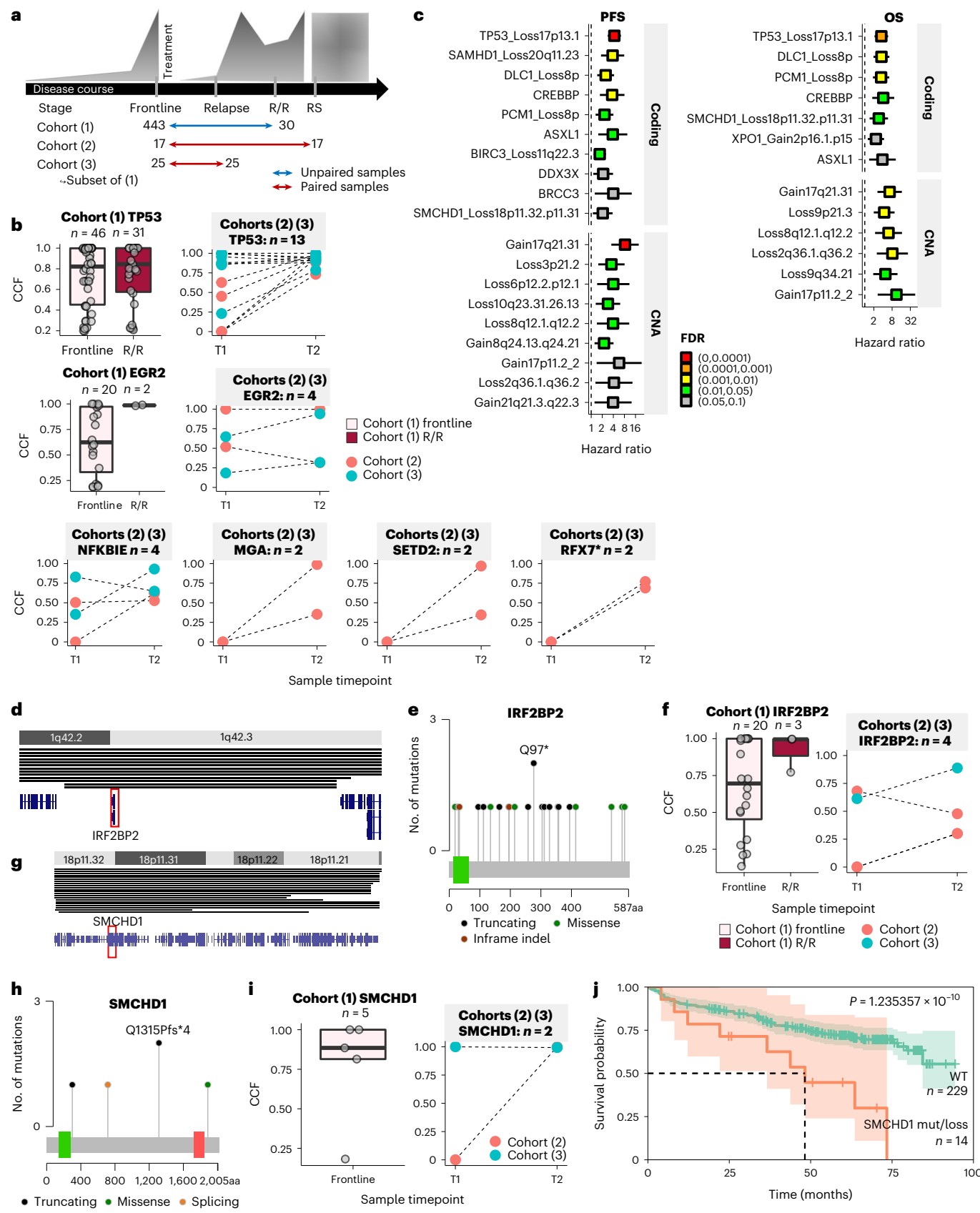

the previously reported *PAX5* superenhancer[3,6,35] (Extended Data Fig. 5a and Supplementary Fig. 4b). Another region spanning 325 kb on chr3q27.2 contained seven significantly mutated enhancers and linked to *BCL6* (Extended Data Fig. 5a and Supplementary Table 17). RNA-seq of eight samples with mutations in this region showed overall increased expression of *BCL6*, although the effect was heterogenous (Fig. 5c), suggesting that some variants are more or less pathogenic than others and variants might exert a positional effect (Fig. 5d).

When considering the 72 significantly mutated promoters, we found mutations of known CLL drivers including *BIRC3* ($n = 31$, 6.4%, FDR < 1.15 ×10$^{-15}$), *IKZF3* ($n = 12$, 2.5%, FDR = 8.16 × 10$^{-13}$) and *TP53* affecting splicing regions of noncoding exons/5′ UTR/promoter region ($n = 2$, 0.4%, FDR = 5.55 × 10$^{-6}$). Next, we investigated mutations in these promoters further to identify those predicted to change chromatin state, using DeepHaem[36], a deep neural network trained on chromatin feature data of 73 immune cell types. Seventy-four variants were predicted to lead to a loss of open chromatin (that is, loss-of-function variants), including those in the *BACH2* promoter (Fig. 6a and Extended Data Fig. 6a). A recent study showed that decreased BACH2 expression in CLL is associated with adverse outcomes[37]. Notably, the mutations we detected in this promoter were mostly clonal (median CCF = 0.99). We therefore investigated this promoter further by performing ATAC-seq and RNA-seq (Fig. 6b) on mutated samples, when available (13 variants investigated, Supplementary Table 12; Methods) to understand the impact of these variants on chromatin accessibility and gene expression. Three variants within a 14-bp region were associated with allelic skew in the ATAC-seq compared with WGS data, demonstrating a preference for accessibility on the reference allele (Fig. 6c), which mirrored the decrease in chromatin accessibility in that region compared with WT samples (Fig. 6d). This allelic skew was also detected at the RNA level (Fig. 6e and Extended Data Fig. 6b). In addition, the same three samples also showed decreased BACH2 gene expression (Fig. 6f).

Finally, we analyzed 20 cases with paired WGS, ATAC-seq and RNA-seq data (Supplementary Table 4). We identified five recurrently mutated promoters with allelic skewing of chromatin accessibility and RNA expression. Three, *BTG2*, *CCND1* and *ST6GAL1*, were associated with allelic skewing towards the mutant allele, whereas *ATAD1* and *BIRC3* showed the opposite effect (Extended Data Fig. 6c). In the case of *ATAD1*, which plays a role in mitochondria protein degradation, we additionally observed reduced expression in promoter-mutated samples ($P = 7.0 × 10^{-4}$) (Extended Data Fig. 6d–f).

Collectively, these data suggest that a small subset of the noncoding mutations in CLL have characteristics indicative of a driver and target regulatory elements of genes that are critical for B cell development and function as well as cancer progression. However, the effects on chromatin accessibility and gene expression levels were subtle and require further indepth functional characterization.

## Clinical impact of combined and global genome features

We recalculated the occurrence of mutations in each known or putative driver in CLL by combining coding mutations, noncoding mutations in regulatory elements and CNAs (Fig. 7a and Supplementary Table 19). In total, 33 of the 58 coding, known or putative driver genes were also affected by noncoding mutations in associated regulatory elements or by CNAs. Overall, 412 (29%) of all alterations in these genes were either CNAs or affected regulatory elements. *ATM* and *BIRC3* were most frequently targeted by genetic lesions. The median number of mutated known or putative drivers in each tumor was 2 (0–7) or 5 (0–21) when excluding or including CNA/copy neutral loss of heterozygosity (cnLOHs) and noncoding variants, respectively (Fig. 7b). A higher number of mutated genes was associated with worse PFS, especially when noncoding variants were included (Extended Data Fig. 7a,b and Supplementary Tables 15 and 16). Furthermore, the number of samples containing mutations in particular pathways also increased (by 3.3%) (Fig. 7c and Supplementary Fig. 5), in particular for the NOTCH and the transcriptional regulations pathways.

We explored whether global genomic features could also be associated with clinical outcome. Firstly, we evaluated telomere length and observed that it was reduced in CLL samples compared with paired germline (median length of 2.7 kb versus 3.8 kb, $P < 2.2 × 10^{-16}$, median content of 405 versus 467, $P = 3.9 × 10^{-6}$, paired Wilcoxon test) (Fig. 7d and Extended Data Fig. 8a,b). Shorter telomeres were significantly enriched in samples with p53 pathway alterations ($P = 1.99 × 10^{-36}$; Fig. 7d), with R/R samples compared with frontline (FDR = 5.37 × 10$^{-7}$; Supplementary Table 14) and were associated with poorer PFS (FDR = 4.39 × 10$^{-4}$; Supplementary Table 15 and Extended Data Fig. 8c,d).

Secondly, we explored the clinical associations of mutation signatures including single base substitution (SBS), doublet base substitutions (DBS) and small insertions and deletions (ID)[38] (Fig. 7e,f and Supplementary Table 20). Considering signatures with known or probable etiology, the most prevalent were SBS5 (clock-like), DBS11 (APOBEC activity) and ID2 followed by other clock-like signatures: SBS1 (deamination of 5-methylcytosines), SBS8, DBS2 and the AID signature SBS9. As previously documented, SBS9 was highly enriched in m-IGHV CLLs (FDR = 4.80 × 10$^{-57}$, Fisher's exact test; Supplementary Table 14), was mutually exclusive with *TP53* alterations (2.29 × 10$^{-3}$) and associated with good PFS (Supplementary Table 15 and Extended Data Fig. 8e). De novo signature ID83C was found associated with *TP53* alterations (FDR = 2.53 × 10$^{-2}$; Supplementary Table 14) and poorer PFS (1.57 × 10$^{-2}$; Extended Data Fig. 8f and Supplementary Table 15). SBS1 was also associated with adverse outcome (3.70 × 10$^{-2}$; Supplementary Table 15 and Extended Data Fig. 8g).

Thirdly, we analyzed GC using unsupervised clustering (multiple correspondence analysis (MCA)) of 17 features related to CNAs (Extended Data Fig. 9a,b; Methods). These defined eight groups (GC1–GC8) (Extended Data Fig. 9c,d) with distinct genomic profiles

---

**Fig. 4 | Significantly mutated noncoding REs. a**, Methodology to localize noncoding REs in CLL primary cells. These REs were defined across the whole genome based on chromatin state data from CLL primary cells. We intersected H3K27ac peaks and open chromatin regions defined by ATAC-seq (derived from 104 and 106 primary CLL, respectively)[27]. Next, these regions were annotated using genome-wide segmentations of seven CLL samples (five mutated and two unmutated IGHV cases) with available chromatin immunoprecipitation followed by sequencing (ChIP–seq) data of six histone marks including H3K4me3, H3K4me1, H3K27ac, H3K36me3, H3K27me3 and H3K9me3. As our annotations of noncoding variants were based on CLL samples from different cohorts, chromatin states defined by ChIP–seq were considered only for regions that were seen in at least two samples. Common regions based on shared overlaps were used to define these REs. REs active exclusively in samples with m-IGHV and u-IGHV mutational status were also defined. REs were linked to target genes by correlating RNA expression (gene) and H3k27ac (REs) (Pearson correlation 0.3, FDR ≤ 0.05), within topologically associated domains of GM12878 defined by Hi-C[30]. For additional annotations and more details, see Methods. **b**, Candidate noncoding drivers including UTRs, promoters and enhancers affected by SNVs/indels, were revealed using several discovery algorithms and regions with FDR below the significance threshold were selected. The presence of single-site hotspots, and regions with high mutational density/kataegis were reported and regions with FDR below the significance threshold were selected. Annotations and postfiltering of somatic noncoding hits were including immunoglobulin loci and known false positive exclusion, AID and APOBEC signature annotations, and additional genomic and functional annotations from the literature. **c**,**d**, Significantly mutated REs for which target genes are CLL drivers or in the COSMIC database (**c**) or other genes (**d**). Upper panel, number of samples mutated; middle panel, proportion of variants with signature attributed to AID, APOBEC or other processes; lower panel, FDR of the likelihood these regions as mutated more frequently than expected. **e**, Gene set enrichment analysis based on the target genes of all noncoding candidate drivers for gene ontology terms biological process (GO:BP) and human phenotype ontology (HP). We applied a hypergeometric test and multiple testing correction of *P* value using the g:SCS algorithm[41].

(Fig. 7g and Extended Data Fig. 9e). GC4 (presenting CN losses only, $n = 210$) was enriched in del13q14.2 (FDR = $3.26 \times 10^{-23}$). GC7 (presenting both CN gains and losses, $n = 127$) was associated with ten recurrent CNAs and seven known coding drivers including *XPO1* (FDR = $3.98 \times 10^{-11}$) and *TP53* (FDR = $8.36 \times 10^{-9}$). Together with GC8 (presenting trisomy, CN gains and losses, $n = 15$), GC7 comprised the most patients with conventional genomic complexity, defined by the presence of at least four CNAs (Extended Data Fig. 9f). None of the

genomic complexity groups was significantly enriched in stereotyped subsets (Extended Data Fig. 9g). For the subset of samples with survival data ($n = 243$), we combined genomic complexity groups with copy number gains only, copy number losses only and both copy number gains and losses to increase statistical power. Interestingly, the eight groups were associated with different PFS and OS (Extended Data Fig. 10a,b), independent of *TP53* status (Extended Data Fig. 10c,d). Furthermore, patients with both *TP53* mutations and GC7/8

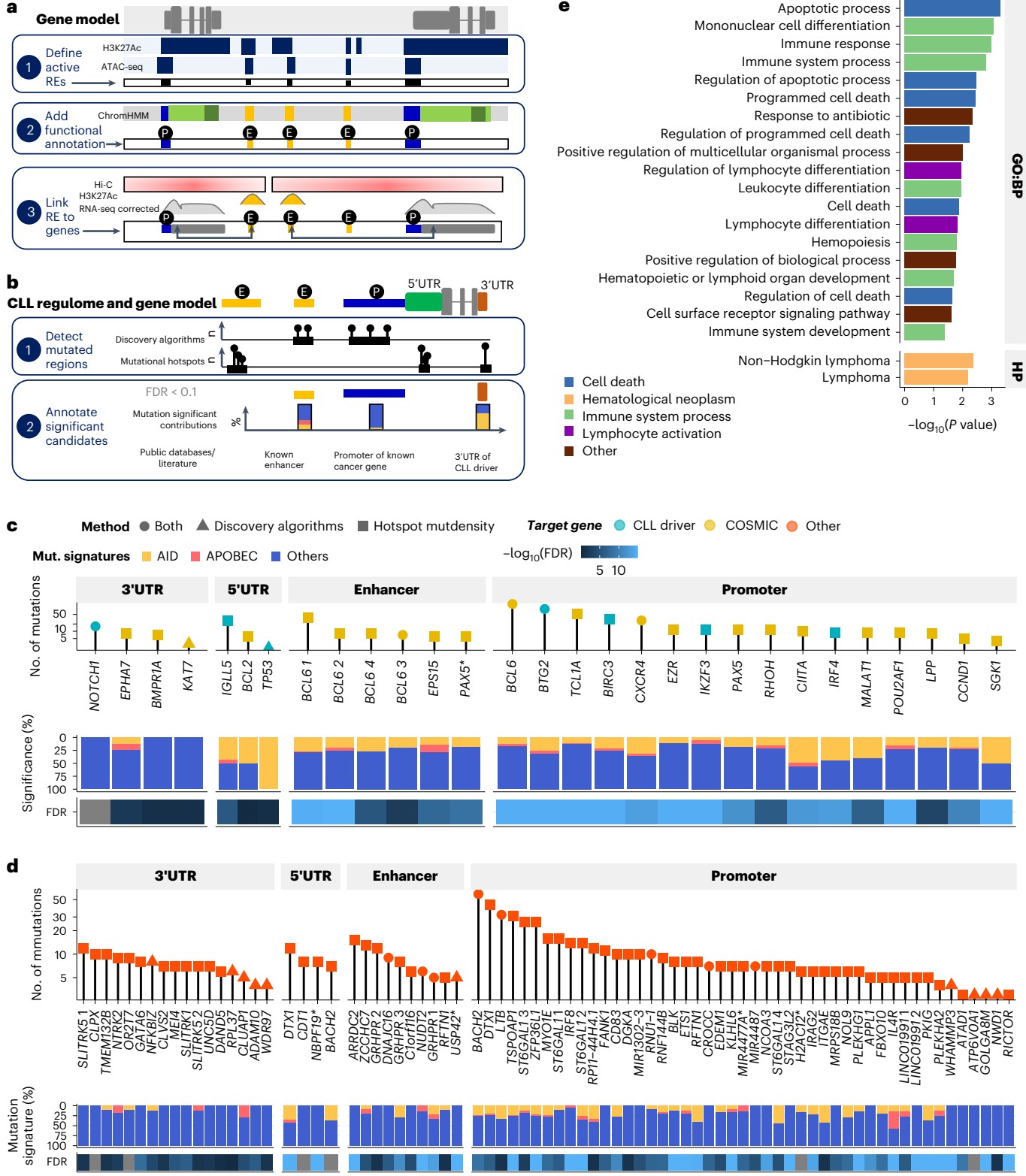

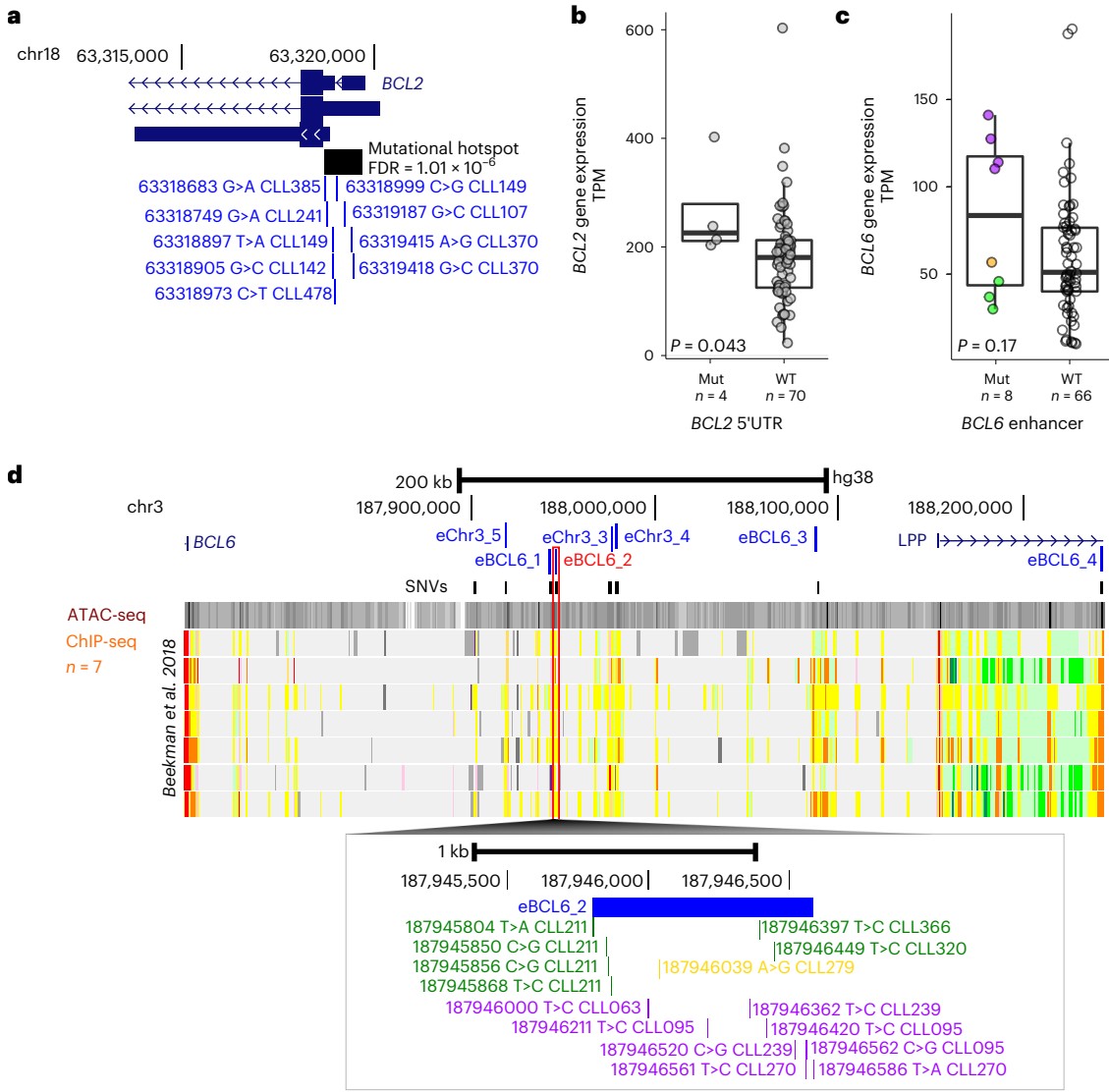

**Fig. 5 | Noncoding mutations impacting BCL genes. a**, Genome view of BCL2 5′ UTR. The significantly mutated region is indicated by a black rectangle. Individual somatic mutations are shown in blue. **b**, Gene expression of *BCL2* in TPM determined by RNA-seq in samples with *BCL2* 5′ UTR mutations versus WT. Black dots are marks as outliers. *P* value was derived from a two-sided Welch's *t*-test. **c**, Gene expression in TPM determined by RNA-seq of *BCL6* in samples with *BCL6* enhancer mutations versus WT. Expression levels were split in low (less than median expression; green), medium (between median expression and 100; orange) and high (≥100 TPM; purple). *P* value was derived from a two-sided Wilcoxon test. **d**, Genome view of the *BCL6* gene and enhancers. Enhancers within these regions are annotated in blue font. eBCL6_2, which was the target of several variants is indicated in red. Annotation tracks of ATAC-seq and ChIP−seq are from publicly available RE annotation detailed above[27] (references containing detailed of datasets and figure legends). The lower panel shows the individual mutations color coded as defined in **c**. All boxplots show the minimum and maximum values and interquartile range.

changes had ultrahigh-risk disease (median PFS = 8 months, median OS = 15 months) and fared worse compared with patients with *TP53* mutations but no GC7/8 status (*P* = 0.03; Fig. 8a,b).

### Towards a patient classifier
To evaluate the potential clinical relevance of combining different genomic features, we first used penalized multivariate regression analysis for least absolute shrinkage and selection operator. This analysis led to the identification of 56 individual genomic features that predicted PFS and/or OS including *SMCHD1*/del18p11.32-p11.31, which retained significance as an independent predictor of OS (Extended Data Fig. 10e and Supplementary Fig. 6a).

Next, we applied non-negative matrix factorization (NMF) to identify robust subgroups of CLLs sharing subsets of the 186 different genetic alterations (Supplementary Table 21; Methods). Considering

the profound clinical impact of the IGHV mutational status, we initially divided patients into m-IGHV and u-IGHV. Using this approach, we identified five distinct GS: three were u-IGHV (u-GS1, 2 and 3) and two m-IGHV (m-GS1 and 2) (Fig. 8c,d and Supplementary Table 22).

When considering u-IGHV CLL (Fig. 8c and Supplementary Table 23), u-GS1 was characterized by the presence of high-risk features including *TP53* disruption, GC7, short telomeres and mutations in targetable pathways such as MAPK, PI3K and apoptosis, but there was no DNA damage response signature. By contrast, u-GS2 was defined by *ATM/BIRC3*/del11q22.2-22.3 alterations, as well as mutations in DNA damage response pathways, but without *TP53* mutations or genomic complexity as defining features. Patients in u-GS2 were predominately male. u-GS3 had a high number of mutations in known and putative coding drivers, introns and UTRs, CN gains including trisomy 12, *NOTCH1* mutations, and was enriched for older patients. All three subgroups

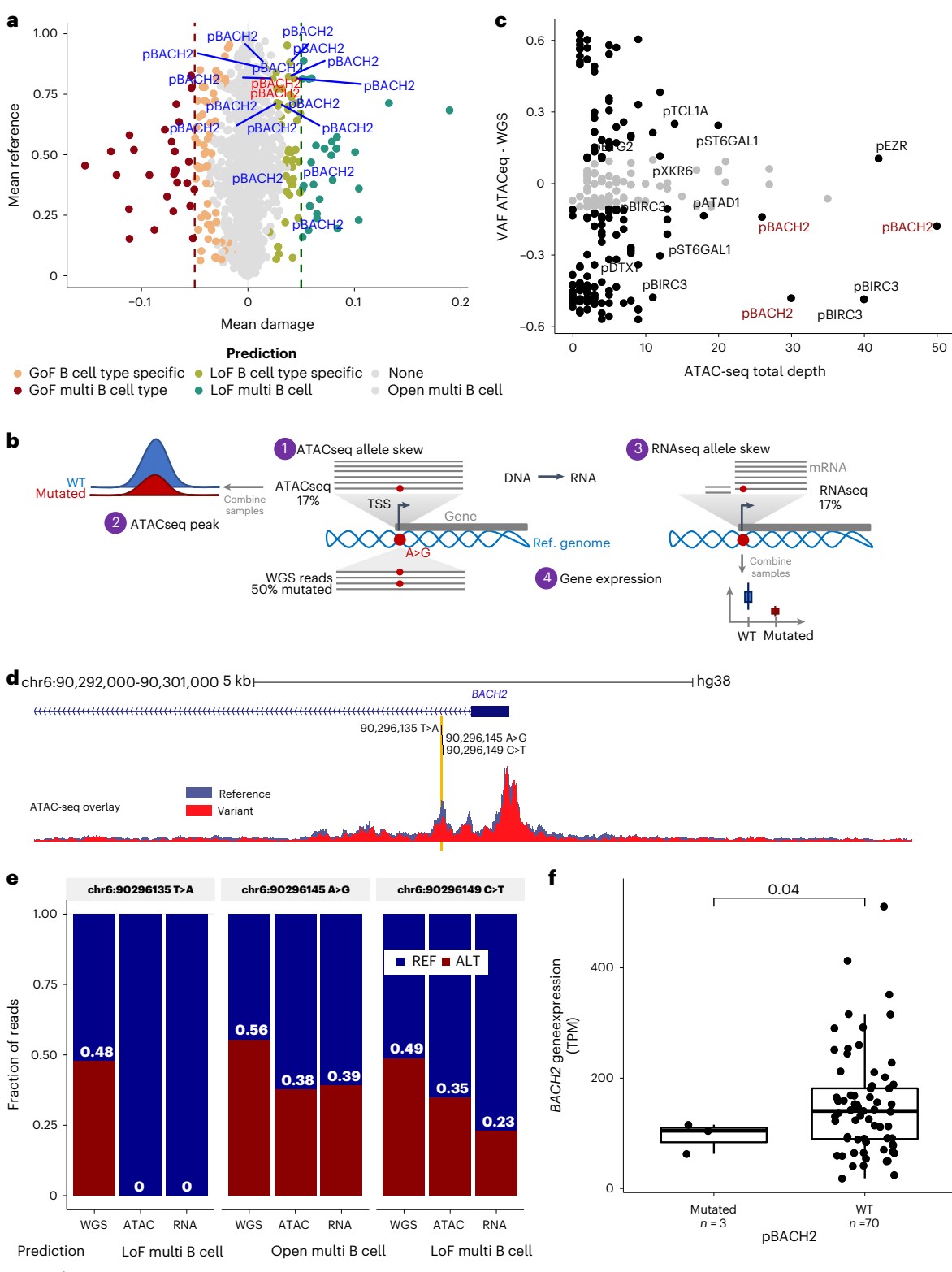

included patients with BCR IG subsets 1 and 8, which are known to be associated with aggressive disease[39] (Supplementary Fig. 6b). Although u-GS2 and u-GS3 were clearly distinct, they were associated with similar PFS after chemoimmunotherapy (Fig. 8e).

Regarding m-IGHV CLL (Fig. 8d), m-GS1 was similar to u-GS1 (cosine similarity of 0.81) and also to u-GS2 (cosine similarity of 0.7) (Supplementary Table 24). In contrast, m-GS1 was enriched for older men, BCR IG subset 2 (FDR = 2.96 × 10⁻⁶) and IGHV3-21 (FDR = 7.50 × 10⁻⁹)

(Supplementary Fig. 6b), although most patients in m-GS1 did not have any defined CLL stereotype. m-GS2 had high mutation burden in enhancers, UTRs and promoters, was enriched for del13q4.2 but no other CNAs and had longer telomeres compared with the mean length in CLL. Additional clustering (Methods) further refined m-GS2 into distinct two clusters (Supplementary Fig. 6c). m-GS2 cluster 1 stood out by the high frequency of SBS9, the presence of GC4 and the absence of any other features. In comparison, m-GS2 cluster 2 had

**Fig. 6 | Mutations in the promoter of BACH2 associated with reduction of chromatin accessibility and RNA expression. a**, Prediction of the impact of noncoding mutations in promoters on transcription factor binding from cell- and tissue-specific DNase footprints. Mutations in *BACH2* promoters are annotated (blue), specific *BACH2* promoters detailed later are in red. GoF/LoF, gain/loss of function; B cell type specific, prediction observed in dataset examined; multi-B cell type, prediction observed in several dataset examined (robust); open multi-B cell, open chromatin region predicted; none, no prediction. **b**, Methodology to explore the effect of *BACH2* promoter mutations. (1) We compared VAF of WGS data and ATAC-seq data to find allelic skew, that is, a preference for accessibility on the reference or the mutant allele. TSS, transcription start site. (2) We examined the change in chromatin accessibility in regions of interest in mutated compared with WT samples. (3) We compared VAF of WGS data and RNA-seq data to find allelic skew, that is, a preference for RNA expression on the reference or the mutant allele. (4) We compared the gene expression in mutated versus WT

samples by RNA-seq. **c**, Prioritization of noncoding variants based on sequencing depth at the loci in the ATAC-seq data and allelic skew between the ATAC-seq and WGS data. Datapoints with difference less then −0.1 or greater than 0.1 and sequencing depth of at least ten times are annotated in black font. The *BACH2* promoter is indicated in red font. **d**, ATAC-seq signal at the promoter of *BACH2*. The blue track shows the combined signal from all 24 patient samples; overlaid is the signal from a sample (pink) with a variant in the center of the RE. The location of variants in the same RE from three other patients are highlighted. **e**, Fraction of mutant and WT read in three *BACH2* promoter variants showing allelic skew in ATAC-seq and RNA-seq compared with WGS. Prediction and mean damage scores were calculated with DeepHaem. **f**, Gene expression distribution (the minimum and maximum values and interquartile range) of *BACH2* in TPM determined by RNA-seq in samples with promoter mutations versus sample WT. The statistical test used was a two-sided Welch's *t*-test.

---

*MYD88* mutations, trisomy 12 and other CN gains but no CN losses (Supplementary Table 25). Both clusters of m-GS2 had a very favorable PFS of 75% and showed a plateau of PFS, implying cure after chemo-immunotherapy (Fig. 8f). By contrast, patients belonging to m-GS1 had a shorter PFS than u-GS2/u-GS3 (median PFS = 38 versus 50 months; Fig. 8e) and there was no plateau.

In our analysis of patients treated with chemoimmunotherapy, NMF subgroups could not be defined without the different acquired local and global noncoding genomic changes, since combining all known coding drivers and the four common recurrent CNAs did not cluster patients into the GSs (Supplementary Figs. 6d and 7). Based on this observation, we examined whether the NMF method could be used to prospectively and precisely assign individual patients into their subgroup for individualized outcome prediction in the clinic. Our validation, performed by subsetting the dataset (Methods) showed that a total of 15/16 m-IGHV samples and 48/51 of u-IGHV samples were assigned correctly to their respective subgroup (Fig. 8g).

## Discussion

Our study presents the first comprehensive WGS analysis of a large series of CLL patients requiring treatment. A main strength of our study is that it is based on patients enrolled into multicenter clinical trials, thereby reducing heterogeneity. This allowed us to not only define the genomic landscape of different stages of CLL[3,4], but also to identify mutations associated with disease relapse and transformation.

Based on a strict pipeline for discovery of coding drivers, we selected the top ranked recurrently mutated genes, which comprised 36 known CLL drivers[3,6,7] and 22 putative drivers. Only 32% of variants in those putative driver genes were missense variants, with most being truncating and stop-gain mutations. Although these putative drivers shared characteristics of known drivers (that is, damaging mutations in protein domains, impact on RNA expression, high CCF that further increased at disease progression, association with survival), we cannot exclude the possibility that some may simply represent passengers.

We defined recurrent translocations (with breakpoints in *WDHD1; CTNND2-ARHGAP18*) and 126 candidate noncoding drivers within REs

pinpointing potentially druggable target genes (*NOTCH1, DTX1, NFKBIZ, NTRK2* and *BACH2*). For a small subset of selected noncoding candidate mutations, we were able to demonstrate a modest impact on chromatin accessibility and/or target gene expression (5′ UTR of *BCL2*, enhancer of *BCL6*, promoter of *BACH2* and promoter of *ATAD1*).

Exploring different layers of genomic data including coding, non-coding and genome-wide global changes allowed us to (1) derive a WGS-derived genomic complexity classification that further refines risk by identifying an independent ultrahigh-risk group associated with complex genomic alterations (GC7/8); (2) more precisely predict individual patients who achieve a plateau after chemoimmunotherapy (m-GS2) and are functionally cured, thereby clearly differentiating them from progressors in the m-GS1 subgroup.

Ideally, only genomic features experimentally validated as disease drivers should be included in any prognostic classification system, even if they were selected by very stringent criteria as those applied in this study (see above). However, it is well recognized that some genomic features are clearly not disease drivers, yet carry prognostic relevance. For example, in CLL, the IGHV mutation status representing the cell-of-origin or telomere length reflecting proliferative activity, are associated strongly with clinical outcome, but are not considered disease drivers.

In our NMF model using only the known coding drivers and recurrent CNAs did not allow us to recover the same level of discrimination as that afforded by inclusion of additional local and global noncoding information. This observation implies that the combination of coding and noncoding information in the classifier increases the precision of clinical risk prediction at least in our cohort of clinical trial patients.

Although treatment algorithms for CLL are shifting away from chemoimmunotherapy to targeted agents, the subgroups we define remain potentially clinically relevant as they reflect distinct biological entities. Collectively, our study provides a springboard for down-stream functional analyses of putative coding and noncoding drivers. Robust testing on independent cohorts of patients undergoing targeted therapy will be required to further establish the clinical utility of this WGS-based classifier.

---

**Fig. 7 | Data integration and genome-wide global lesions. a**, Distribution of the type of alterations in CLL coding drivers affected only by coding mutations (top panel) and affected by coding, CNAs and/or mutations in their REs (bottom panels). **b**, Distribution of the number of mutations per sample when considering all functional mutations (blue shading), SNVs/indels in coding drivers (green shading) and coding and noncoding drivers and CNAs (purple shading). **c**, Proportion of samples with mutated pathways, when considering coding drivers only (green), coding drivers and other genes with high impact mutations (involving frameshift and stop coding mutations (yellow)) and all coding as well as noncoding drivers (red). **d**, Telomere lengths distribution (showing the minimum and maximum values and interquartile range) in normal samples

and matched CLL samples. Lines link matched tumor-normal datapoints. Significance level shows two-sided paired Wilcoxon test of *P* value <0.001. **e**, Fraction of each mutational signature detected in different genomic scopes: the 58 coding drivers; exonic regions; promoters, enhancers and UTRs; and whole genome. **f**, Fraction of each mutational signature detected in each coding driver. DBS not shown as data were too scarce. **g**, Distribution of the eight GC groups, based on presence (dark gray) or absence (light gray) of the three variables selected as best predictor by MCA: CN losses (Loss), CN gains (Gain) and trisomy (Tri). *TP53* alteration status and conventional GC status are indicated in the top panel.

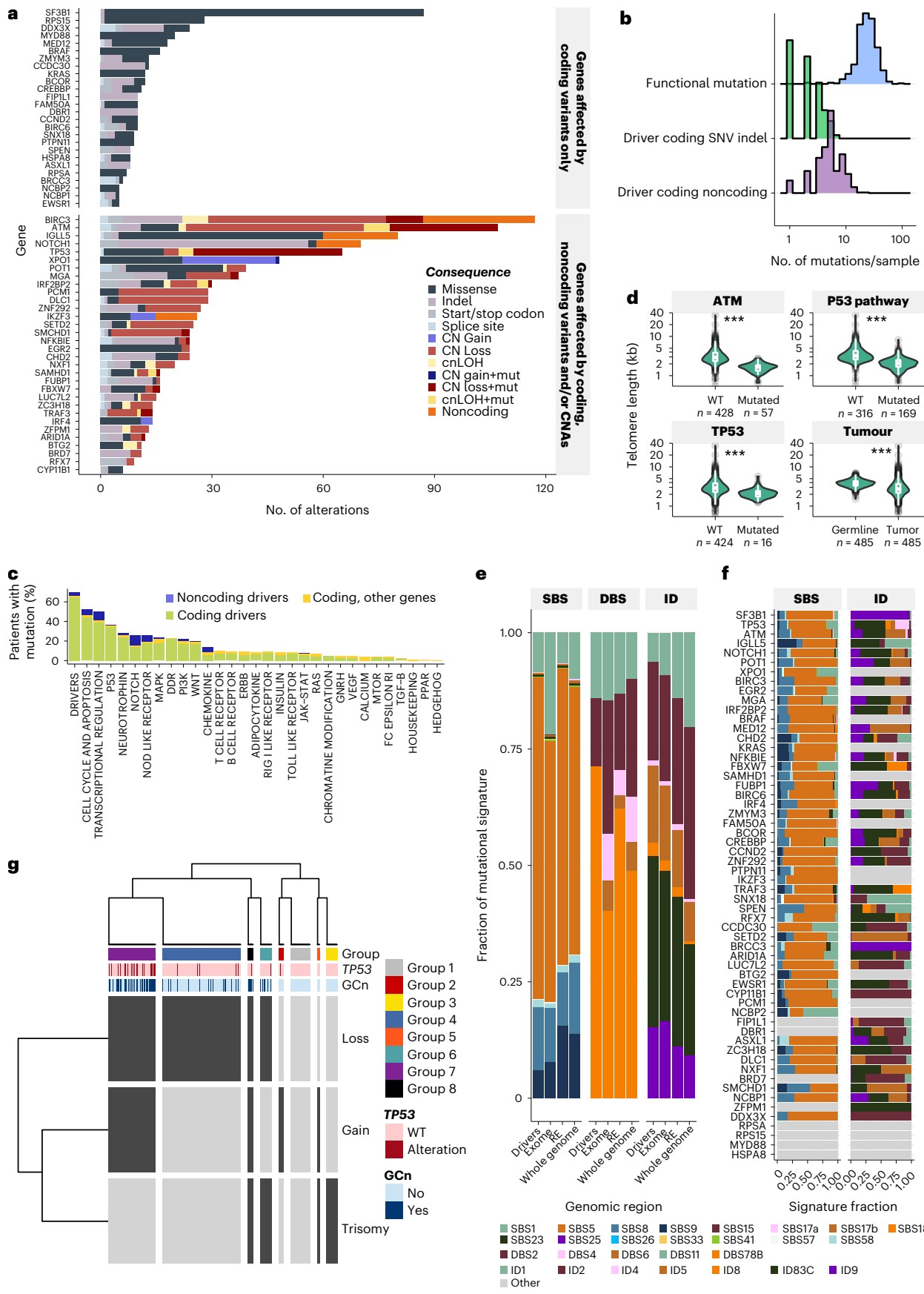

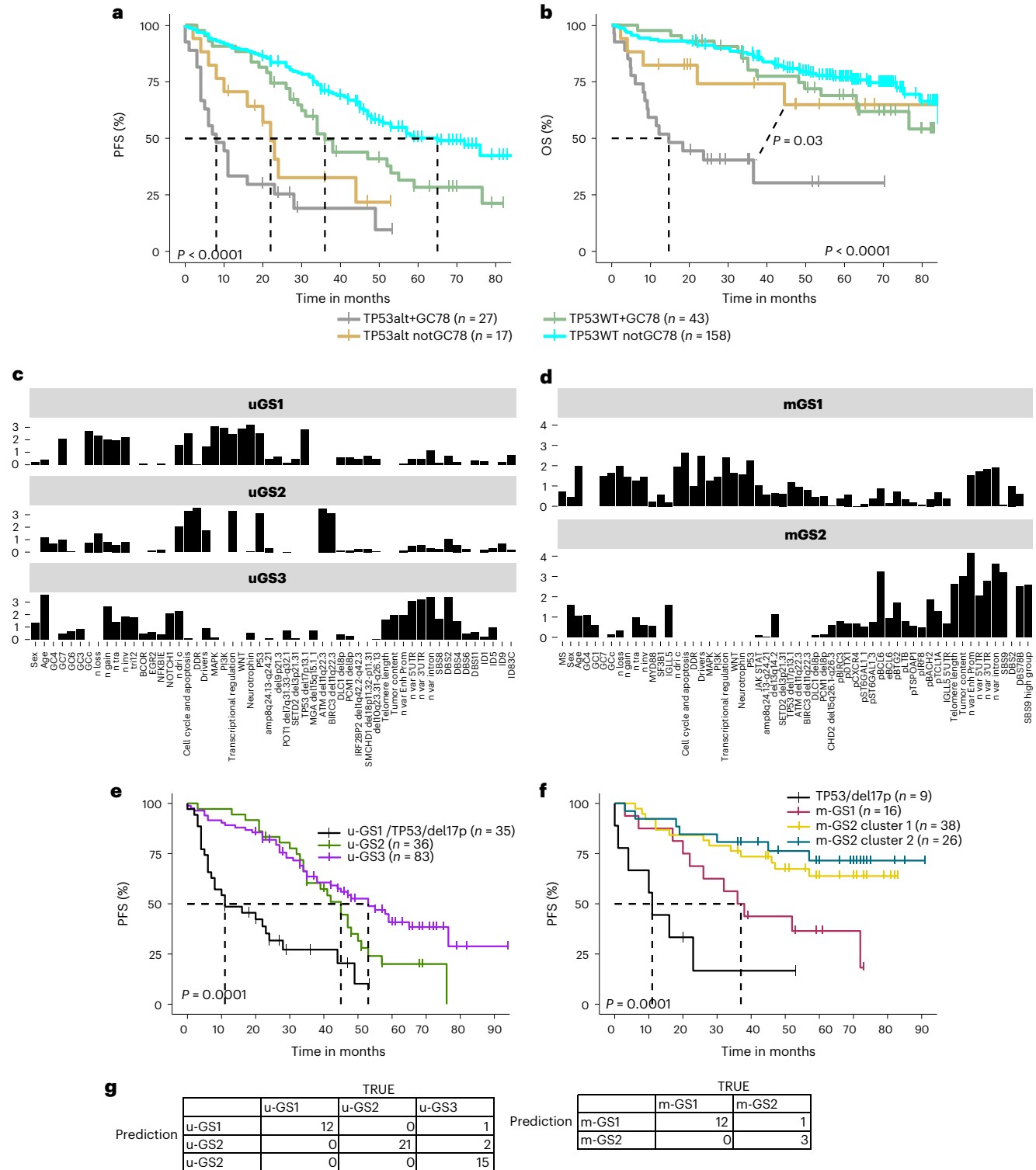

**Fig. 8 | Relationship between genomic features and patient outcome.**
**a,b**, Kaplan–Meier curve on PFS (**a**) and OS (**b**) of *TP53* altered/WT in combination with GC7/8. The *P* value was derived from a log-rank test comparing the most two extreme curves (additional data in Extended Data Fig. 10). The dotted lines indicate the median survival for each subgroup. **c,d**, Genomic factors comprising the GS (cut-off 0.5) derived using non-negative matrix factorization, hypermutated subset (u-GS) (**a**), unmutated subset (m-GS) (**b**). The plot only shows features that split the data. **e,f**, Kaplan–Meier curves of PFS of samples divided by GS. Only samples with PFS data were included (*n* = 243). In **e**, the unmutated subset, del17p/TP53 mutated samples are plotted separately (black curve), all u-GS1 cluster 1 samples fell into this grouping; In **f**, the hypermutated

samples, del17p/TP53 mutated samples are plotted separately (black curve). The *P* value was derived from a log-rank test comparing the most two extreme curves. The dotted lines indicate the median survival for each subgroup **g**, Confusion matrix showing agreement between true and predicted subgroup assignment. The true subgroup assignment was determined by applying the previously described NMF approach (Methods) to the whole set of genomic data. The predicted subgroup assignment was determined by first using 80% of the genomic data for subgroup assignment (training phase) followed by predicting the subgroup assignment in the remaining 20% of the data (testing). In all cases, sex and age were included to inform the model (Methods).

## Online content

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

[1]Department of Oncology, University of Oxford, Oxford, UK. [2]RIKEN Center for Integrative Medical Sciences, Yokohama, Japan. [3]Division of Genetics and Epidemiology, The Institute of Cancer Research, Sutton, UK. [4]Department of Medicine, Medical Research Council Molecular Haematology Unit, Medical Research Council Weatherall Institute of Molecular Medicine, University of Oxford, Oxford, UK. [5]UK Health Security Agency, London, UK. [6]Department of Haematology, University Hospital Limerick, Limerick, Ireland. [7]Limerick Digital Cancer Research Centre, School of Medicine,University of Limerick, Limerick, Ireland. [8]Laboratory of Clinical Cell Therapy, Jules Bordet Institute, ULB Cancer Research Center (U-CRC)- Université Libre de Bruxelles (ULB), Brussels, Belgium. [9]Oxford Molecular Diagnostics Centre, John Radcliffe Hospital, Oxford University Hospitals NHS Trust, Oxford, UK. [10]Department of Medical Laboratory Technology, King Abdulaziz University, Jeddah, Saudi Arabia. [11]Illumina Cambridge Ltd., Cambridge, UK. [12]University of Liverpool, Liverpool, UK. [13]Department of Haematology, CHU de Clermont-Ferrand, Clermont-Ferrand, France. [14]Cancer Sciences, Faculty of Medicine, University of Southampton, Southampton, UK. [15]Biomedical Epigenomics Group, Institut d'Investigacions Biomédiques August Pi i Sunyer (IDIBAPS), University of Barcelona, Barcelona, Spain. [16]Human Technopole, Milan, Italy. [17]Oxford University Clinical Academic Graduate School, University of Oxford Medical Sciences Division, University of Oxford, John Radcliffe Hospital, Oxford, UK. [18]Center for Hematology and Regenerative Medicine Huddinge, Karolinska Institute, Stockholm, Sweden. [19]Cancer Genomics, Cancer Sciences, Faculty of Medicine, Group University of Southampton, Southampton, UK. [20]King's College Hospital, NHS Foundation Trust, London, UK. [21]Kings College London, London, UK. [22]Department of Molecular and Clinical Cancer Medicine, University of Liverpool, Liverpool, UK. [23]Clatterbridge Cancer Centre NHS Foundation Trust, Liverpool, UK. [24]St James's University Hospital, Leeds, UK. [25]Genomics England, London, UK. [26]William Harvey Research Institute, Queen Mary University of London, London, UK. [27]Institució Catalana de Recerca i Estudis Avançats (ICREA), Barcelona, Spain. [43]These authors contributed equally: Pauline Robbe, Kate E. Ridout. *Lists of authors and their affiliations appear at the end of the paper. ✉e-mail: anna.schuh@oncology.ox.ac.uk

## Genomics England Research Consortium

J. C. Ambrose[25], P. Arumugam[25], R. Bevers[25], M. Bleda[25], F. Boardman-Pretty[25,26], C. R. Boustred[25], H. Brittain[25], M. A. Brown[25], Marc J. Caulfield[25,26], G. C. Chan[25], T. Fowler[25], A. Giess[25], A. Hamblin[25], S. Henderson[25,26], T. J. P. Hubbard[25], R. Jackson[25], L. J. Jones[25,26], D. Kasperaviciute[25,26], M. Kayikci[25], A. Kousathanas[25], L. Lahnstein[25], S. E. A. Leigh[25], I. U. S. Leong[25], F. J. Lopez[25], F. Maleady-Crowe[25], M. McEntagart[25], F. Minneci[25], L. Moutsianas[25,26], M. Mueller[25,26], N. Murugaesu[25], A. C. Need[25,26], P. O'Donovan[25], C. A. Odhams[25], C. Patch[25,26], D. Perez-Gil[25], M. B. Pereira[25], J. Pullinger[25], T. Rahim[25], A. Rendon[25], T. Rogers[25], K. Savage[25], K. Sawant[25], R. H. Scott[25], A. Siddiq[25], A. Sieghart[25], S. C. Smith[25], Alona Sosinsky[25,26], A. Stuckey[25], M. Tanguy[25], A. L. Taylor Tavares[25], E. R. A. Thomas[25,26], S. R. Thompson[25], A. Tucci[25,26], M. J. Welland[25], E. Williams[25], K. Witkowska[25,26] & S. M. Wood[25,26]

## CLL pilot consortium

James Allan[28], Niamh Appleby[1], Garry Bisshopp[29], Stuart Blakemore[30], Jacqueline Boultwood[31], David Bruce[1], Francesca Buffa[31], Andrea Buggins[21], Adam Burns[1], Ruth Clifford[6,7], Gerald Cohen[32], Kate Cwynarski[33], Claire Dearden[34], Stephen Devereux[20,21], Richard Dillon[21], Sarah Ennis[30], Francesco Falciani[32], George Follows[35], Francesco Forconi[30], Jade Forster[30], Christopher Fox[36], Jane Gibson[14], John Gribben[26], Peter Hillmen[24], Anna Hockaday[37], Richard S. Houlston[3], Dena Howard[37], Andrew Jackson[38], Nagesh Kalakonda[32], Umair Khan[32], Philip Law[39], Pascal Lefevre[37], Ke Lin[32], Sandra Maseno[9], Paul Moss[38], Melanie Oates[12], Graham Packham[30], Claire Palles[38], Helen Parker[30], Piers Patten[20], Andrea Pellagatti[31], Andrew R. Pettitt[22,23], Guy Pratt[40], Alan Ramsay[20], Andy Rawstron[41], Kate E. Ridout[1,43], Pauline Robbe[1,2,43], Matthew Rose-Zerilli[30], Anna Schuh[1], Joseph Slupsky[32], Tatjana Stankovic[38], Andrew Steele[30], Jonathan Strefford[19], Shankar Varadarajan[32], Dimitrios V. Vavoulis[2], Simon Wagner[42], David Westhead[37], Sarah Wordsworth[31] & Jack Zhuang[32]

[28]University of Newcastle Upon Tyne, Newcastle Upon Tyne, UK. [29]Brighton and Sussex University Hospitals, Brighton, UK. [30]University of Southampton, Southampton, UK. [31]University of Oxford, Oxford, UK. [32]University of Liverpool, Liverpool, UK. [33]University College London Hospitals, London, UK. [34]Royal Marsden NHS Foundation Trust, London, UK. [35]Cambridge University Hospitals NHS Foundation Trust, Cambridge, UK. [36]Nottingham University Hospitals NHS Trust, Nottingham, UK. [37]University of Leeds, Leeds, UK. [38]University of Birmingham, Birmingham, UK. [39]Division of Genetics and Epidemiology, The Institute of Cancer Research, Surrey, UK. [40]University Hospitals Birmingham NHS Foundation Trust, Birmingham, UK. [41]Leeds Teaching Hospitals NHS Trust, Leeds, UK. [42]University of Leicester, Leicester, UK.

## Methods

### Patient cohorts, samples and ethics

All patients gave written informed consent and the study was approved under the 100,000 Genomes Project Ethics and the CLL Pilot ethics (MREC 09/H1306/54). A total of 485 patients with CLL were included in the study. A small subset was enrolled into CLEAR (CLL Empirical Antibiotic Regimen, early stage of the disease, n = 12, NCT01279252) and CLL210 (ref. [42]) (relapsed/refractory patients, n = 30, EudraCT 2010-019575-29). All other patients were treatment-naïve and required treatment according to iwCLL criteria[43]. They were either fit patients receiving frontline treatment with fludarabine, cyclophosphamide, rituximab (FCR)-based treatment in ARCTIC[44] (Attenuated dose Rituximab with ChemoTherapy In CLL, n = 61, EudraCT Number:2009-010998-20) or AdMIRe[45] (Does the ADdition of Mitoxantrone Improve REsponse to FCR chemotherapy in patients with CLL, n = 65, EudraCT number: 2008-006342-25) or frail patients receiving ofatumumab with either bendamustine or chlorambucil chemoimmunotherapy in RIAltO (A Trial Looking at Ofatumumab for People With Chronic Lymphocytic Leukemia Who Cannot Have More Intensive Treatment, n = 92, NCT01678430). Patients recruited into FLAIR[46] (Front-Line therapy in CLL: Assessment of Ibrutinib + Rituximab, n = 225, EudraCT 2013-001944-76) were randomized to ibrutinib alone or in combination with rituximab or venetoclax or standard first-line FCR treatment. In line with the studies' data monitoring committees, baseline characteristics and clinical outcomes data were available only from studies once closed to recruitment (see Supplementary Table 1 for details of all patients recruited into the 100,000 Genomes Project). For patients recruited into the FLAIR study, these data are still awaited.

For a subset of 25 patients, we obtained a sample taken at relapse (Supplementary Table 4).

To investigate findings in more advanced disease, we reanalyzed WGS data coming from a cohort of 17 patients from whom two concurrent samples were collected: the CLL phase and the transformed phase (RS). This cohort includes samples and data generation as described in Klintman et al.[23].

Only samples with a lymphocyte count of greater than $25 \times 10^9 \, l^{-1}$ were included in the study ensuring a tumor purity greater than 80% and a median lymphocyte count of $80 \times 10^9 \, l^{-1}$ (range, 33.9–166.5) (Supplementary Table 1).

Peripheral blood mononuclear cells (PBMCs) and a saliva sample were collected from each patient, which served as a source of tumor and germline DNA, respectively. DNA was extracted from PBMCs and saliva using QIAamp DNA mini kit (Qiagen) and the Oragene DNA saliva kit (DNA Genotek Inc) kits, respectively, according to the manufacturer's instructions. DNA quality was assessed using Nanodrop (Thermo Fisher Scientific) and quantified using Qubit (Thermo Fisher Scientific) technology. RNA was extracted from PBMCs using the RNeasy Mini Kit (Qiagen) according to the manufacturer's instructions. The quality of RNA was assessed using the Agilent 4200 Tapestation System, using High Sensitivity tapes. The concentration was assessed using the GeminiTM XPS Microplate Spectrofluorometer from Molecular Devices and the Quant-iT HS RNA assay.

### Whole-genome sequencing

Whole-genome 125 bp paired-end TruSeq PCR–free libraries were sequenced using Illumina HiSeq2500 technology. Raw sequencing data was aligned with using Isaac v.03.16.02.19 to GRCh38. Alignment and coverage metrics were calculated using Picard v.2.12.1 and Bwtool[47] showing a mean read depth of 36× and 109× for normal and tumor samples, respectively. All downstream analysis of WGS data was performed on the whole dataset of 485 samples, unless otherwise stated.

### RNA-seq

Libraries were prepared from samples of 74 patients using the Illumina Stranded Total RNA Prep, Ligation with Ribo-Zero Plus, with additional custom depletion probes, using 100 ng RNA. Libraries were sequenced on a NovaSeq 6000 system (Illumina) using 100 base paired-end chemistry (108–455 million read-pairs per sample). Sequencing reads were processed and aligned to Human Reference genome GRCh38 using the Illumina Dragen RNA pipeline v.3.8.4. Gentoyping was performed using bcftools mpileup[48]. Allele specific read counts were generated at sites of acquired SNVs determined by WGS.

### ATAC-seq

ATAC-seq was performed as previously described[49]. Briefly $7.5 \times 10^4$ cells per technical replicate were resuspended in lysis buffer (10 mM Tris-HCl, pH 7.5, 10 mM NaCl, 3 mM MgCl$_2$, 0.1% IGEPAL CA-630). Nuclei were pelleted (500g for 10 min), PBS was discarded and nuclei were resuspended in tagmentation buffer (25 µl 2× tagmentation DNA buffer, 2.5 µl Tn5 Transposase (Illumina) and 22.5 µl water) then incubated (37 °C for 30 min). DNA was extracted using the MinElute PCR Purification Kit (Qiagen), half the DNA was amplified (NEBNext High-Fidelity 2× PCR Master Mix (New England Biolabs)) and purified with the QIAquick PCR Purification Kit (Qiagen). Libraries were sequenced using 40-bp paired-end reads (Illumina NextSeq).

Reads were mapped to GRCh38 using the PEPATAC pipeline with prealignment to the mitochondrial genome and default settings[50]. Gentoyping was performed using bcftools mpileup[48]. Allele specific ATAC-seq read counts were generated at sites of acquired SNVs determined by WGS.

### Immunoglobulin gene characterization

To determine the IGHV status of our cohort, we prioritized data from Sanger sequencing, followed by WGS-derived data including IgCaller[51] results and the presence of noncanonical AID mutational signature (SBS9). This prioritizing scheme resulted in 54% (264/485) cases classified by Sanger sequencing, 40% (194/485) by the IgCaller algorithm and 6% (27/485) by the mutational signature SBS9. The correlation between these three methodologies was high, as can be seen in Supplementary Table 26. In addition, the IgCaller algorithm was used to further characterize the IG genes, including to define the IGHV3-21 rearrangement in 10% (47/485) of cases and CLL stereotypy in 27% (132/485). To assign CLL stereotypes, the IgCaller output was used as input for AssignSubsets online tool[52], which annotates the 19 main subsets, including subsets 1, 2, 4 and 8, as recommended by ERIC guidelines[39]. In cases more than one rearrangement were detected, we selected the rearrangement with the highest score to define the main CLL stereotype. In cases where a rearrangement was not assigned, but there was a proximal rearrangement reported, we included this rearrangement in our analysis.

### Somatic variant calling and filtering

SNVs and indels were called using Strelka v.2.8.4 7 adopting default parameters. Filtering of SNVs/indels was performed as follow: depth required greater than ten and allele fraction (AF) greater than 0.05; the quality filter annotation should be 'PASS' and quality score greater than 30; variants with allele frequency less than 0.05 from 1KGP phase 3 1405.34_GRCh38.p8 and EXAC v.0.3 data (annotated from using Ensembl VEP GRCh38 release v.89.4 (ref. [53])). Additional filters according to the Illumina v.4 Genomics England annotation pipeline removed variants as follows: variants with a population germline frequency greater than 1% in either the Genomics England dataset or in the gnomAD v.3; recurrent somatic variants with a frequency greater than 5% in the Genomics England cohort; variants overlapping with LINE repeats or simple repeats found with Tandem Repeats Finder v.4.09 (ref. [54]); calls within 50 bp either side of an indel where at least 10% of variants have been filtered due to quality; locus depth is greater than three times mean chromosomal depth in the germline sample; contains multiple alternate alleles; germline sample is not the homozygous reference or indel Q-score is less than 30; variant quality score recalibration (VQSR) score less than 2.75; most overlapping reads do not map uniquely to

variant position; within ten bases of Genomics England inhouse database or Gnomad v.3 germline indel with frequency greater than 1%; SNVs resulting from systematic mapping and calling artefacts; fails somatic panel of normal Phred cut-off (< 80).

The Supplementary Notes include details on cancer cell fraction calculation as well as coding and noncoding variant annotations. In addition, it includes our approach for assigning target genes of regulatory elements, identifying of coding and noncoding candidate drivers.

### Structural variant identification

The structural variant (SV) calling pipeline for detection of inversions and translocations was as follows. (1) Delly[55] was used to call variants in each tumor–germline pair, with the following steps: complete somatic prefiltering, genotype all potentially somatic sites across all CLL germline samples, postfilter for somatic SVs using control samples. Variants with an alternative AF less than 0.05 were removed. (2) Lumpy v.0.2.13 (ref. [56]) and (3) Manta 0.28.0 were also used to call SVs. Variants with an alternative AF < 0.05 or for which there was any evidence in the germline were removed for consistency. (4) The pcawg-merge-sv consensus calling pipeline[57] was adapted for this analysis. SVs supported by two or more callers were reported.

### Identification of CNAs

We used both DNA microarray (n = 109 samples) and WGS (n = 485 samples) to determine CNAs and observed high concordance between the two methods. Of 282 CNAs detected by WGS, 240 (85%) were also reported by DNA microarray with high confidence (Supplementary Table 6). In addition, we further reduced false positive signals using a combination of intersects between several variant callers and visual inspection as detailed below.

Samples from subset of 109 patients enrolled in ARCTIC and AdMIRe trials were genotyped using HumanOmni2.5-8 BeadChip arrays (Illumina Inc.). Genotypes were called using GenomeStudio v.2009.2 (Illumina Inc.). CN gains and losses greater than 50 kb and cnLOH less than 5 Mb were reported using Nexus Copy Number v.10 (BioDiscovery, Inc.), as previously described[16,58], with the following settings (SNPRank Segmentation): significance threshold, $1 \times 10^{-5}$; max contiguous probe spacing (kb), 1000.0; minimum number of probes per segment, 5; high gain, 0.6; gain, 0.2; loss, −0.2; big loss, −1.0; 3:1 sex chromosome gain, 1.2; homozygous frequency threshold, 0.95; homozygous value threshold, 0.8; heterozygous imbalance threshold, 0.4; minimum LOH length (kb), 20; percentage outliers to remove, 3%. We also inspected all genomes to scan visually for changes not identified using these analysis settings using Nexus visualization tool.

In the case of WGS, Canvas v.1.3.1 (ref. [59]) and Manta v.0.28.0 were used to call CNAs, filtering out centromeric and telomeric regions as defined in the UCSC cytoband table. Variants reported by Canvas with a quality score less than ten were filtered out. Variants reported by Manta were filtered out as follows: (1) variants with a normal sample depth near one or both variant break-ends three times higher than the chromosomal mean, and (2) variants with somatic quality score of less than 30.

For each remaining CNA, its presence and type (gain or loss) were confirmed by visually inspecting the genome-wide mean coverage and B-allele frequency data, derived from the aligned reads in 100 kb windows. Calls with continuous copy number changes of length greater than 100 kb were kept. The Supplementary Notes include details on cancer cell fraction calculation.

### Counts of number of drivers

We calculated the total number of drivers in each patient by the following methodologies: we established (1) the total mutational burden by counting the number of functional variants (that is, with the following exonic consequences splice acceptor variant, splice donor variant, stop gained, frameshift variant, stop lost, start lost, transcript amplification, in-frame insertion, in-frame deletion, missense variant, protein-altering variant or incomplete terminal codon variant), (2) the number of mutated coding drivers (out of 58) SNVs/ indels and (3) the number of mutated coding (SNVs/indels and CNAs) and noncoding drivers.

### Pathway analysis

Two pathway datasets were used: PANCANCER containing 14 pathways from The NanoString PanCancer Pathways Panel and KEGG containing 23 signaling pathways[60]. For the six pathways in common between the two lists, the PANCANCER pathway was selected, resulting in 31 unique pathways included in the analysis. We counted the number of patients with mutations per pathway considering (1) a gene panel of the coding drivers (n = 58); (2) the exome (coding drivers plus exonic mutation with high impact according to VEP annotations: splice_acceptor_variant, splice_donor_variant, stop_gained, frameshift_variant, stop_lost, start_lost); (3) a larger driver panel containing both coding drivers and regulatory candidate drivers (n = 58 + 126) and (4) all of the above combined (coding and noncoding drivers plus exonic mutation with high impact according to VEP annotations).

### Telomere analysis

Telomere analysis was carried out on all 485 CLL tumor-normal pairs. Telomere content was estimated using Telomere Hunter v.1.1.0 (ref. [61]). Telomere content is normalized by the total number of overall reads that comprise a 'telomere-like' GC-content range (48–52%). Telomere length in basepairs was estimated using Telomerecat v.1.0 (ref. [62]). We found that telomere content assessed using Telomere Hunter and telomere length assessed using Telomerecat were highly correlated (P = 0.84, P < 2.2 × 10^{-16}, Extended Data Fig. 8a). We compared the telomere lengths and contents between CLL samples and matched saliva samples as germline[63], considering that different cell types can naturally present different telomere lengths[64].

### Chromothripsis analysis

Chromothripsis was identified using Shatterseek[65], which aims to detect candidate regions on the basis of oscillating copy number states (using CNAs as previously described), as well as intersection with clusters of interleaved structural variants (SVs; that is, deletions, duplications, inversions and translocations) identified from the SV consensus pipeline previously described. Potential regions of chromothripsis were classified as 'high confidence' or 'low confidence' using criteria as per Cortés-Ciriano et al.[65].

### Mutational signatures

Extraction of SBS, DBS and small ID signatures was performed using SigProfilerExtractor v.1.0.1810 (ref. [66]). SigProfilerExtractor de novo signature extraction and decomposition were carried out according to default parameters, with potential de novo extracted signature solutions tested between 1 and 25 signatures. Signatures were referenced to the Catalogue of Somatic Mutations in Cancer (COSMIC) v.3; SigProfilerExtractor signatures were decomposed based on a cosine similarity greater than 0.9. Following decomposition to COSMIC signatures, SigProfilerExtractor estimated the overall signature contributions per tumor, as well as the per tumor signature estimates for each mutation context. Through associating these context estimates back to the original mutations, signature estimates were attributed to individual driver mutations, as well as genomic regions (exome, promoters, UTRs, and so on).

### GC analysis

We investigated the presence of GC using an unsupervised multiple correspondence analysis with FactoMineR[67]. We included 17 genomic measures as binary data, including variables binned as less than median or greater than or equal to median: number of SNVs, number of indels,

telomere lengths, telomere content and variables binned as presence/absence: SV breakpoint, CNA, CN gain, CN loss, cnLOH, trisomy, aneuploidy, CN gain excluding trisomy, CN loss excluding aneuploidy, cnLOH excluding whole chromosome cnLOH, inversion, translocation and chromothripsis.

### Genomic alterations in known risk factors and disease states

All genomic alterations derived from WGS were combined and included as follows: noncoding candidate drivers mutated in more than 5% of samples; coding drivers were combined according to the presence of an SNVs/indels and CNAs (union); recurrent CNAs that significantly co-occurred (mean square contingency coefficient, mu > 0.3) and defined in the same chromosome were combined (union). In addition, only genomic alterations with at least five occurrences across all the samples were included in the analysis. In total, 186 genomic remained including 58 coding drivers, 36 recurrent CNAs, 44 noncoding drivers, 12 pathways affected by genetic alterations, 28 global genomic features and mutational signatures, and eight genomic complexity groups (Supplementary Table 21).

We tested for enrichment (two-sided Fisher's exact test, FDR ≤ 0.05) of each genomic alteration in several known risk factor and disease state groups for samples with available data: age (195 samples < median age versus 216 samples ≥ median age); sex (338 male versus 136 female); disease stage (443 frontline versus 30 R/R); *TP53* status (420 WT versus 65 disrupted); IGHV mutational status (197 hypermutated versus 288 unmutated); minimal residual disease (MRD; 59 negative versus 57 positive); BCR IG subset 2 (33 presenting 2 versus 450 others); IGHV3-21 rearrangement (47 with versus 436 without).

### Relationship between genomic alterations and patient outcome

We examined the relationship between each of the 186 genomic features as detailed above (Supplementary Table 21) and patient outcomes using Cox proportional hazards models on 243 patients for PFS and 245 patients for OS. FDR-corrected *P* values were reported as significant if less than 0.05. In addition, several particular comparisons with more than two groups were performed using Kaplan–Meier curves and the log-rank test. These were: number of mutated drivers, the eight genomic complexity groups and the combination of different structural rearrangements. We also performed a multivariate analysis using penalized Cox regression, as implemented in the R package glmnet[68], to find a minimal set of predictors with maximal predictive power. An optimal value of the penalization parameter λ was selected using leave-one-out cross-validation; specifically, the value of λ that minimizes the cross-validation error.

### Patient stratification using non-negative matrix factorization

All 186 genomic features, as well as IGHV status including percent homology to germline (labeled MS), age and sex were selected for unsupervised clustering using non-negative matrix factorization[69,70] using the NMF v.0.22.0 R package[71] with the offset method[72,73]. Data were converted to a binary matrix using either presence or absence of a feature, or above or below the mean to avoid a mixture of binary and nonbinary data (Supplementary Note). After removal of samples without age information, samples were divided into m-IGHV (*n* = 168) and u-IGHV (*n* = 243) as defined above. The number of permitted NMF clusters in either the m- or u-IGHV subset was determined using a combination of rank estimation methods including the cophenetic correlation coefficient[74–76]. Data were randomized and the ranks estimated for comparison to avoid overfitting. NMF was carried out on each IG subset of samples separately to produce GSs.

DeconstructSigs v.1.9.0 (ref. [77]) was designed to use the mutation catalog of a sample to define the linear combination of COSMIC signatures that best reconstruct that sample's mutational profile. Here, we used this tool to define the linear combination of GSs calculated using the NMF method that best reconstruct the genomic features of a sample. The proportions of each GS within all patients were then clustered using mclust v.5.4.6 (ref. [78]) and assigned a cluster that maximized parsimony whilst still producing an adequate prediction. The defined GSs were then compared with known subgroups such as BCR IG subsets and patients harboring an IGHV3-21 rearrangement.

Testing of the method was carried out as follows:

(1) Data were randomly split into two trial groups each representing 50% of the dataset: and further divided into m-IGHV and u-IGHV CLL. The NMF was then performed on all genomic features on each group and evaluated using cosine similarity between group signature matrices (Supplementary Table 22);

(2) all samples used for NMF were split into 80% (m-IGHV: *n* = 133, u-IGHV: *n* = 195) training and 20% (m-IGHV: *n* = 34, u-IGHV: *n* = 49) testing at random. The NMF was performed on the training data as described above to produce GS matrixes (m-GS, u-GS). The training data were then assigned to a GS using deconstructSigs to identify the combination of GSs that best reconstructed a sample's genomic feature matrix and then assigning the signature that occurred at the highest percentage. The signature assigned to the test samples was then compared with the signatures assigned to those same individuals when 100% of data was used for both training and testing (Fig. 8g).

### Data wrangling and plotting

Plotting of data was performed using tidyverse v.1.3.0 (refs. [79,80]) in R v.3.6.2 (ref. [81]). Mutation hotspot graphics were plotted using the package GenVisR v.1.18.1 (ref. [82]). Lollipop plots were plotted with the MutationMapper from cbioportal accessible from https://www.cbioportal.org/mutation_mapper. Genomic views were prepared using the UCSC genome browser[83].

### Statistics and reproducibility

The sample size calculation was critical to the success of this program. Our power calculations considered the heterogeneity of CLL and a background somatic mutation frequency of 0.8 mutations per megabase. This means that, to reliably detect somatic mutations recurring in 2% of patients with CLL, we need to sequence approximately 500 CLL genomes (Supplementary Fig. 8). No data were excluded from the analyses. The experiments were not randomized. The investigators were not blinded to allocation during experiments and outcome assessment.

### Reporting summary

Further information on research design is available in the Nature Research Reporting Summary linked to this article.

## Data availability

The National Genomic Research Library (NGRL) is a 'reading library', therefore data cannot be extracted directly. All WGS data, BAM files and processed files cited can be viewed in situ via the Haematological Malignancy Genomics England Clinical Interpretation Partnership (GECIP), once an individual's data access has been approved. The link to becoming a member of GECIP to get access can be found here https://www.genomicsengland.co.uk/research/academic/join-gecip. The process involves an online application, verification by the applicant's institution, completion of a short information governance training course (circa 30 min), and verification of approval by the Haematological Malignancy domain lead (A.S., see contact details for corresponding author). Please see https://www.genomicsengland.co.uk/research/academic for more information.

All RNA sequencing data has been deposited in the European Bioinformatics Institute (EMBL-EBI) ArrayExpress Archive of Functional Genomics Data database under accession number E-MTAB-12124. The outcome of the clinical studies has been published (all references in Methods). Access to clinical datasets is subject to data sharing policies

of the respective clinical trial units that provided legal sponsorship for the studies and can be made available on request to A. Pettitt (arp@liverpool.ac.uk; Department of Molecular and Clinical Cancer Medicine, University of Liverpool, Liverpool, UK) and P. Hillmen (peter.hillmen@nhs.net; St. James's University Hospital, Leeds, UK). Source data are provided with this paper.

## Code availability

All open source tools used in this manuscript were cited in the Methods or Supplementary Notes. No custom code was used for any aspect of data processing or analysis.

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

## Acknowledgements

Patient material was obtained from the UK CLL Biobank, University of Liverpool, which is funded by Blood Cancer UK. This work was supported by the Genomics England Research Consortium and the CLL pilot consortium (full list of Individual consortia authors are listed in the Supplementary Material). This research was made possible through access to the data and findings generated by the 100,000 Genomes Project. The 100,000 Genomes Project is managed by Genomics England Limited (a wholly owned company of the Department of Health and Social Care). The 100,000 Genomes Project is funded by the National Institute for Health Research and NHS England. The Wellcome Trust, Cancer Research UK and the Medical Research Council also funded research infrastructure. The 100,000 Genomes Project uses data provided by patients and collected by the National Health Service as part of their care and support. This work was supported by the National Institute for Health Research Oxford Biomedical Research Centre (A.S., D.V. and K.R.). The views expressed in this publication are those of the authors and not necessarily those of the Department of Health. The work of P.R. was supported by the Japan Society for the Promotion of Science Postdoctoral standard program. The work of R.S.H. is supported Wellcome Trust (214388) and Cancer Research UK (C124388) grants. The work of A.R.P. and S.D. was supported by Blood Cancer UK. A.B. received D.Phil. funding from Health Education England and Genomics England. J.I.M.-S. is funded by the European Research Council under the European Union's Horizon 2020 research and innovation program (Project BCLLATLAS, grant agreement 810287). J.C.S. is funded by Cancer Research UK (ECRIN-M3 accelerator award C42023/A29370, Southampton Experimental Cancer Medicine Centre grant C24563/A15581, Cancer Research UK Southampton Centre grant C34999/A18087 and program C2750/A23669). The funders had no role in study design, data collection and analysis, decision to publish or preparation of the manuscript.

## Author contributions

P.R., K.R. and A.S. designed the experiments. N.A., M.O., A.R.P. and P.H. organized clinical trials, collected clinical data, performed sample collection and performed processing. H.D., A.C., M.C., P.R., D.J. and A.V.P. oversaw and performed DNA and RNA extraction, quality control and additional experiments. M.R. and D.B. organized library preparation and WGS and data transfer. N.D., J.H. and J.D. performed the ATAC-seq experiment and analyzed the data. T.J., U.M. and D.V.V. performed RNA quality control, library preparation and RNA-seq and data processing. P.A. generated alignment metrics and G.E.R.C. was responsible for initial data processing and informatic infrastructure. R.M. provided ChIP–seq and linkage data. M.D. and J.I.M.S. provided ATAC-seq and ChromHMM data and defined regulatory elements active in CLL. K.R., D. Chubb, A.J.C. and J.G. performed variant calling, filtering and annotation. P.R., K.R., B.K., D. Cavalieri and L.L.P. investigated somatic variants and drivers. P.R., R.C. and S.J.L.K. performed and analyzed the DNA microarray data. D.V.V. and P.R. performed pathways analysis. B.K., P.R. and D.V.V. performed genome-wide analysis. K.R. and B.K. performed mutational signature analysis. D.V.V., K.R. and P.R. performed and oversaw statistical analysis. N.A., A.B., B.S., R.A., R.C., J.C.S., P.C., S.D., R.S., J.I.M.S. and RSH contributed key experimental or analytical support. P.R., K.R., D.V.V. and A.S. interpreted the data. P.R., K.R. and A.S. wrote the manuscript. P.R., K.R., D.V.V., N.A., B.N., J.S., R.S.H., J.I.M.S. and A.S. edited the manuscript. M.J.C. provided oversight of the program.

## Competing interests

In the past five years, A.S. has received in-kind contributions from Illumina and Oxford Nanopore Technology and is a shareholder of Illumina. She is a company director and shareholder of SERENOx Ltd. A.S. has received honoraria from Exact Sciences, Janssen, Astra Zeneca, Abbvie and Beigene, non-restricted research grants from Janssen and Astra Zeneca and an educational grant from Abbvie. A.R.P. receives research funding from Celgene/BMS, Gilead, Napp and Roche. N.A. received speaker fees from Gilead. P.A., T.J., U.M., M.R. and D.B. are employees of Illumina, a public company that develops and markets systems for genetic analysis. The remaining authors declare no competing interests.

## Additional information

**Extended data** is available for this paper at https://doi.org/10.1038/s41588-022-01211-y.

**Correspondence and requests for materials** should be addressed to Anna Schuh.

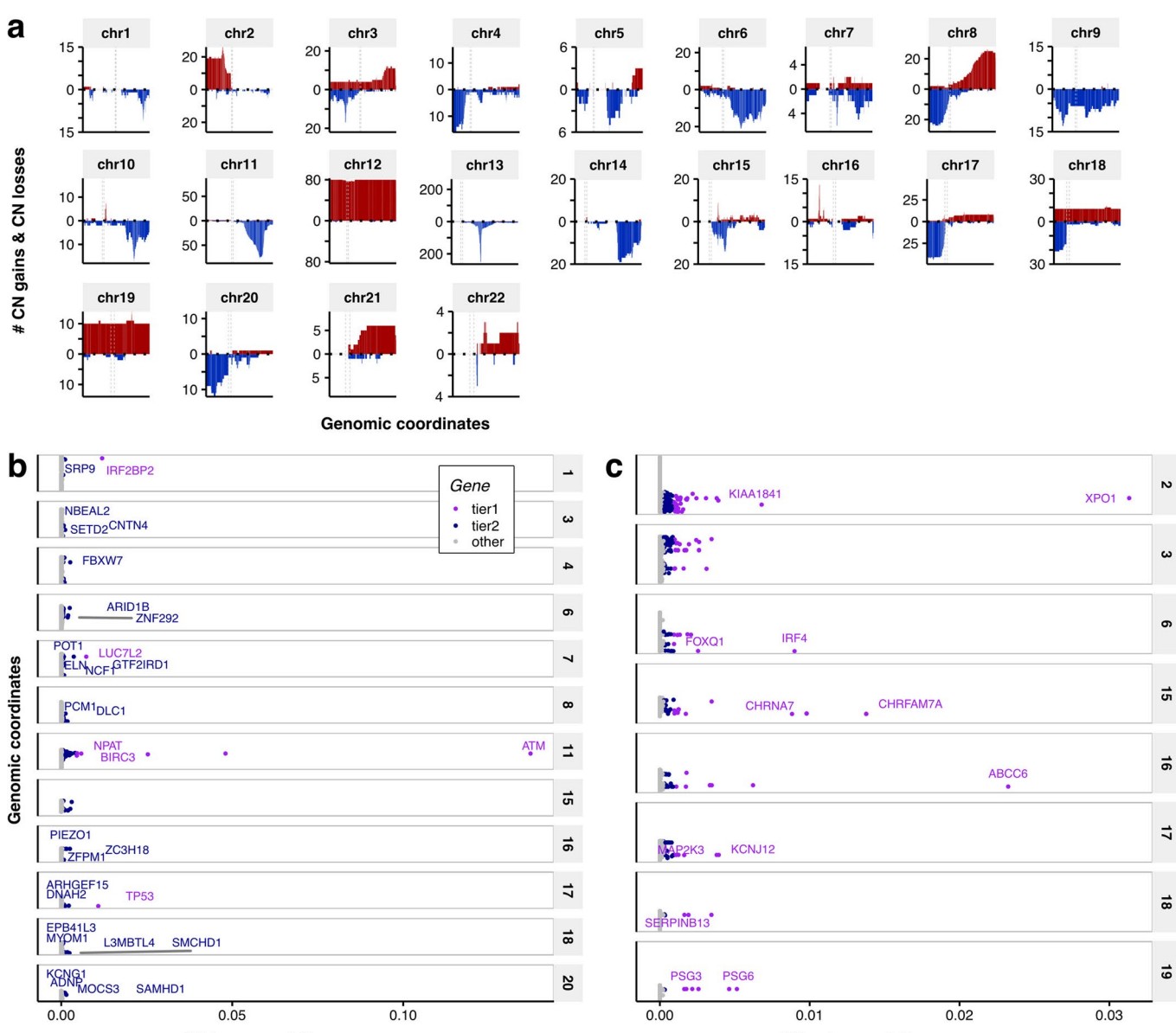

**Extended Data Fig. 1 | Recurrent CNAs contributing to the identification of candidate drivers. a**, Number of samples with copy number gains (upper track, red) and losses (lower track, blue) (y axis) according to the genomic coordinates of the full chromosome from 5' to 3' (x axis), for each chromosome (panels). **b-c**, MutComFocal scores for genes affected by CN losses and mutations (b) and gene significantly affected by CN gains and mutations (c) Genes classified as tier 1 and 2 were selected for further investigations.

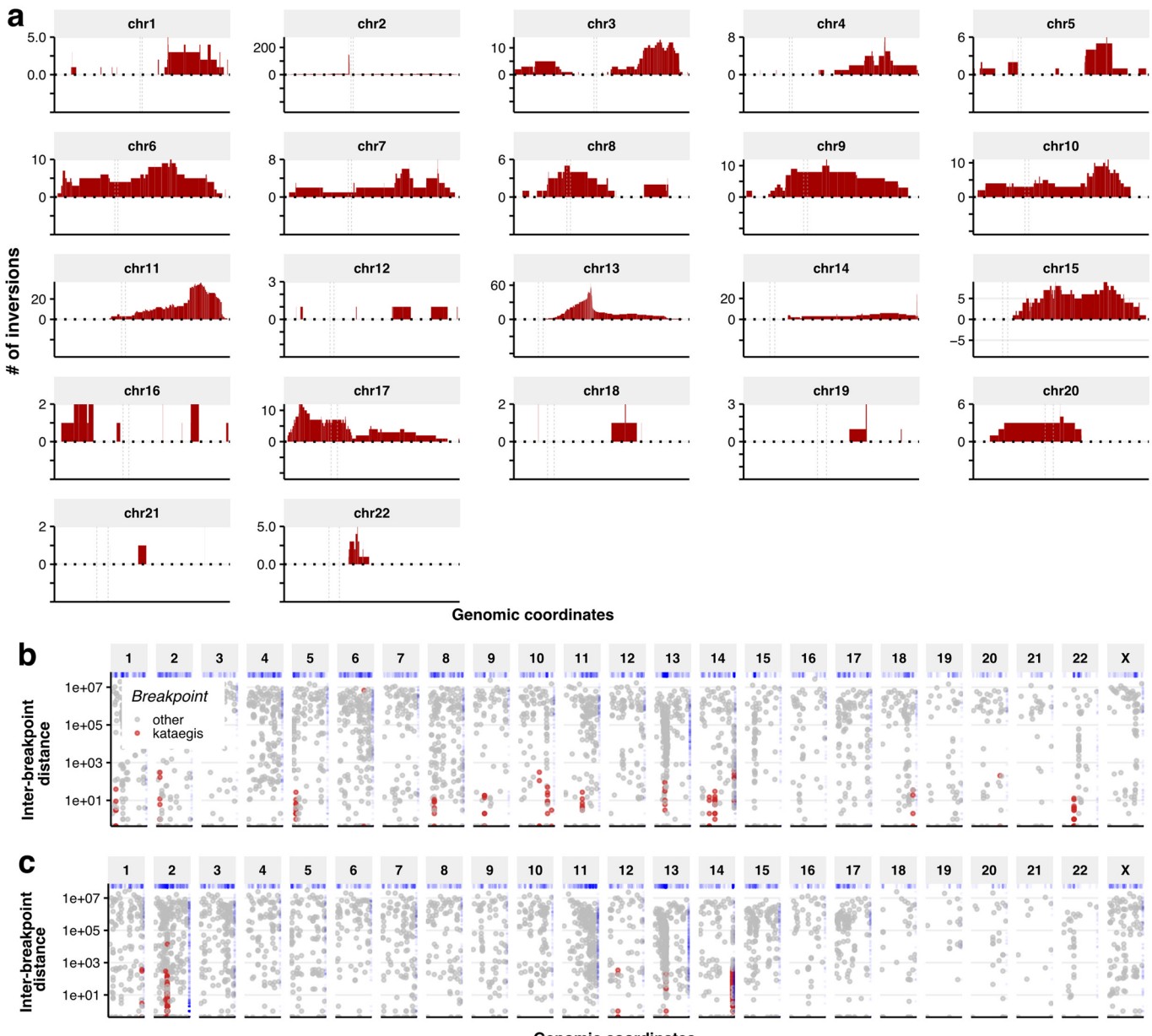

**Extended Data Fig. 2 | Recurrent inversions and translocations. a**, Number of samples with inversions (y axis) according to the genomic coordinates of the full chromosome from 5' to 3' on each chromosome (x axis), for each chromosome (panels). **b-c**, Distance between all inversion (b) and translocation (c) (breakpoints across all 485 samples on each chromosome highlighting hotspot breakpoints (named kataegis, in red).

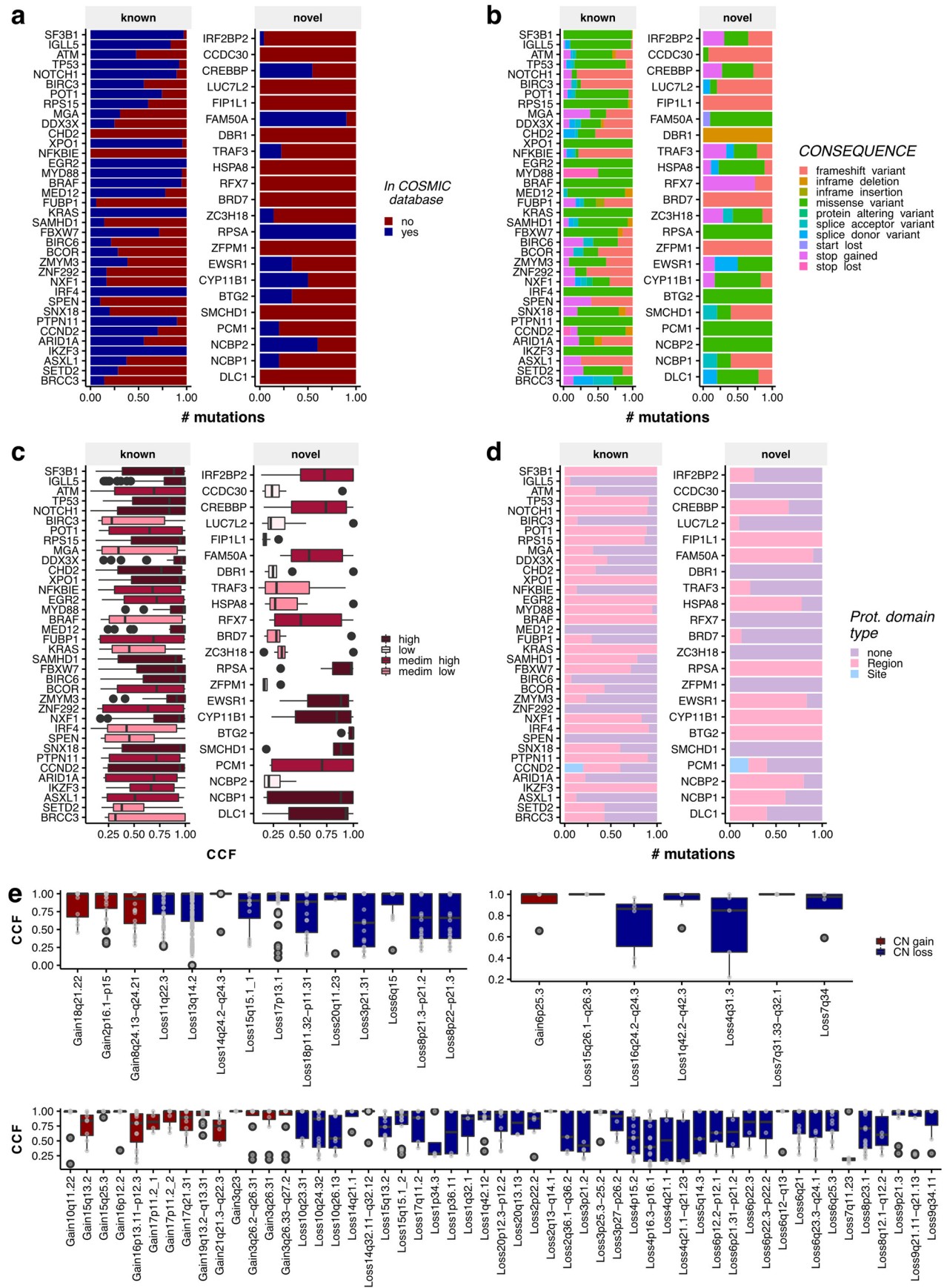

**Extended Data Fig. 3 | See next page for caption.**

**Extended Data Fig. 3 | Characterization of candidate genes and regions of CNAs. a**, Number of variants previously reported in the COSMIC database. **b**, Number of variants for each consequence. **c**, Distribution of cancer cell fractions binned in four groups [1-0.75],]0.75-0.5],]0.5-0.25],]0.25-]. The number of variants represented in each boxplot is detailed in Supplementary Table 12.

**d**, Number of variants occurring in protein domains including two types of protein domains: sites and regions, as defined in by Prot2HG[39]. **e**, Distribution of cancer cell fractions of recurrent CN gains (red) and CN losses (blue). All boxplots show the minimum and maximum values and interquartile range. The number of CNAs represented in each boxplot is detailed in Supplementary Table 6.

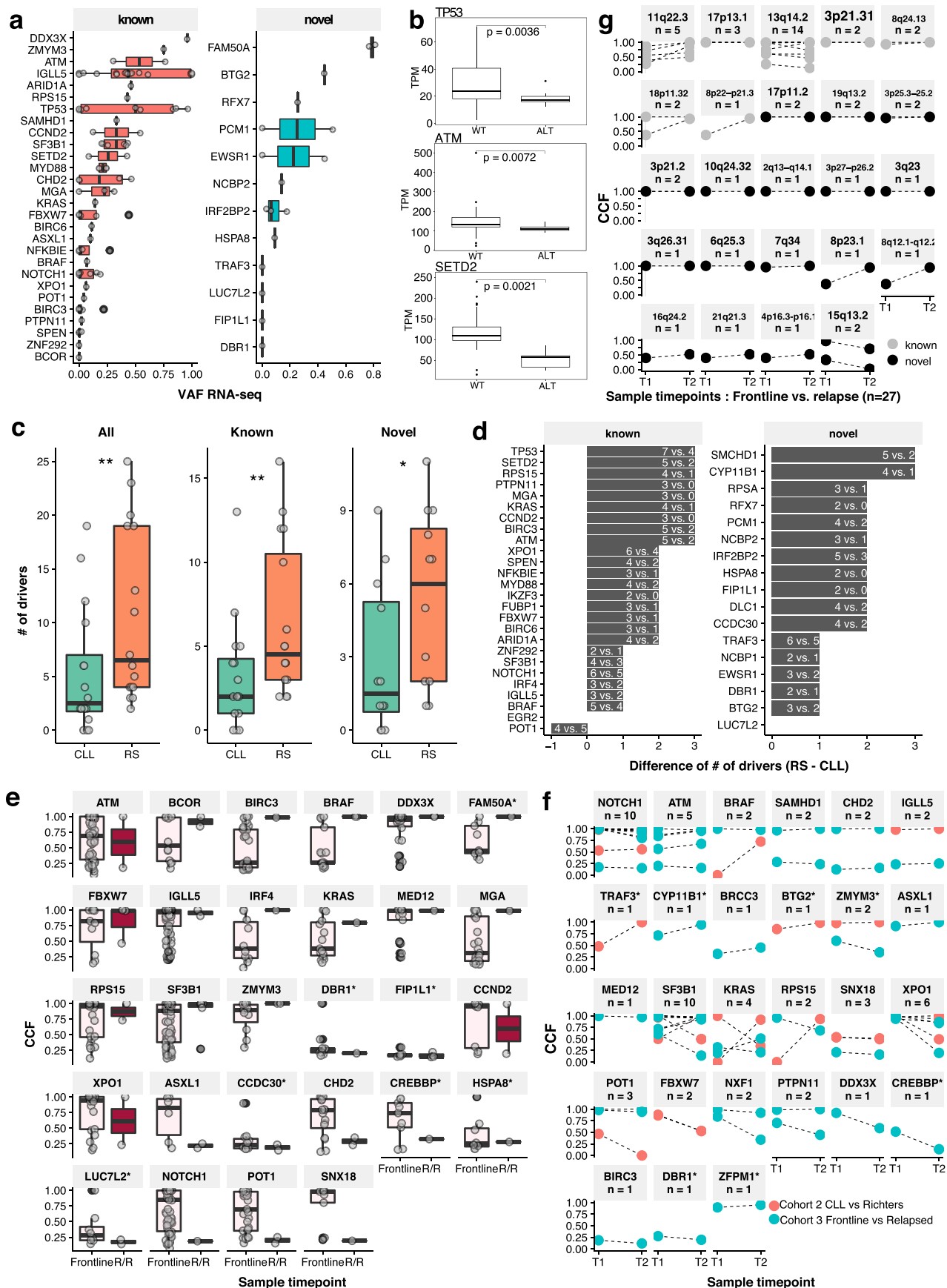

Extended Data Fig. 4 | See next page for caption.

**Extended Data Fig. 4 | Integration of RNA-seq and timeseries to investigate candidate drivers. a**, Distribution of variant allele fraction calculated from the RNA-seq data of variants detected by WGS. The number of variants represented in each boxplot is detailed in Supplementary Table 13. **b**, RNA-seq defined transcripts per million (TPM) of selected genes *TP53* (n = 7 mutated vs. 66 WT), *ATM* (n = 6 mutated vs. 67 WT), and *SETD2* (n = 7 mutated vs. 68 WT), according to the presence of a genomic alterations (ALT) or an absence of alteration (WT). p-values were calculated using a two-sided Welch's t-test. **c**, Number of driver genes per sample for CLL samples and paired Richter's syndrome (RS) samples (n = 16 vs. 16) for all 58 drivers (left panel) p = 2.4 ×10$^{-3}$, the 36 known genes (central panel) p = 5.8 ×10$^{-3}$ and 22 candidate genes (right panel) p = 2.1 ×10$^{-3}$ (two-sided Welch's t-test). **d**, Difference of number of mutated samples in Richter Syndrome (RS) samples vs paired CLL samples. Each bar indicates the absolute number of mutated samples per group. **e-f**, Distribution of cancer cell fractions (CCFs) in frontline samples vs. relapsed/refractory (R/R) samples (cohort 1, unpaired samples) (e), and CLL vs RS as well as frontline vs. relapsed (cohort 2 and 3, paired samples) (f). Corresponding variants are connected by a dotted line. Panels are ordered based on the direction of evolution: high stable CCFs and increasing CCFs, mixed increasing/decreasing CCFs, low/decreasing CCFs. Figures are not shown if no R/R / T2 sample carried a mutation. * indicates candidate drivers. **g**, Distribution of cancer cell fractions in frontline vs. relapsed (paired samples). Corresponding variants are connected by a dotted line. Panels are ordered based on the direction of evolution: high stable CCFs and increasing CCFs, mixed increasing/decreasing CCFs, low/decreasing CCFs. All boxplots show the minimum and maximum values and interquartile range and all datapoints are represented.

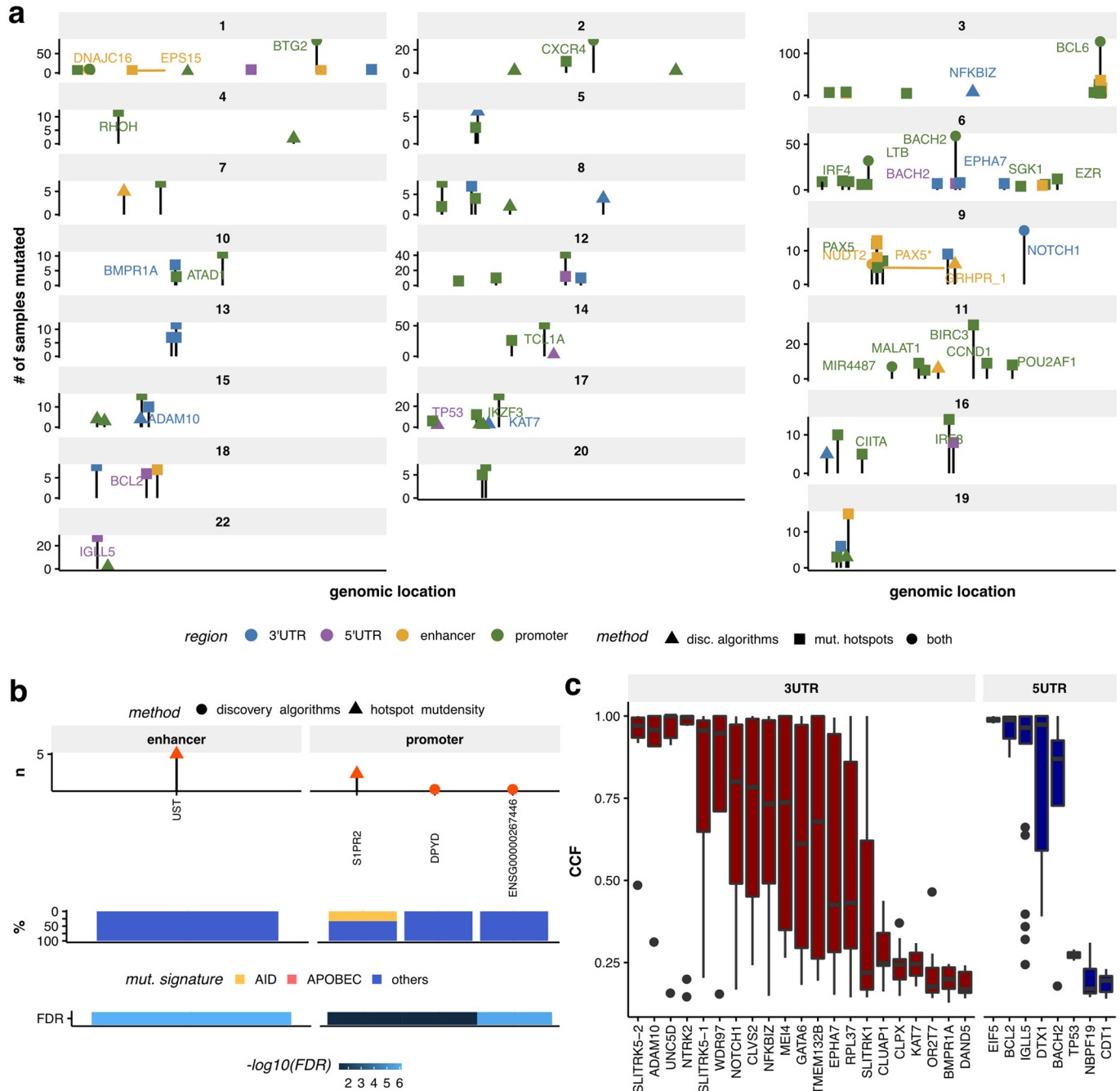

**Extended Data Fig. 5 | Annotations of non-coding candidate CLL drivers in regulatory elements. a**, Genomic map of non-coding drivers per chromosome (panels) according the genomic coordinates of the full chromosome from 5' to 3' (x axis). Methods of detection were discovery algorithms (disc. algorithms), mutational hotspot analysis (mut. hotspots). **b**, Number of mutations in non-coding genomic elements (top panel) and proportion of variants with signature attributed to AID, APOBEC or other processes (bottom panel) for regulatory elements exclusively active in samples with u-IGHV. **c**, Distribution (showing the minimum and maximum values and interquartile range) of CCFs in significantly mutated UTRs. The number of variants represented in each boxplot is detailed in Supplementary Table 17.

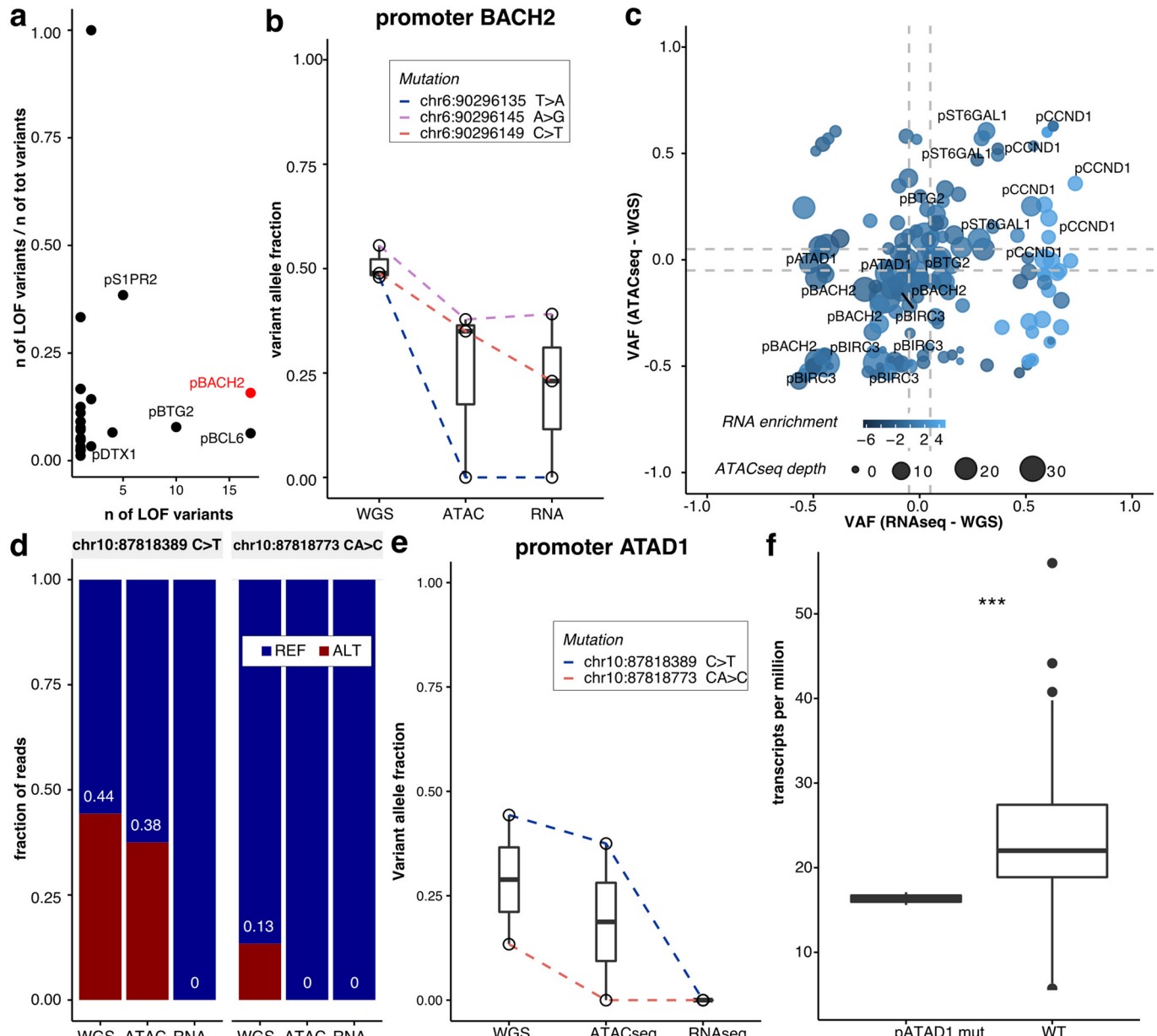

**Extended Data Fig. 6 | Investigation of significantly mutated promoters by ATAC-seq and RNA-seq. a**, Prioritization of significantly mutated promoters using DeepHaem prediction results. *BACH2* showed the highest number of loss-of-function variant compared to the total number of variants in its promoter. **b**, allelic skew of mutations in the promoter *BACH2* (n = 3). **c**, Distribution of allelic skew in ATAC-seq and RNA-seq data compared to WGS data in all variants located in significantly mutated promoters. The RNA enrichment was calculated by the difference of expression in the mutated sample with the mean expression across the cohort. Variants in promoter of interest were located in the top right corner

(increase of allelic fraction of mutant in ATAC-seq and RNA-seq) and bottom left corner (decrease of allelic fraction of mutant in ATAC-seq and RNA-seq). **d**, Fraction of mutant and WT reads in one *ATAD1* promoter variants showing allelic skew in ATAC-seq and RNA-seq compared to WGS. **e**, Allelic skew of mutations in the promoter *ATAD1* (n = 2). All boxplots show the minimum and maximum values and interquartile range. **f**, Gene expression of *ATAD1* in transcript per million (TPM) determined by RNA-seq in samples with *BCL2* 5′UTR mutations vs. WT. Black dots are marks as outliers. P-value was derived from a two-sided t-test.

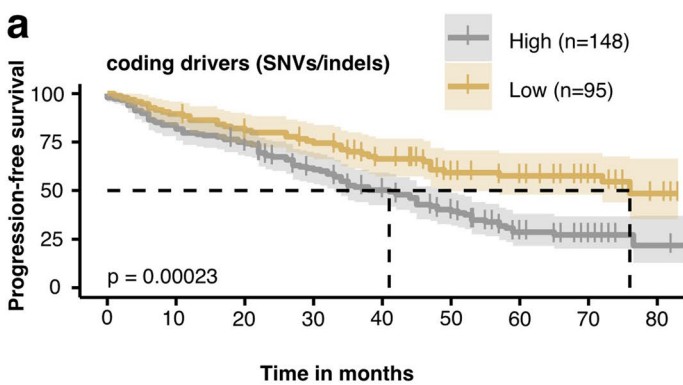

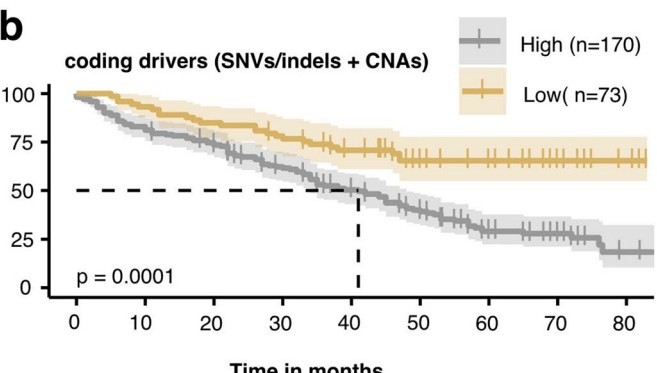

**Extended Data Fig. 7 | Mutational burden influenced patients' survival.**
**a-b**, Kaplan-Meier curves on number of drivers defined as (a) number of mutated coding drivers (SNVs/indels), and (b) number of mutated coding drivers (SNVs/indels + CNAs). High denotes > median and low denotes < = median. P-values were derived from a log-rank test. Shading denotes confidence intervals and dotted lines median survival for each group.

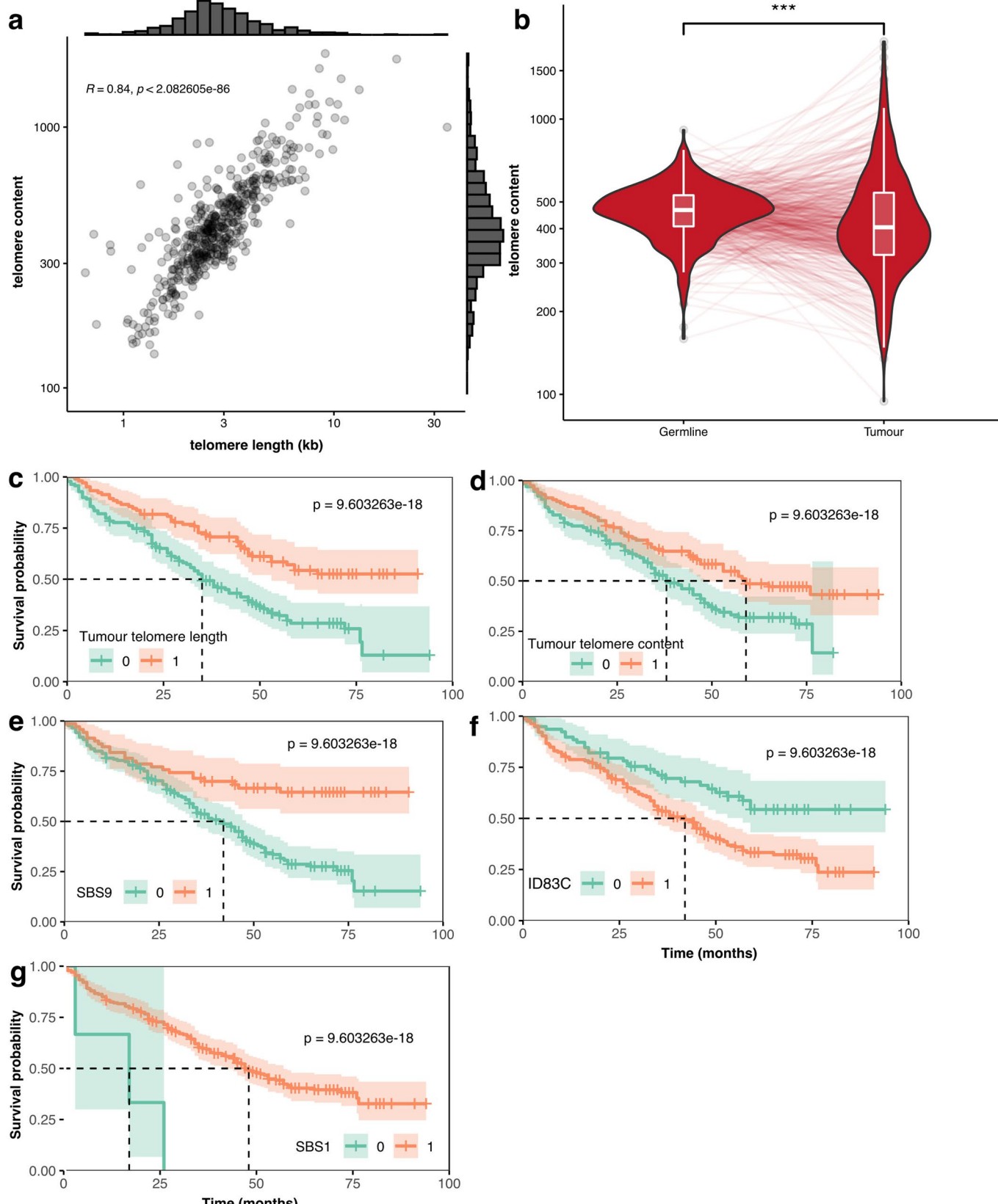

**Extended Data Fig. 8 | Telomere and mutational signatures associated with adverse outcome. a**, Pearson correlation between telomere content (assessed by Telomere Hunter v1.1.0[41]) and telomere length (assessed by Telomerecat v1.0[42]). **b**, Comparison of telomere content distribution (showing the minimum and maximum values and interquartile range) (Telomere Hunter) between normal samples and CLL samples. Lines link match tumor-normal datapoints.

Significance was showing two-sided paired Wilcoxon test of p-value <0.001 (n = 485 tumor samples vs. 485 germline samples). **c-g**, Kaplan-Meier curves of genomic features with lowest false discovery rate (FDR) when tested against progression-free survival (PFS) using a Cox proportional-hazards model (univariate analysis). P-values were derived from a log-rank test. Shading denotes 95% confidence intervals (additional data in Supplementary Table 15).

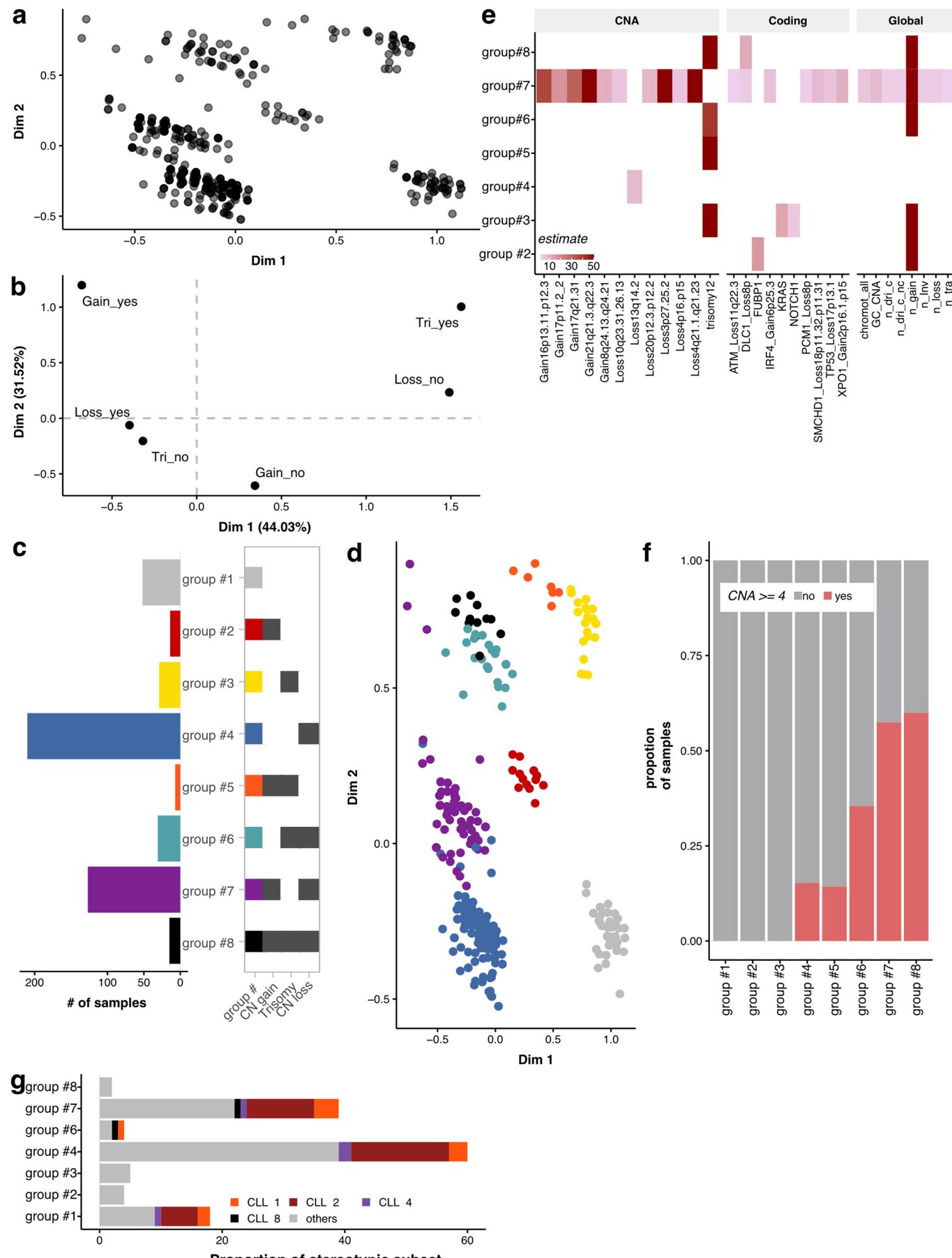

**Extended Data Fig. 9 | See next page for caption.**

**Extended Data Fig. 9 | Genomic complexity sub-groups. a**, Multiple correspondence analysis (MCA) plot showing the top two dimensions: dimension 1 (44% of variance) and dimension 2 (21% of variance). Each datapoint represents one sample. **b**, MCA variable representation according to the top two dimensions. Variables are represented as binary: "no" is "absence" and "yes" is "presence". **c**, Number of samples and description of the 8 groups defining genomic complexity (GC): each group is presented with a different color (matching other figures). Dark grey squares denote the presence of the alteration and white squares denote the absence of alteration. **d**, MCA plot showing the 8 GC groups. Colours keys are provided in c. **e**, Enrichment of genomic features for each group according to the Fisher's exact test for count data (FDR < 0.05 and estimate > 1), CNA: recurrent copy number alterations, coding: coding drivers, Global: genome-wide global lesions. **f**, Proportion of samples reported with GC (the conventional definition of > = 4 CNAs) in each GC group. **g**, Enrichment of GC groups for stereotyped subsets. No significant results were found using a two-sided Fisher's Exact test. Group #5 was not tested as sample size was lower than threshold.

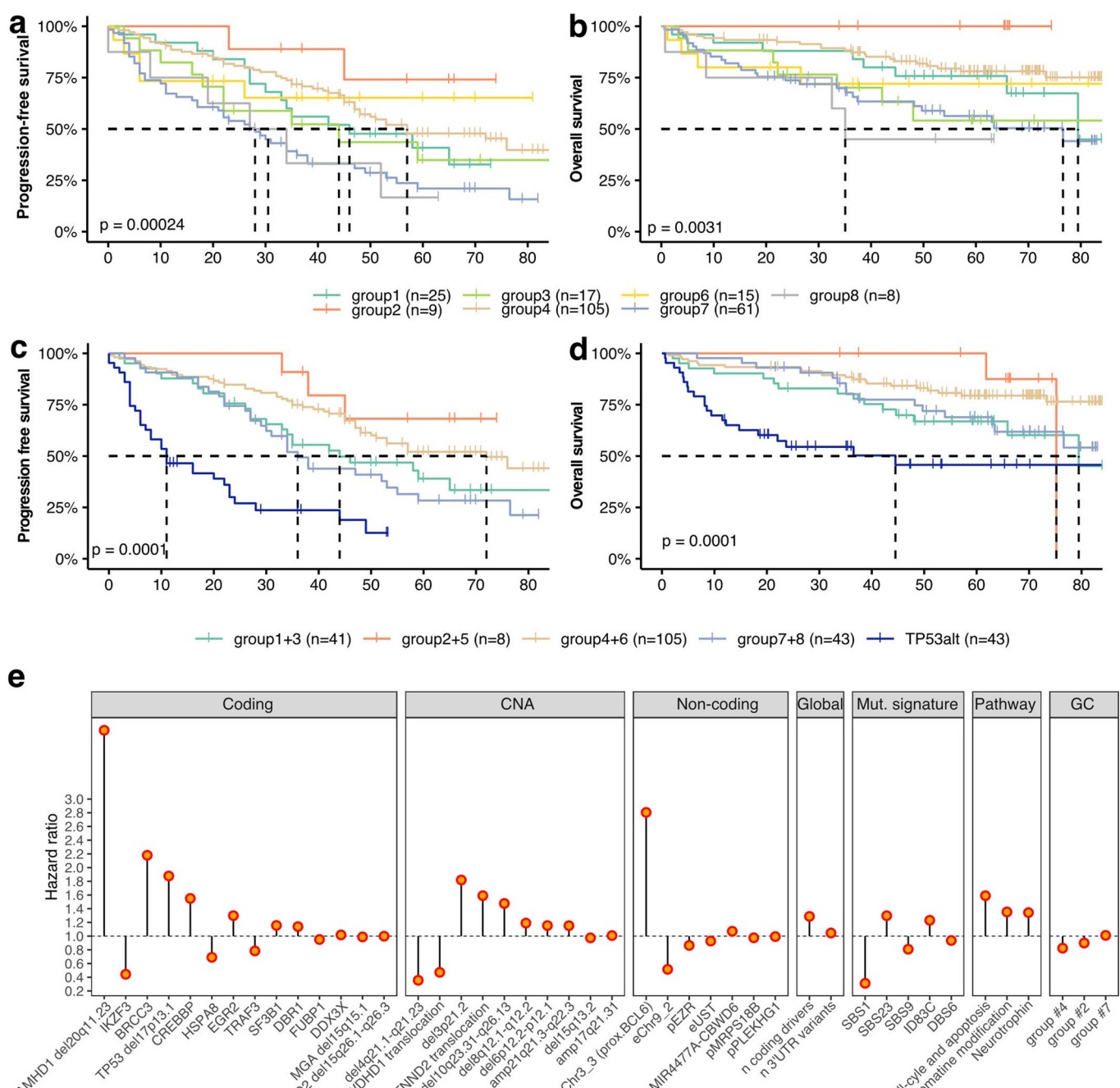

**Extended Data Fig. 10 | Genomic complexity and survival. a-b,** Kaplan-Meier curve on (a) progression-free survival (PFS) and (b) overall survival (OS) of all 8 genomic complexity (GC) groups. **c-d,** Kaplan-Meier curve on (c) PFS and (d) OS of all 8 GC groups independently from *TP53* alterations. Group 5 is not presented due to small sample size. Groups were further combined to increase power, by focusing on presence / absence of CN gains and losses and omitting trisomy data.

P-values were derived from a log-rank test. Shading denotes confidence intervals. **e,** Penalised Cox Regression performed on 186 genomic features identified independent predictors for progression-free survival. Each panel shows the hazard ratios for those genomic features that jointly minimise the out-of-sample prediction error (estimated through leave-one-out cross-validation). The full list and details of each genomic feature is presented in Supplementary Table 21.

# Reporting Summary

## Statistics

For all statistical analyses, confirm that the following items are present in the figure legend, table legend, main text, or Methods section.

| n/a | Confirmed | |
|---|---|---|
| ☐ | ☒ | The exact sample size (*n*) for each experimental group/condition, given as a discrete number and unit of measurement |
| ☐ | ☒ | A statement on whether measurements were taken from distinct samples or whether the same sample was measured repeatedly |
| ☐ | ☒ | The statistical test(s) used AND whether they are one- or two-sided<br>*Only common tests should be described solely by name; describe more complex techniques in the Methods section.* |
| ☐ | ☒ | A description of all covariates tested |
| ☐ | ☒ | A description of any assumptions or corrections, such as tests of normality and adjustment for multiple comparisons |
| ☐ | ☒ | A full description of the statistical parameters including central tendency (e.g. means) or other basic estimates (e.g. regression coefficient) AND variation (e.g. standard deviation) or associated estimates of uncertainty (e.g. confidence intervals) |
| ☐ | ☒ | For null hypothesis testing, the test statistic (e.g. *F*, *t*, *r*) with confidence intervals, effect sizes, degrees of freedom and *P* value noted<br>*Give P values as exact values whenever suitable.* |
| ☒ | ☐ | For Bayesian analysis, information on the choice of priors and Markov chain Monte Carlo settings |
| ☐ | ☒ | For hierarchical and complex designs, identification of the appropriate level for tests and full reporting of outcomes |
| ☐ | ☒ | Estimates of effect sizes (e.g. Cohen's *d*, Pearson's *r*), indicating how they were calculated |

*Our web collection on statistics for biologists contains articles on many of the points above.*

## Software and code

Policy information about availability of computer code

| Data collection | no |
|---|---|
| Data analysis | BioDiscovery Nexus v10<br>Isaac v03.16.02.19<br>Picard v2.12.1<br>Bwtool1.0<br>Strelka v2.8.4 7<br>Delly1<br>Lumpy 0.2.13<br>Manta 0.28.0<br>Canvas 1.3.1<br>MutSigCV v1.41<br>OncodriveFML v2.2.0<br>OncodriveCLUSTL v1.1.1<br>dNdScv1<br>CovGen v1.0.2<br>SNPeff v4.3<br>survcomp 1.40.0<br>MutComFocal v1.0<br>gprofiler2 v0.2.0<br>Telomere Hunter v1.1.0<br>Telomerecat v1.0<br>Shatterseek v1 |

deconstructSigs v1.9.0
SigProfilerExtractor v1.0.1810
FactoMineR2.4
glmnet4.1
NMF v0.22.0 R package
mclust V5.4.6, V5.4.9
tidyverse v1.3.0
GenVisR version 1.18.1
DeepHaem v1
IgCaller v1.2.1
AssignSubsets online tool 07.01.22
GenomeStudioV2009.2
Tandem Repeats Finder v4.09
Ensembl VEP GRCh38 release 89.4
MutationMapper from cbioportal accessible from https://www.cbioportal.org/mutation_mapper

For manuscripts utilizing custom algorithms or software that are central to the research but not yet described in published literature, software must be made available to editors and reviewers. We strongly encourage code deposition in a community repository (e.g. GitHub). See the Nature Portfolio guidelines for submitting code & software for further information.

## Data

Policy information about availability of data

All manuscripts must include a data availability statement. This statement should provide the following information, where applicable:

- Accession codes, unique identifiers, or web links for publicly available datasets
- A description of any restrictions on data availability
- For clinical datasets or third party data, please ensure that the statement adheres to our policy

All data and analytical tools are available within the Genomics England secure 'Research Environment'. Access is subject to institutional Participation Agreement as per Genomics England guidelines https://www.genomicsengland.co.uk/about-gecip/for-gecip-members/data-and-data-access/.

All DNA sequencing data generated within the study is deposited in the National Genomics Research Library. Data from the National Genomics Research Library, which are held in a secure Research Environment, are available freely to registered users. Please, see https://www.genomicsengland.co.uk/research/academic for more information.
All RNA sequencing data has been deposited in EGA and is available following EGA's data access policy.
The outcome of the clinical studies has been published (See all references in Methods). Access to clinical datasets is subject to data sharing policies of the respective clinical trial units that provided legal sponsorship for the studies and can be made available on request.

Publicly available datasets:
 gnomAD v3: https://gnomad.broadinstitute.org/news/2019-10-gnomad-v3-0/
UCSC cytoband table and GeneHancer: downloaded from the UCSC genome browser tables,  https://genome.ucsc.edu/cgi-bin/hgTables
GENCODE v29 : Downloaded on the gencode website, https://www.gencodegenes.org/human/release_29.html
All ChIP-seq data and chromatin state segmentation of primary CLL samples: EGA accession number EGAD00001004046.
3D chromatin interactions in GM128786: Under accession number GSE63525 in the Gene Expression Omnibus (GEO)
Prot2HG database freely available at www.prot2hg.com
PAN-CANCER pathway list: https://canopybiosciences.com/product/pancancer-pathways/#panel-details, and listed in supplemental Table 2 in the following publication:  https://doi.org/10.1182/blood.2020005650
3D chromatin interactions in one CLL sample : in EGA, under accession number EGAD00001004046.

# Field-specific reporting

Please select the one below that is the best fit for your research. If you are not sure, read the appropriate sections before making your selection.

☒ Life sciences          ☐ Behavioural & social sciences          ☐ Ecological, evolutionary & environmental sciences

For a reference copy of the document with all sections, see nature.com/documents/nr-reporting-summary-flat.pdf

# Life sciences study design

All studies must disclose on these points even when the disclosure is negative.

| | |
|---|---|
| Sample size | The sample size calculation was critical to the success of this programme. Our power calculations considered the heterogeneity of CLL and a background somatic mutation frequency of 0.8 mutations per Mb. This means that in order to reliably detect somatic mutations recurring in 2% of patients with CLL, we need to sequence approximately 500 CLL genomes.  No data were excluded from the analyses. The experiments were not randomized. The Investigators were not blinded to allocation during experiments and outcome assessment. |
| Data exclusions | No data was excluded from the genomics analysis. Analysis including clinical data only included a subset of patients for which clinical outcome data was available. |
| Replication | not applicable. Each sample was collected from a different patient which totalled in 485 biological replicates. In addition, technical replicates |

| Replication | were not needed as the sequencing noise and artefacts were controlled for using bioinformatic approaches. |
| Randomization | not applicable. This study is a non-interventional retrospective and descriptive analysis of the genomic landscape of CLL. Randomization is therefore not required. |
| Blinding | not applicable. This study is a non-interventional retrospective and descriptive analysis of the genomic landscape of CLL. Blinding is therefore not required. |

# Behavioural & social sciences study design

All studies must disclose on these points even when the disclosure is negative.

| Study description | Briefly describe the study type including whether data are quantitative, qualitative, or mixed-methods (e.g. qualitative cross-sectional, quantitative experimental, mixed-methods case study). |
| Research sample | State the research sample (e.g. Harvard university undergraduates, villagers in rural India) and provide relevant demographic information (e.g. age, sex) and indicate whether the sample is representative. Provide a rationale for the study sample chosen. For studies involving existing datasets, please describe the dataset and source. |
| Sampling strategy | Describe the sampling procedure (e.g. random, snowball, stratified, convenience). Describe the statistical methods that were used to predetermine sample size OR if no sample-size calculation was performed, describe how sample sizes were chosen and provide a rationale for why these sample sizes are sufficient. For qualitative data, please indicate whether data saturation was considered, and what criteria were used to decide that no further sampling was needed. |
| Data collection | Provide details about the data collection procedure, including the instruments or devices used to record the data (e.g. pen and paper, computer, eye tracker, video or audio equipment) whether anyone was present besides the participant(s) and the researcher, and whether the researcher was blind to experimental condition and/or the study hypothesis during data collection. |
| Timing | Indicate the start and stop dates of data collection. If there is a gap between collection periods, state the dates for each sample cohort. |
| Data exclusions | If no data were excluded from the analyses, state so OR if data were excluded, provide the exact number of exclusions and the rationale behind them, indicating whether exclusion criteria were pre-established. |
| Non-participation | State how many participants dropped out/declined participation and the reason(s) given OR provide response rate OR state that no participants dropped out/declined participation. |
| Randomization | If participants were not allocated into experimental groups, state so OR describe how participants were allocated to groups, and if allocation was not random, describe how covariates were controlled. |

# Ecological, evolutionary & environmental sciences study design

All studies must disclose on these points even when the disclosure is negative.

| Study description | Briefly describe the study. For quantitative data include treatment factors and interactions, design structure (e.g. factorial, nested, hierarchical), nature and number of experimental units and replicates. |
| Research sample | Describe the research sample (e.g. a group of tagged Passer domesticus, all Stenocereus thurberi within Organ Pipe Cactus National Monument), and provide a rationale for the sample choice. When relevant, describe the organism taxa, source, sex, age range and any manipulations. State what population the sample is meant to represent when applicable. For studies involving existing datasets, describe the data and its source. |
| Sampling strategy | Note the sampling procedure. Describe the statistical methods that were used to predetermine sample size OR if no sample-size calculation was performed, describe how sample sizes were chosen and provide a rationale for why these sample sizes are sufficient. |
| Data collection | Describe the data collection procedure, including who recorded the data and how. |
| Timing and spatial scale | Indicate the start and stop dates of data collection, noting the frequency and periodicity of sampling and providing a rationale for these choices. If there is a gap between collection periods, state the dates for each sample cohort. Specify the spatial scale from which the data are taken |
| Data exclusions | If no data were excluded from the analyses, state so OR if data were excluded, describe the exclusions and the rationale behind them, indicating whether exclusion criteria were pre-established. |
| Reproducibility | Describe the measures taken to verify the reproducibility of experimental findings. For each experiment, note whether any attempts to repeat the experiment failed OR state that all attempts to repeat the experiment were successful. |
| Randomization | Describe how samples/organisms/participants were allocated into groups. If allocation was not random, describe how covariates were controlled. If this is not relevant to your study, explain why. |

| | |
|---|---|
| Blinding | *Describe the extent of blinding used during data acquisition and analysis. If blinding was not possible, describe why OR explain why blinding was not relevant to your study.* |

Did the study involve field work? ☐ Yes ☐ No

## Field work, collection and transport

| | |
|---|---|
| Field conditions | *Describe the study conditions for field work, providing relevant parameters (e.g. temperature, rainfall).* |
| Location | *State the location of the sampling or experiment, providing relevant parameters (e.g. latitude and longitude, elevation, water depth).* |
| Access & import/export | *Describe the efforts you have made to access habitats and to collect and import/export your samples in a responsible manner and in compliance with local, national and international laws, noting any permits that were obtained (give the name of the issuing authority, the date of issue, and any identifying information).* |
| Disturbance | *Describe any disturbance caused by the study and how it was minimized.* |

# Reporting for specific materials, systems and methods

We require information from authors about some types of materials, experimental systems and methods used in many studies. Here, indicate whether each material, system or method listed is relevant to your study. If you are not sure if a list item applies to your research, read the appropriate section before selecting a response.

### Materials & experimental systems

| n/a | Involved in the study |
|---|---|
| ☒ | Antibodies |
| ☒ | Eukaryotic cell lines |
| ☒ | Palaeontology and archaeology |
| ☒ | Animals and other organisms |
| ☐ | ☒ Human research participants |
| ☐ | ☒ Clinical data |
| ☒ | Dual use research of concern |

### Methods

| n/a | Involved in the study |
|---|---|
| ☒ | ChIP-seq |
| ☒ | Flow cytometry |
| ☒ | MRI-based neuroimaging |

## Antibodies

| | |
|---|---|
| Antibodies used | *Describe all antibodies used in the study; as applicable, provide supplier name, catalog number, clone name, and lot number.* |
| Validation | *Describe the validation of each primary antibody for the species and application, noting any validation statements on the manufacturer's website, relevant citations, antibody profiles in online databases, or data provided in the manuscript.* |

## Palaeontology and Archaeology

| | |
|---|---|
| Specimen provenance | *Provide provenance information for specimens and describe permits that were obtained for the work (including the name of the issuing authority, the date of issue, and any identifying information). Permits should encompass collection and, where applicable, export.* |
| Specimen deposition | *Indicate where the specimens have been deposited to permit free access by other researchers.* |
| Dating methods | *If new dates are provided, describe how they were obtained (e.g. collection, storage, sample pretreatment and measurement), where they were obtained (i.e. lab name), the calibration program and the protocol for quality assurance OR state that no new dates are provided.* |

☐ Tick this box to confirm that the raw and calibrated dates are available in the paper or in Supplementary Information.

| | |
|---|---|
| Ethics oversight | *Identify the organization(s) that approved or provided guidance on the study protocol, OR state that no ethical approval or guidance was required and explain why not.* |

Note that full information on the approval of the study protocol must also be provided in the manuscript.

## Animals and other organisms

Policy information about studies involving animals; ARRIVE guidelines recommended for reporting animal research

| | |
|---|---|
| Laboratory animals | *For laboratory animals, report species, strain, sex and age OR state that the study did not involve laboratory animals.* |
| Wild animals | *Provide details on animals observed in or captured in the field; report species, sex and age where possible. Describe how animals were* |

| Wild animals | *caught and transported and what happened to captive animals after the study (if killed, explain why and describe method; if released, say where and when) OR state that the study did not involve wild animals.* |
| --- | --- |
| Field-collected samples | *For laboratory work with field-collected samples, describe all relevant parameters such as housing, maintenance, temperature, photoperiod and end-of-experiment protocol OR state that the study did not involve samples collected from the field.* |
| Ethics oversight | *Identify the organization(s) that approved or provided guidance on the study protocol, OR state that no ethical approval or guidance was required and explain why not.* |

Note that full information on the approval of the study protocol must also be provided in the manuscript.

# Human research participants

Policy information about studies involving human research participants

| Population characteristics | 485 patients with CLL were included in the study. The median age of patients (for those were the data was available) was 65 years old. 74% of patients were males.<br><br>A small subset was enrolled into CLEAR (early stage of the disease, n = 12) and CLL210 (relapse refractory patients, n = 30). All remaining patients were treatment-naïve and in need of treatment according to iwCLL criteria. They were either fit patients receiving frontline treatment with: fludarabine, cyclophosphamide, rituximab (FCR)-based treatment in ARCTIC (n = 61) or AdMIRe (n = 65); or frail patients receiving ofatumumab with either bendamustine or chlorambucil chemo-immunotherapy in RIAltO (n = 92). Patients recruited into FLAIR (n = 225) were randomised to ibrutinib alone or in combination with rituximab or venetoclax or standard first-line FCR treatment. |
| --- | --- |
| Recruitment | Patients were recruited into the clinical trials mentioned above and in parallel to the 100,000 Genome Program in accordance to the Declaration of Helsinki Declaration of ethical principles for medical research. Patients therefore could freely decline participation in either of the studies, but still participate in the other. Recruitment occurred in specialist haematology clinics held in over 200 sites across the UK. Patients received written information and were given at least 24hrs, usually more, to reflect on participation and were allowed to withdraw at any point in the study without having to give a reason. Translations of PIS and ICFs were available for patients belonging to any of the major ethnic minorities in the UK. All participants with good quality DNA available for WGS were included in the study. |
| Ethics oversight | Genomics England Project Ethics and the CLL Pilot ethics (MREC 09/H1306/54) |

Note that full information on the approval of the study protocol must also be provided in the manuscript.

# Clinical data

Policy information about clinical studies

All manuscripts should comply with the ICMJE guidelines for publication of clinical research and a completed CONSORT checklist must be included with all submissions.

| Clinical trial registration | CLEAR (NCT01279252)<br>CLL210 (EudraCT 2010-019575-29)<br>RIAltO (NCT01678430)<br>FLAIR (EudraCT 2013-001944-76)<br>ARCTIC (EudraCT Number:2009-010998-20)<br>AdMIRe (EudraCT number: 2008-006342-25 |
| --- | --- |
| Study protocol | Full protocols of these clinical studies are considered confidential by the clinical trial units of Leeds and Liverpool Universities and Kings College London<br><br>Details of the clinical trial protocols are published:<br><br>Andrew R. Pettitt et al. Lenalidomide, dexamethasone and alemtuzumab or ofatumumab in<br><br>high-risk chronic lymphocytic leukaemia: final results of the NCRI CLL210 trial. Haematologica<br><br>105, 2868–2871 (2020)<br><br>Howard, D. R. et al. Results of the randomized phase IIB ARCTIC trial of low-dose rituximab in<br><br>previously untreated CLL. Leukemia 31, 2416–2425 (2017).<br><br>Munir, T. et al. Results of the randomized phase IIB ADMIRE trial of FCR with or without<br><br>mitoxantrone in previously untreated CLL. Leukemia 31, 2085–2093 (2017).<br><br>Collett, L. et al. Assessment of ibrutinib plus rituximab in front-line CLL (FLAIR trial): study<br><br>protocol for a phase III randomised controlled trial. Trials 18, 387 (2017). |
| Data collection | Patients were recruited from over 200 secondary and tertiary haematology referral services across the UK between 2014 and 2018 |

| | |
|---|---|
| | Baseline and clinical outcome data were collected prospectively across all sites and underwent source verification and data monitoring by an independent data monitoring committee. |
| Outcomes | For all trials, the primary endpoint was progression free survival. Secondary endpoints were overall response, safety and overall survival. Outcome measures were assessed according to iwCLL response criteria (Hallek M et al 2008). Safety data was recorded following CTCAE vs 4.0.<br>Outcomes were collected in accordance to internationally agreed definitions for overall response rate, progression free survival, overall survival and minimal residual disease as per Hallek, M. et al. iwCLL guidelines for diagnosis, indications for treatment, response assessment, and supportive management of CLL. Blood 131, 2745–2760 (2018).<br><br>Safety outcomes were defined by CTCAE (for version, see publications of the respective studies). |

# Dual use research of concern

Policy information about dual use research of concern

## Hazards

Could the accidental, deliberate or reckless misuse of agents or technologies generated in the work, or the application of information presented in the manuscript, pose a threat to:

No | Yes
☐ | ☐ Public health
☐ | ☐ National security
☐ | ☐ Crops and/or livestock
☐ | ☐ Ecosystems
☐ | ☐ Any other significant area

## Experiments of concern

Does the work involve any of these experiments of concern:

No | Yes
☐ | ☐ Demonstrate how to render a vaccine ineffective
☐ | ☐ Confer resistance to therapeutically useful antibiotics or antiviral agents
☐ | ☐ Enhance the virulence of a pathogen or render a nonpathogen virulent
☐ | ☐ Increase transmissibility of a pathogen
☐ | ☐ Alter the host range of a pathogen
☐ | ☐ Enable evasion of diagnostic/detection modalities
☐ | ☐ Enable the weaponization of a biological agent or toxin
☐ | ☐ Any other potentially harmful combination of experiments and agents

# ChIP-seq

## Data deposition

☐ Confirm that both raw and final processed data have been deposited in a public database such as GEO.

☐ Confirm that you have deposited or provided access to graph files (e.g. BED files) for the called peaks.

Data access links
*May remain private before publication.*
*For "Initial submission" or "Revised version" documents, provide reviewer access links.  For your "Final submission" document, provide a link to the deposited data.*

Files in database submission
*Provide a list of all files available in the database submission.*

Genome browser session
(e.g. UCSC)
*Provide a link to an anonymized genome browser session for "Initial submission" and "Revised version" documents only, to enable peer review.  Write "no longer applicable" for "Final submission" documents.*

## Methodology

Replicates
*Describe the experimental replicates, specifying number, type and replicate agreement.*

Sequencing depth
*Describe the sequencing depth for each experiment, providing the total number of reads, uniquely mapped reads, length of reads and whether they were paired- or single-end.*

Antibodies
*Describe the antibodies used for the ChIP-seq experiments; as applicable, provide supplier name, catalog number, clone name, and lot number.*

| Peak calling parameters | *Specify the command line program and parameters used for read mapping and peak calling, including the ChIP, control and index files used.* |
|---|---|
| Data quality | *Describe the methods used to ensure data quality in full detail, including how many peaks are at FDR 5% and above 5-fold enrichment.* |
| Software | *Describe the software used to collect and analyze the ChIP-seq data. For custom code that has been deposited into a community repository, provide accession details.* |

# Flow Cytometry

## Plots

Confirm that:

☐ The axis labels state the marker and fluorochrome used (e.g. CD4-FITC).

☐ The axis scales are clearly visible. Include numbers along axes only for bottom left plot of group (a 'group' is an analysis of identical markers).

☐ All plots are contour plots with outliers or pseudocolor plots.

☐ A numerical value for number of cells or percentage (with statistics) is provided.

## Methodology

| Sample preparation | *Describe the sample preparation, detailing the biological source of the cells and any tissue processing steps used.* |
|---|---|
| Instrument | *Identify the instrument used for data collection, specifying make and model number.* |
| Software | *Describe the software used to collect and analyze the flow cytometry data. For custom code that has been deposited into a community repository, provide accession details.* |
| Cell population abundance | *Describe the abundance of the relevant cell populations within post-sort fractions, providing details on the purity of the samples and how it was determined.* |
| Gating strategy | *Describe the gating strategy used for all relevant experiments, specifying the preliminary FSC/SSC gates of the starting cell population, indicating where boundaries between "positive" and "negative" staining cell populations are defined.* |

☐ Tick this box to confirm that a figure exemplifying the gating strategy is provided in the Supplementary Information.

# Magnetic resonance imaging

## Experimental design

| Design type | *Indicate task or resting state; event-related or block design.* |
|---|---|
| Design specifications | *Specify the number of blocks, trials or experimental units per session and/or subject, and specify the length of each trial or block (if trials are blocked) and interval between trials.* |
| Behavioral performance measures | *State number and/or type of variables recorded (e.g. correct button press, response time) and what statistics were used to establish that the subjects were performing the task as expected (e.g. mean, range, and/or standard deviation across subjects).* |

## Acquisition

| Imaging type(s) | *Specify: functional, structural, diffusion, perfusion.* |
|---|---|
| Field strength | *Specify in Tesla* |
| Sequence & imaging parameters | *Specify the pulse sequence type (gradient echo, spin echo, etc.), imaging type (EPI, spiral, etc.), field of view, matrix size, slice thickness, orientation and TE/TR/flip angle.* |
| Area of acquisition | *State whether a whole brain scan was used OR define the area of acquisition, describing how the region was determined.* |

Diffusion MRI   ☐ Used   ☐ Not used

## Preprocessing

| Preprocessing software | *Provide detail on software version and revision number and on specific parameters (model/functions, brain extraction, segmentation, smoothing kernel size, etc.).* |
|---|---|
| Normalization | *If data were normalized/standardized, describe the approach(es): specify linear or non-linear and define image types used for* |

| Normalization | *transformation OR indicate that data were not normalized and explain rationale for lack of normalization.* |

| Normalization template | *Describe the template used for normalization/transformation, specifying subject space or group standardized space (e.g. original Talairach, MNI305, ICBM152) OR indicate that the data were not normalized.* |

| Noise and artifact removal | *Describe your procedure(s) for artifact and structured noise removal, specifying motion parameters, tissue signals and physiological signals (heart rate, respiration).* |

| Volume censoring | *Define your software and/or method and criteria for volume censoring, and state the extent of such censoring.* |

## Statistical modeling & inference

| Model type and settings | *Specify type (mass univariate, multivariate, RSA, predictive, etc.) and describe essential details of the model at the first and second levels (e.g. fixed, random or mixed effects; drift or auto-correlation).* |

| Effect(s) tested | *Define precise effect in terms of the task or stimulus conditions instead of psychological concepts and indicate whether ANOVA or factorial designs were used.* |

Specify type of analysis: ☐ Whole brain ☐ ROI-based ☐ Both

| Statistic type for inference (See Eklund et al. 2016) | *Specify voxel-wise or cluster-wise and report all relevant parameters for cluster-wise methods.* |

| Correction | *Describe the type of correction and how it is obtained for multiple comparisons (e.g. FWE, FDR, permutation or Monte Carlo).* |

## Models & analysis

| n/a | Involved in the study |
|-----|------------------------|
| ☐ | ☐ Functional and/or effective connectivity |
| ☐ | ☐ Graph analysis |
| ☐ | ☐ Multivariate modeling or predictive analysis |

| Functional and/or effective connectivity | *Report the measures of dependence used and the model details (e.g. Pearson correlation, partial correlation, mutual information).* |

| Graph analysis | *Report the dependent variable and connectivity measure, specifying weighted graph or binarized graph, subject- or group-level, and the global and/or node summaries used (e.g. clustering coefficient, efficiency, etc.).* |

| Multivariate modeling and predictive analysis | *Specify independent variables, features extraction and dimension reduction, model, training and evaluation metrics.* |

