## [Peer Review File · Nature Genetics]

Peer Review Information

Manuscript Title: Whole-genome sequencing of chronic lymphocytic leukemia identifies subgroups with distinct biological and clinical features

Corresponding author name(s): Professor Anna Schuh

Reviewer Comments & Decisions:

Decision Letter, initial version:

29th Jun 2021

Dear Professor Schuh,

I'm so sorry this decision has taken so long. In the end, we chose to move forward with the two reports we had as your third reviewer was not responding to any of our queries. Thank you so much for your patience.

Your Article, "The non-coding landscape of chronic lymphocytic leukaemia and its association with clinical outcome" has now been seen by 2 referees. You will see from their comments below that while they find your work of interest, some important points are raised. We are interested in the possibility of publishing your study in Nature Genetics, but would like to consider your response to these concerns in the form of a revised manuscript before we make a final decision on publication.

To guide the scope of the revisions, the editors discuss the referee reports in detail within the team, including with the chief editor, with a view to identifying key priorities that should be addressed in revision and sometimes overruling referee requests that are deemed beyond the scope of the current study. In this case, we believe that both reviewers have provided clear guidance on how to improve the technical robustness of the analysis and we agree that all their comments should be addressed in full.

We therefore invite you to revise your manuscript taking into account all reviewer and editor comments. Please highlight all changes in the manuscript text file. At this stage we will need you to upload a copy of the manuscript in MS Word .docx or similar editable format.

*2) If you have not done so already please begin to revise your manuscript so that it conforms to our Article format instructions, available

[here](http://www.nature.com/ng/authors/article_types/index.html).

*3) Include a revised version of any required Reporting Summary:

[REDACTED]

We hope to receive your revised manuscript within six months. If you cannot send it within this time, please let us know.

2Sincerely,

Safia Danovi
Editor
Nature Genetics

Referee expertise:

Referee #1: CLL genomics

Referee #2: CLL

Please note that Reviewer #3 did not return their report.

Reviewers' Comments:

Reviewer #1:

Remarks to the Author:

The authors performed a comprehensive analysis of the whole genome of 485 chronic lymphocytic leukemia (CLL) patients to define new genomic features of the disease. They report the identification of 25 novel coding "drivers" and new genomic subgroups associated with clinical outcome, refining previous prognostic models based on the analysis of coding mutations only.

The strengths of this study are the large number of cases with whole genome sequencing data (although obtained at different stages of the disease), and the controlled cohort that includes patients enrolled in clinical trials, thus allowing to study uniformly treated populations. However, there are three major areas of weakness: 1) evidence for a functional impact of the noncoding mutations; 2) robustness of the omics analyses; 3) critical interpretation of affected biological programs and clinical subgroups. Overall, the results are of purely descriptive nature, and conclusions are often of largely speculative nature.

Specific comments:

1. IDENTIFICATION OF NON-CODING DRIVERS:

- evidence for the functional relevance of representative mutations should be provided to strengthen the overall impact and originality of the study.

- It is unclear how the authors functionally annotated the non-coding regions. Specifically, while the

3ATAC-Seq analysis appears relatively straightforward, there is no explanation on how the many ChIP-seq histone modification data were integrated. Besides basic information such as cutoffs for significance of the peak calls, reproducibility through duplicate experiments, representative plots of the histone/ATAC-Seq profiles for the novel mutated enhancers, etc, what is completely missing is how activating (e.g. H3K27Ac, H3K4me1) vs silencing (e.g. H3K27me3) marks were integrated, and how chromatin domains were annotated as poised, silenced, active, primed enhancer/promoter based on their overlap. Moreover, the same regulatory element may be active in some cases and inactive in others. This is likely the case for mutated vs unmutated CLL, which have been shown to display distinct transcriptional signatures when compared directly. In the absence of ChIP-Seq data from the very same case (only 12 samples with complete RNA-seq and histone marks), how did the authors assign a mutation to an open or closed region of the genome? Finally, even for "overlapping" regions, the genomic coordinates of any given histone modification peak are quite heterogeneous across different samples: did the authors redefine a "common region" based on their overlap, or did they use the "union" of the peaks (the latter may be less robust).

-It is also unclear how the identified promoter-distal regulatory elements were assigned to predicted target genes. Expression is valid, but insufficient per se: mapping to the same topologically associated domain, proximity, and specific cutoffs for expression level should be integrated, again separately for M and UM cases.

-The size of REs can span several megabases, and the presence of sparse genetic events may not translate into a common functional impact: how did the authors define "recurrent" non-coding mutation? Frequencies should be provided, rather than p values, and display items showing representative "hotspot" regions at base pair resolution, with the mutations found at specific binding sites, would be of help.

2. IDENTIFICATION OF NOVEL CODING DRIVERS:

- There is no information about the nature of the mutation in these new genes (missense? Truncating? Heterozygous? Homozygous?). Since most appear to be missense events, their functional significance is unclear, preventing their interpretation as pathogenic.

-What are the "more permissive criteria" used for gene selection of additional hotspot genes?

-What is the allelic fraction of the variants identified (same question applies to the copy number aberrations below). Are the authors considering subclonal events – which again would have unclear functional significance to the pathogenesis of the disease, unless the authors can demonstrate their progressive expansion during disease evolution.

3. COPY NUMBER ANALYSIS:

- It seems that 105 samples were analyzed by SNP genotyping and it is thus presumed that the rest were called based on sequencing-based tools: please clarify and provide criteria for CN calls.

- What is the basis for nominating individual genes as the targets of the aberration? Were these in focal CN losses/gains? Were these clonal or subclonal? If these alterations are heterozygous, is expression of the residual allele lost? Or do the authors imply that these new targets of CN losses are

4haploinsufficient tumor suppressors?

4. MUTATIONAL SIGNATURES:

-The proposed use of SBS9 as a surrogate marker to assess IGHV mutated and unmutated CLL is interesting but not convincing (please provide cosine similarity for all analyses). Indeed, the % of CLL-M samples correctly identified using this signature was >80% (supplementary methods 2). Moreover, the SBS9 signature has been associated with both AID and pol-eta. This proposed approach would probably require a manuscript on its own.

5. DATA ORGANIZATION:

- Key information and conceptual criteria allowing the reader to understand how the analyses were performed and how conclusions were made should be provided in the main body of the manuscript.

- Some of the figures are impossible to read (eg Figure2, Figure 4, extended Figure 7)

- Figure 1: there is no labeling to the y axis of Figure 1b

Reviewer #2:

Remarks to the Author:

This manuscript describes a comprehensive whole genome analysis, encompassing both coding and non-coding variants, and genome-wide genomic lesions / CNAs in a cohort of 485 chronic lymphocytic leukemia (CLL) patients. Several novel coding driver genes were identified, as well as non-coding variants through the use of known regulatory elements in CLL. These non-coding variants partly occurred in genes and pathways that have been known to be affected by coding variants. Multi-layer genomic data integration (coding, non-coding and genome-wide lesions) allowed to distinguish CLL into genomic subgroups. Comparison to clinical data in a sub-cohort of the CLL patients resulted in associations between these genomic subgroups and outcome.

Building on existing genomic data sets in CLL that mostly concerned either coding variants, or -to some extent- non-coding variants, the clear added value of this study is the integration of coding and non-coding SNVs / indels plus genomic lesions, which allows a much more comprehensive evaluation of the entire mutational landscape in the largest CLL cohort evaluated in this way so far. Another plus of this study is that for interpreting non-coding variants the authors have made use of ATAC-seq and ChIP-seq data pointing to relevant regulatory elements (promoters, enhancers, UTR) to interpret non-coding variants. Overall, this is an interesting paper, but I have several major comments, as listed below.

1. A major comment is that the whole study is very oncogenetically oriented, whilst it is obvious from numerous papers over a period of 20 years or so, that immunogenetics is another important factor in CLL. Admitted, the authors divide CLL into mutated and unmutated CLL, but immunogenetics meanwhile goes far beyond this. Bifurcation into M-CLL and U-CLL is too simple. It is well known that there is an intermediate group, with borderline mutation status and a different epigenetic profile, and also particular stereotypic CLL subgroups are now well-established. In this respect this reviewer finds it crucial to make further comparisons with immunogenetically defined to better understand the biology of the here introduced CLL subgroups. This especially holds for the big division into M-CLL and

5U-CLL in comprehensively analyzing genomic data, as in lines 219 and 220, where esp. a borderline category could be introduced. And it also holds in a more specific way for the subgroups as summarized in lines 258-261. Is m-GS2 perhaps equivalent to stereotypic subsets #4 or satellites? And does the GC 7/8 subgroups harbor poor stereotypic subsets such as #1, #2, and #8?

2. In comparing outcome to subgroups, 44 genomic features were found to predict PFS and/or OS. These are all univariate analyses, but how many did survive multivariate analysis. Could the ones that remain after multivariate analyses perhaps be the real relevant ones?
3. Are the identified coding mutations in known drivers in the same hotspots or are there new hotspots. Also, with respect to new drivers, are the found mutations clustered? All this information is very useful for further future analysis in new patient samples and should be provided somewhere.
4. With respect to mutations in UTRs, there are currently only associations based on identified RE. But is there any proven effect on expression levels. For me it would be mandatory to check this based on RNA seq data or qPCR experiments to make statements that these variants are truly pathogenic?
5. In line 62 I would like to see split out inversions in immunoglobulin light chain genes into kappa and lambda, and it should also be added what partners are seen in those inversions.
6. In Figure 3 genomic complexity leads to subgrouping into 8 group. Group 5 seems to be lacking in the survival analysis in Extended data Figure 15, or is this due to the small group size? The rationale to further lump GC groups is unclear, is this purely empirical, is it eyeballing, or is there any idea behind it?
7. In Figure 4e the PFS curve comes with a clear p value for the different u-GS groups, yet in line 234 this is referred to as similar PFS. What is correct here?

Minor comments

- Readability of many Figures is poor; the information is dense and fonts of labels, axes, tables are often too small to read or grasp the message.
- Figure 2c: label is included for "coding" variants, but the whole point of Figure 2 is the "non-coding" variants, so in my opinion it should be left out.
- Figure 2b: the legend is entirely unclear as to what is exactly shown in the figure.
- Figure 3f: the DBS seems to be gone; is this grouped with SBS or simply left out?
- Figure 4: the number of u-GS and m-GS groups is not consistent between main text (lines 221-222), graph and legend. Please clarify whether it should read as 2 or 3 groups in both m-GS and u-GS.
- Extended data Figure 2c: lacks a color coding, either in the graph or in the legend.
- Extended data Figure 5a: the bottom panel is a complete repeat of what is in Figure 2c, so can be left out here.

Reviewer #3:

None

Author Rebuttal to Initial comments

61

We thank both reviewers for the time they took to carefully read and constructively critique our
manuscript. We are grateful for their comment and suggestions and hope that the additional work
over the past 9 months has addressed their questions to full satisfaction. The review process has been
certainly very beneficial to us, and we hope that it has improved the quality of the data presented
significantly.

Below, we address each of the reviewer's comments.

The document is written as follow for clarity: text in green font indicates comment/question from the
reviewers; text in black font denotes our response to the question and text in blue font presents
quotes directly taken from the revised manuscript.

Reviewer #1

Remarks to the Author:

The authors performed a comprehensive analysis of the whole genome of 485 chronic lymphocytic leukemia (CLL) patients to define new genomic features of the disease. They report the identification of 25 novel coding “drivers” and new genomic subgroups associated with clinical outcome, refining previous prognostic models based on the analysis of coding mutations only.

The strengths of this study are the large number of cases with whole genome sequencing data (although obtained at different stages of the disease), and the controlled cohort that includes patients enrolled in clinical trials, thus allowing to study uniformly treated populations. However, there are three major areas of weakness:

- 1) evidence for a functional impact of the noncoding mutations;
- 2) robustness of the omics analyses;
- 3) critical interpretation of affected biological programs and clinical subgroups. Overall, the results are of purely descriptive nature, and conclusions are often of largely speculative nature.

Specific comments:

1. IDENTIFICATION OF NON-CODING DRIVERS:

1.1- Evidence for the functional relevance of representative mutations should be provided to strengthen the overall impact and originality of the study.

While we agree that there is a need to better understand the biological function of individual genes, this was not the aim of this current study. Our goal was to define the combined impact of the genomic landscape on clinical outcomes for individual patients. To study the biological function of individual mutations in previously characterised or novel regulatory elements requires a completely different approach. Moreover, it is known that differences in gene expression and other readouts from functional assays caused by these mutations can be subtle and often do not explain clinical outcomes. Nevertheless, we have addressed the reviewers’ comments for a selected number of candidate non-coding drivers as a proof-of-principle that our bioinformatics pipeline can robustly predict enhancer-gene links and detect candidate non-coding drivers.

This was explored by first performing in-silico prediction of interesting candidate regions affected by acquired mutations and then generating RNA-seq data and/or ATAC-seq data (to identify allelic skewing of chromatin accessibility) in selected samples that carried somatic mutation in different regulatory elements (BCL2 5’UTR, BCL6 enhancer, BACH2 and ATAD1 promoters). We also performed CRISPR-Cas9-based genome editing of one of the hypermutated candidate regions to confirm the enhancer-gene link.

Regarding BCL2 5’UTR, the results are shown in the manuscript as follow:
Novel significantly mutated UTRs included the 5’UTR of BCL2 (n = 6; FDR = 1.01 x 10⁻⁶ Fig.5a). We performed RNA-seq on samples carrying these mutations (Methods, Supplementary table 4)

demonstrating that 5'UTR mutations were associated with *BCL2* overexpression ($p = 4.3 \times 10^{-2}$, **Fig.5b**),
which is noteworthy given that *BCL2* inhibitors are used therapeutically ¹.

Regarding *BCL6* enhancer, the results are shown in the manuscript as follow:

Another region spanning 325kb on chr3q27.2 contained seven significantly mutated enhancers and
linked to *BCL6* (**Extended Data Fig.6a, Supplementary Table 17**). In CLL, this region contains scattered
enhancers whereas in germinal centre B-cells, they are joined into a large super-enhancer element^{2,3}.
Using CRISPR/Cas9 genome editing in lymphoblast-like Raji cells, we introduced a 202 bp deletion in
the most significantly mutated region (e*BCL6_2*, $n = 8$ samples, $FDR = 5.87 \times 10^{-16}$) 200 kb downstream
of *BCL6* (**Extended Data Fig.7c**). The deletion led to a 35% reduction in *BCL6* expression and a 40%
decrease in cell proliferation, confirming this region as an active enhancer of *BCL6* (**Extended Data**
**Fig.7d**). RNA-seq of these 8 samples with mutations in this region showed overall increased expression
of *BCL6*, although the effect was heterogenous (**Fig.5c**) and suggesting that some variants are more or
less pathogenic and might exert a positional effect (**Fig.5d**).

To narrow down the list of candidates to examine with additional experiments, we used DeepHaem
to investigate the functional impact of mutations in promoters on transcription factor binding. This
analysis identified *BACH2* promoter. The results of these analyses are presented in the manuscript, as
follow:

Next, we investigated mutations in these promoters further to identify those predicted to change
chromatin state, using DeepHaem⁴, a deep neural network trained on chromatin feature data of 73
immune cell types. Seventy-four variants were predicted to lead to a loss of open chromatin (*i.e.* loss-
of-function variants), including those in the *BACH2* promoter (**Fig.6a, Extended Data Fig.8a**). A recent
study showed that decreased *BACH2* expression in CLL is associated with adverse outcomes⁵. Besides,
the mutations we detected in this promoter were mostly clonal (median CCF = 0.99). We therefore
investigated this promoter further by performing ATAC-seq and RNA-seq (**Fig.6b**) on mutated samples,
when available (13 variants investigated, **Methods, Supplementary table 12**) to understand the
impact of these variants on chromatin accessibility and gene expression. Three variants within a 14 bp
region were associated with allelic skew in the ATAC-seq compared to WGS data, demonstrating a
preference for accessibility on the reference allele (**Fig.6c**), which mirrored the decrease in chromatin
accessibility in that region compared to WT samples (**Fig.6d**). This allelic skew was also detected at the
RNA level (**Fig.6e, Extended Data Fig.8b**). In addition, the same three samples also showed decreased
*BACH2* gene expression (**Fig.6f**).

Finally, our analysis of all samples with trios of paired WGS, ATAC-seq and RNA-seq data ($n=20$)
suggested that five promoters showed interesting potential gain/loss of function mutations, as
detailed in the manuscript, focusing on *ATAD1*.

Finally, we analysed 20 cases with paired WGS, ATAC-seq and RNA-seq data (**Supplementary table 4**).
We identified five recurrently mutated promoters with allelic skewing of chromatin accessibility and
RNA expression. Three, *BTG2*, *CCND1* and *ST6GAL1*, were associated with allelic skewing towards the
mutant allele, whereas *ATAD1* and *BIRC3* showed the opposite effect (**Extended Data Fig.8c**). In the
case of *ATAD1*, which plays a role in mitochondria protein degradation, we additionally observed
reduced expression in promoter-mutated samples ($p = 7.0 \times 10^{-4}$) (**Extended Data Fig.8d-e-f**).

Collectively, these data suggest that at least some of the non-coding mutations in CLL have driver characteristics, target REs, and exert subtle effects on chromatin accessibility and gene expression levels of target genes that are critical for B-cell development and function as well as cancer progression.

Associated methods for these additional experiments are detailed in the methods section.

RNA-seq

Libraries were prepared from samples of 74 patients using the Illumina Stranded Total RNA Prep, Ligation with Ribo-Zero Plus, with additional custom depletion probes, using 100ng RNA. Libraries were sequenced on the Illumina NovaSeqTM 6000 using 100 base paired-end chemistry (108 – 455 million read-pairs per sample). Sequencing reads were processed and aligned to Human Reference genome GRCh38 using the Illumina Dragen RNA pipeline v3.8.4. Gentyotyping was performed using bcftools mpileup⁶. Allele specific read counts were generated at sites of acquired SNVs determined by WGS.

ATAC-seq

ATAC-seq was performed as previously described⁷. Briefly 7.5×10^4 cells per technical replicate were resuspended in lysis buffer (10 mM of Tris-HCl, pH 7.5, 10 mM of NaCl, 3 mM of MgCl₂, 0.1% IGEPAL CA-630). Nuclei were pelleted (500g for 10 min), PBS was discarded and nuclei were resuspended in tagmentation buffer (25 μ l of 2 \times tagmentation DNA buffer, 2.5 μ l of Tn5 Transposase (Illumina) and 22.5 μ l of water) then incubated (37 °C for 30 min). DNA was extracted using the MinElute PCR Purification Kit (QIAGEN), half the DNA was amplified (NEBNext High-Fidelity 2 \times PCR Master Mix (New England Biolabs)) and purified with the QIAquick PCR Purification Kit (QIAGEN). Libraries were sequenced using 40-bp paired-end reads (Illumina NextSeq).

Reads were mapped to GRCh38 using the PEPATAC pipeline with pre-alignment to the mitochondrial genome and default settings⁸. Gentyotyping was performed using bcftools mpileup⁶. Allele specific ATAC-seq read counts were generated at sites of acquired SNVs determined by WGS.

Deletion of *BCL6* enhancer in Raji cells using CRISPR/Cas9

The *BCL6* enhancer was deleted in Raji cells using CRISPR/Cas9 genome editing. Guide RNAs (gRNAs) were designed using the CRISPick^{9,10}, flanking the enhancers predicted transcriptional start site (TSS) as determined by examining the Cap Analysis of Gene Expression data of the Raji cells in the FANTOM5¹¹ dataset in ZENBU genome browser¹². The sequences of the gRNAs used, from 5' to 3', are: AGAGCATTCATGTCCTGAAA and TACTGAGATGATTACTACC. The deleted region was 202 bp.

Raji cells were cultured in 10% FCS/RPMI 1640 medium and transfected with an RNP complex of 3.9 pmol of total synthetic sgRNA (Synthego) and 3 pmol Alt-R S.p. HiFi Cas9 Nuclease V3 (Integrated DNA Technologies) using the Neon transfection system (ThermoFisher), as per the manufacturer's instructions.

Three days post-transfection, genomic DNA was harvested using the DNeasy Blood & Tissue Kit (Qiagen) and deletion events evaluated by PCR with primers flanking the deleted region. A reduction in amplicon size was observed from 383 bp (WT) to 181 bp. A noticeable drop in cell number (60% the number of the control cells 3 days post-transfection) was also observed in the eBCL6 deleted cells.

Additionally, 3 days post-transfection, total RNA was extracted using Trizol LS (Invitrogen), as per the
manufacturer's instructions. Real-time quantitative RT-PCR was performed using the One Step
PrimeScript™ RT-PCR Kit (Takara Bio) using a 7500 Real-Time PCR system (Applied Biosystems). Two
primer pairs were used against each target (*BCL6* gene and e*BCL6*), and expression levels normalized
to the housekeeping gene *GAPDH*. The primers were as follow: e*BCL6*_F1
CCAGAGCATTCATGTCCTGAAAAG; e*BCL6*_R1 GCTAGTCAGAGGAATCATCCGTT; e*BCL6*_F2
TCAAAGGCATAGCGAGTGGC; e*BCL6*_R2 GGCCTGTGACTGAAGACACTT; *BCL6*_F1
CACACCCGTCCATCATTGAA; *BCL6*_R1 TGTCTCACGGTGCCTTTTT; *BCL6*_F2 GGCCGGACACCAGGTTTT;
*BCL6*_R2 TGGGCTCTAAACTGCTCACG. Expression fold changes were calculated using the $\Delta\Delta C_t$ method.

1.2-

**(a) It is unclear how the authors functionally annotated the non-coding regions. Specifically, while**
**the ATAC-Seq analysis appears relatively straightforward, there is no explanation on how the many**
**ChIP-seq histone modification data were integrated. Besides basic information such as cutoffs for**
**significance of the peak calls, reproducibility through duplicate experiments, representative plots of**
**the histone/ATAC-Seq profiles for the novel mutated enhancers, etc, what is completely missing is**
**how activating (e.g. H3K27Ac, H3K4me1) vs silencing (e.g. H3K27me3) marks were integrated, and**
**how chromatin domains were annotated as poised, silenced, active, primed enhancer/promoter**
**based on their overlap.**

Functional annotation of the non-coding regions was carried out using chromatin state data including
ChIP-seq analysis that was previously generated in primary CLL cells and that has been published
(Beekman et al. Nat Med. 2018). Briefly, the integration of multiple ChIP-seq data was carried out
using the chromHMM software. This software defines a matrix of presence or absence of methylation
marks across binned genome segments and uses a multivariate hidden Markov model to learn the
chromatin state signatures. The data used here comprises 6 methylation marks; H3K4me3, H3K4me1,
H3K27ac, H3K36me3, H3K27me3 and H3K9me3, resulting in 12 chromatin states over 200 base pair
genome-wide bins. The ChIP-seq histone modification data used for this analysis was derived from 7
CLL samples, including 5 hypermutated samples and 2 unmutated; peaks were called using MACS2
and processed as described in the published work.

Initially, only mutations in regions overlapping strong enhancers or active promoters were considered
for downstream analysis. This analysis has been expanded to include a comprehensive
characterization of the non-coding genome using the chromatin state data processed with
chromHMM as described above. Mutations were annotated into 5 functional states based on their
overlaps to characterised regions;

- 1. Heterochromatin/silenced (Het;LowSign and H3K9me3 chmm).
- 2. H3K27me3 repressed (poised promoters and H3K27me3 chmm).
- 3. Transcription (WkTxn, H3K36me3 and TxnTrans chmm)
- 4. Enhancers (WkEnh, StrEnh2).
- 5. Promoters (WkProm, ActProm, StrEnh1)

Variants overlapping with the chromatin states annotated as promoters and enhancers without
significant genetic imprints of AID-mediated somatic hypermutation were considered for all
downstream analyses.

The methods in the manuscript have been altered to describe this extended approach to defining the
noncoding regions:

**Non-coding variant annotations**

Variants outside coding regions were annotated against non-coding regulatory elements (RE) datasets
(**Fig.4a**). All promoter regions were defined as +/- 250bp of GENCODE v29 transcription start sites. In
addition, these REs were further complemented using previous available ChIP-seq data and chromatin
state segmentation of primary CLL samples¹³. More specifically, these REs were defined by first
intersecting previously identified H3K27ac peaks and open chromatin regions defined by ATAC-seq
(derived from 104 and 106 primary CLL cases, respectively)¹³. To consider potential differences related
to IGHV status, we performed a differential analysis between m-IGHV and u-IGHV CLL cases using
H3K27ac and ATAC-seq peaks in 39 u-IGHV and 63 m-IGHV samples for H3K27ac ChIP-seq data, and
38 u-IGHV and 66 m-IGHV cases for ATAC-seq data. These differential analyses allowed us to precisely
classify active/open chromatin regions in (1) significantly higher in u-IGHV CLL, (2) significantly higher
in m-IGHV CLL and (3) no significantly different between u- and m-IGHV CLL, and therefore named
"CLL". Mutations were then annotated using this classification allowing us to identify potential
enrichments in particular IGHV subgroups.

Next, mutations were further annotated using genome-wide segmentations of 7 previously published
CLL samples (5 mutated and 2 unmutated IGHV cases) with available ChIP-seq data of six non-
redundant histone marks including H3K4me3, H3K4me1, H3K27ac, H3K36me3, H3K27me3 and
H3K9me3. The segmentation was done with the chromHMM software¹⁴ as previously described¹³ and
gave rise to 12 chromatin states (chmm). These 12 chmm were ActProm (active promoter, with
H3K27ac and H3K4me3), WkProm (weak promoter, with H3K4me1 and H3K4me3), PoisProm (poised
promoter, with H3K27me3, H3K4me1 and H3K4me3), StrEnh1 (strong enhancer 1, with H3K27ac,
H3K4me1 and H3K4me3), StrEnh2 (strong enhancer 2, with H3K27ac and H3K4me1), WkEnh (weak
enhancer, with H3K4me1), TxnTrans (transcription transition, with H3K36me3, H3K27ac and
H3K4me1), TxnElong (transcription elongation, with H3K36me3), WkTxn (weak transcription, with low
H3K36me3), H3K9me3 (H3K9me3-marked repressed heterochromatin), H3K27me3 (H3K27me3-
marked repressed heterochromatin) and Het;LowSign (low-signal heterochromatin, with the absence
of all six histone marks). Of note, as our annotations of non-coding variants were based on CLL samples
from different cohorts than the WGS cohort, regions with the same chromatin state in at least 2
samples were used.

All non-coding variants were annotated with the chmm of the region they occurred in according to
five grouped functional states:

1. Heterochromatin/silenced (Het;LowSign and H3K9me3 chmm).
2. H3K27me3 repressed (poised promoters and H3K27me3 chmm).
3. Transcription (WkTxn, H3K36me3 and TxnTrans chmm)
4. Enhancers (WkEnh, StrEnh2).
5. Promoters (WkProm, ActProm, StrEnh1)

Variants overlapping with the chromatin states annotated as promoters and enhancers without
significant genetic imprints of AID-mediated somatic hypermutation were considered for all

238 downstream analyses. Previous work has established a cut-off of 35% of variants caused by off-target
AID activity¹⁵ in REs as highly enriched in AID signature, and thus was used in our analyses
(**Supplementary table S17**). C>T/G mutations in WR_CY motifs and their reverse complement
(canonical AID) were considered to be AID-linked. The remaining variants not overlapping with
protein-coding sequences, non-coding gene sequences, splice sites, UTRs, promoters or enhancers
were annotated as non-regulatory regions.

An overview of the methods is also written in the result section, including a figure:

**Non-coding driver mutations**

To gain insight into the functional impact of non-coding mutations, we first identified CLL-specific
regulatory elements (RE) by integrating ATAC-seq and H3K27ac profiles^{13,16} as well as chromatin
states¹⁴ from publicly available primary CLL (Fig.4a, Methods). Out of the 29,224 promoters and 56,137
enhancers identified, 90% were present in CLL as a whole, whereas the remaining 10% were specific
for IGHV subgroups and were used for the IGHV subtype-specific annotation (Methods).

All chromatin data can be viewed in the Genome browser:

[http://resources.idibaps.org/paper/the-reference-epigenome-and-regulatory-chromatin-landscape-](http://resources.idibaps.org/paper/the-reference-epigenome-and-regulatory-chromatin-landscape-of-chronic-lymphocytic-leukemia)
[of-chronic-lymphocytic-leukemia](http://resources.idibaps.org/paper/the-reference-epigenome-and-regulatory-chromatin-landscape-of-chronic-lymphocytic-leukemia)

Representative genomic views were included in the manuscript in Figure 5d (enhancers of BCL6) and
Supplementary Figure 7b (enhancers of PAX5).

**(b) Moreover, the same regulatory element may be active in some cases and inactive in others. This**
**is likely the case for mutated vs unmutated CLL, which have been shown to display distinct**
**transcriptional signatures when compared directly.**

We agree that the same regulatory element may be active in some cases and inactive in others and in
particular between hypermutated and unmutated CLL cases. Therefore, we generated matrices for
ATAC-Seq and H3K27ac of regions active in m-CLL and u-CLL separately. This allowed us to produce a
list of m-IGHV and u-IGHV-specific regulatory elements. We detailed our approach in the Methods
section, as follow:

To consider potential differences related to IGHV status, we performed a differential analysis between
m-IGHV and u-IGHV CLL cases using H3K27ac and ATAC-seq peaks in 39 u-IGHV and 63 m-IGHV
samples for H3K27ac ChIP-seq data, and 38 u-IGHV and 66 m-IGHV cases for ATAC-seq data. These
differential analyses allowed us to precisely classify active/open chromatin regions in (1) significantly
higher in u-IGHV CLL, (2) significantly higher in m-IGHV CLL and (3) no significantly different between
u- and m-IGHV CLL, and therefore named "CLL". Mutations were then annotated using this
classification allowing us to identify potential enrichments in particular IGHV subgroups.

We found REs significantly mutated in u-IGHV CLLs, as presented in Extended Data Fig.6b

**(c) In the absence of ChIP-Seq data from the very same case (only 12 samples with complete RNA-seq**
**and histone marks), how did the authors assign a mutation to an open or closed region of the genome?**

As it was not possible to obtain CHIP-Seq data from paired samples, we mitigate this issue in several
ways.

Firstly, chromatin states were considered only for regions that were seen in 2 or more samples
(recurrent regions).

Secondly, using 104 ATAC-Seq and 102 H3K27ac CLL data, a differential analysis was performed
between M-CLL and U-CLL in order to define mutation status specific regulatory regions. Mutations
were then annotated using this data allowing us to identify CLL mutation status specific regulatory
regions in open chromatin.

(d) Finally, even for “overlapping” regions, the genomic coordinates of any given histone modification
peak are quite heterogeneous across different samples: did the authors redefined a “common region”
based on their overlap, or did they use the “union” of the peaks (the latter may be less robust).

Common regions based on shared overlaps were used to define these regions. This has been clarified
in the text as seen in questions 1-2-(a).

Importantly, as the regulatory elements defined here were updated compared to our original
manuscript, we re-analysed the WGS data to define an updated set of non-coding variants in active
regulatory elements. All downstream analyses using these regions have also been updated
throughout the manuscript. The updated number of significantly mutated regulatory elements is as
follow:

Mapping non-coding mutations to RE (Fig.4b, Methods), we could identify 29 UTRs, 25 enhancers (23
of them catalogued by the GeneHancer database¹⁷) and 72 promoters that had hotspot mutations or
were significantly mutated (Extended Data Fig.6a, Supplementary Table 17).

Next, we defined the candidate target genes of these 126 mutated non-coding REs. Mutations within
UTRs and promoters were annotated predominately according to proximity (Methods). For
enhancers, we calculated the correlation between H3K27ac levels for each RE and the gene
expression levels of surrounding genes located within the same topologically associated domain
(TAD) of the B-cell lymphoblastoid cell line GM12878¹⁸ (Methods). In total, 29 REs had target genes
known to be CLL drivers or cancer drivers in the COSMIC database (Fig.4c); 89 were linked to other
genes (Fig.4d) and 8 to none (Extended Data Fig.6a, Supplementary Table 17). Four mutated REs
were specific for u-IGHV (Extended Data Fig.6b), none for m-IGHV. Overall, genes targeted by
mutated REs were enriched for Gene Ontology (GO) terms linked to the immune system, lymphocyte
activation and cell death (Fig.4e, Supplementary table 18).

**1.3-It is also unclear how the identified promoter-distal regulatory elements were assigned to**
**predicted target genes. Expression is valid, but insufficient per se: mapping to the same topologically**
**associated domain, proximity, and specific cutoffs for expression level should be integrated, again**
**separately for M and UM cases.**

In the original manuscript, we combined proximity data for promoters with expression and CHIP-Seq
derived annotation data from 12 CLL samples to infer the target gene of active distal regulatory
elements (De Paepe et al. 2018). Annotation data were not paired to our samples.

In the revised manuscript, our methodology has been extended to use additional data, including larger
ATAC-seq and ChIP datasets and promoter capture Hi-C which are publicly available and derived from
CLL primary cells. This was performed separately for M and UM cases. The details of our methods have
been included in the methods section, as follow:

Target genes of regulatory elements

To link REs to their potential target genes, we first selected the regions previously classified as
promoters (including WkProm, ActProm, StrEnh1 chmm) and enhancers (including WkEnh, StrEnh2
chmm) in at least 2/7 of the previous CLL samples. Next, these regions were intersected with the
previous ChIP-seq H3K27ac data showing at least 1 peak in all the CLL samples analysed (n=104). This
intersection ensures to obtain H3K27ac peaks associated with REs. Afterwards, we calculated
correlation coefficients between H3K27ac levels of all the previously identified peaks related to RE
with the gene expression levels of all the surrounding genes. The surrounding genes were defined
according to the gene coordinates with a 2kb upstream extension with respect their start and being
in the same TAD as the interrogated RE (TADs were defined by Hi-C in GM12878¹⁸). This strategy
allowed us to identify both proximal and distal associations between regulatory elements and their
target genes. To perform the correlations, we used 74 previously available CLL samples with
concomitant H3K27ac and RNA-seq¹³. For each defined RE, we calculated the Pearson correlation
coefficients between H3K27ac levels and gene expression levels for all possible genes with the
aforementioned restrictions. Potential target genes were considered to be those showing correlations
of $r \geq 0.3$, $FDR \leq 0.05$ (FDR correction per each RE) with the RE. Correlations were only performed with
protein coding and lncRNA genes defined by Ensembl Genes version 105. Annotation was retrieved
using the biomaRt R package version 2.50. Preferentially, we have chosen the annotations from this
strategy (named CLL dataset 1).

In the case where no target gene was identified using the previous strategy, we assigned the promoter
class to a target gene based on the nearest TSS using the GENCODE database (for 7/72 promoters).
For the enhancer class, we used additional publicly available datasets derived from CLL primary
samples. This was the case of 16 out of 126 enhancers. Firstly, three-dimensional chromatin
interactions publicly available for one CLL sample was downloaded (EGA, under accession number
EGAD00001004046). We used this data to annotate genes found in the same TAD as the RE (CLL
dataset 2, used for 2 enhancers). Secondly, we used a dataset containing target genes of enhancers
predicted using correlation of RNA-seq expression data and H3K27ac data and ATAC-seq from 12
matched CLL samples previously defined¹⁶ (CLL dataset 3, used for 6 enhancers). Briefly, we required
a minimum of 10 reads in at least 1 sample for transcripts detected by RNA-seq to be considered.
Within each TAD defined using HOMER¹⁹ from Hi-C data of the GM12878 cell line [GEO accession:
GSM1181867¹⁸], we performed a peak-transcript correlation (Spearman correlation). All correlations
> 0.3 were selected. If the distance between the H3K27ac peak centre and the nearest TSS was < 2 kb
the peak was classified as a promoter, if not, it was classified as an enhancer.

Annotations for each enhancer-target are specified in Supplementary table S17. After using these
three strategies to annotate enhancers to target genes, eight enhancers did not have any predicted
target genes. These were left without CLL specific annotations and were intersected with the Double
Elite enhancers reported in the public database GeneHancer¹⁷.

In addition, all enhancers reported as significantly mutated were further annotated as super
enhancers using previously published CLL-specific super enhancers datasets obtained from chromatin
accessibility and H3K27ac ChIP-seq data using 18 CLL samples²⁰, available in Supplementary table S17.

A summary of this was also included in the results and in a figure to allow the readers to have the
context of the analysis.

Next, we defined the candidate target genes of these 126 mutated non-coding REs. Mutations within
UTRs and promoters were annotated predominately according to proximity (Methods). For enhancers,
we calculated the correlation between H3K27ac levels for each RE and the gene expression levels of
surrounding genes located within the same topologically associated domain (TAD) of the B-cell
lymphoblastoid cell line GM12878¹⁸ (Methods). In total, 29 REs had target genes known to be CLL
drivers or cancer drivers in the COSMIC database (Fig.4c); 89 were linked to other genes (Fig.4d) and
8 to none (Extended Data Fig.6a, Supplementary Table 17).

1.4-

**(a) The size of REs can span several megabases, and the presence of sparse genetic events may not**
**translate into a common functional impact: how did the authors define “recurrent” non-coding**
**mutation?**

We used three well established approaches accepted in the cancer genomics literature to uncover
these recurrently and significantly mutated REs: (1) single-site hotspots, (2) regions with high
mutational density/kataegis and (3) REs significantly mutated according to discovery algorithms.

These significantly mutated REs found by these approaches had to be recurrently mutated, i.e. the
regulatory element was mutated in several samples, although not always at the same nucleotide. The
median length of REs defined as significantly mutated was 1.09 kb.

We agree that the presence of sparse genetic events may not translate into a common functional
impact. To mitigate this, we used DeepHaem to identify variants that are the most likely to affect gene
regulation. This algorithm assesses the potential of a mutation to cause a change in gene expression
based on open chromatin data and the likelihood of causing a change in TF binding. This is detailed in
the revised methods, as follow:

We used DeepHaem⁴ to predict the effects of the non-coding mutations in the 72 promoters with
open chromatin using 73 immune cell types with open chromatin assay data. Briefly, the model looks
at 1 kb DNA sequence and calculates a probability of a DNA sequence belonging to an open chromatin
site in each cell type. We use an empirical threshold to classify anything above a 20 % probability of
being at least a weak open chromatin site. To assess the effect of each mutation, we predicted the
probability score with and without the variant in the DNA sequence and calculated the damage by
subtracting the variant probability score to the reference probability score. A positive score meant the
variant reduced the chromatin accessibility, such as in the case of a TF binding site loss (loss of
function). A negative damage score meant the variant increased chromatin accessibility, for example
by creating a novel transcription factor binding site. Empirical thresholds were set as an absolute
damage above 0.1 and 0.05 to select candidate variants with varying levels of stringency (changes
above 10 and 5%, respectively).

As reviewer 1 predicted, our experimental work highlighted that, “genetic events may not translate
into a common functional impact”. This was added into the manuscript as follow, about the enhancer
of BCL6 investigated here:

RNA-seq of these 8 samples with mutations in this region showed overall increased expression of *BCL6*,
although the effect was heterogenous (Fig.5c) and suggesting that some variants are more or less
pathogenic and might exert a positional effect (Fig.5d).

Other examples, like the *BACH2* and *ATAD1* promoters showed similar effects of different mutations.
This was already detailed in response to question 1-1.

**(b) Frequencies should be provided, rather than p values, and display items showing representative**
**“hotspot” regions at base pair resolution, with the mutations found at specific binding sites, would**
**be of help.**

Numbers of mutated samples and cohort frequencies are provided in the text and in the figures 4c-d,
supplementary figure 7a-b and Supplementary table 17.

We have now prepared additional figures showing representative “hotspot” regions at base pair
resolution, with the mutations found at specific binding sites in figures: Figure 5a (BCL2 UTR), Figure
5d (BCL6 enhancer), Figure 6d (BACH2 promoter), Supplementary figure 8b (PAX5 enhancer).

Additional general comment:
We added an introductory paragraph detailing the samples used for each method.
We performed WGS of tumour and matched normal samples from 485 patients with treatment-naïve
CLL enrolled in clinical trials to a median depth of 109X and 36X, respectively (Supplementary tables
1-3). A second tumour sample was available for a subset of 25 patients at relapse. In addition, RNA-
seq (n=73) and ATAC-seq (n=24) data were generated for informative CLL samples (Supplementary
table 4).

**2. IDENTIFICATION OF NOVEL CODING DRIVERS:**
**2.1- There is no information about the nature of the mutation in these new genes (missense?**
**Truncating? Heterozygous? Homozygous?). Since most appear to be missense events, their**
**functional significance is unclear, preventing their interpretation as pathogenic.**

We agree with the reviewer that it is difficult to infer the functional significance of novel mutations.
The coding drivers in our analysis were selected as the top most probable coding drivers according to
our strict discovery pipeline (with $q < 0.001$ for at least two cancer driver discovery algorithms) which
exclude genes with less certainty of having mutations with benign significance and that are non-
pathogenic. The validity of our approach was confirmed as the most reported CLL drivers previously
discovered were also found by our method, including known genes mostly affected by missense
events. We included a detailed explanation of our selection approach in Figure 1a.

To further address this question, we annotated the coding mutations detected by our pipeline. This
allowed us to ensure that our stringent pipeline detected robust candidate drivers. We added the
following annotations: (i) the impact of the mutation (missense / truncating), (ii) if the gene was also

recurrently affected by CNAs, (iii) if the variant was previously reported in the COSMIC database, (iv)
the cancer cell fractions (CCF) of the variants, (v) if the variants affected a protein domain. These
annotations were added in Figure 2a and Supplementary table S11.

We agree that in silico prediction of the pathogenicity of missense somatic variants needs to be
interpreted with caution. However, several well described CLL driver genes are almost exclusively
affected by missense mutations. For example, in our cohort, 62% of known drivers are affected by
missense mutations and for some of them, including TP53 (74%), SF3B1 (99%), and ATM (53%), the
majority are missense mutations. In order to address this issue, our discovery algorithms for
uncovering novel CLL drivers were conservative and assigned more weight into variants with high
functional impact, such as frameshift variants and stop-gain variants. Therefore, by refining the novel
driver list to include only the most significant genes, our pipeline preferentially selected genes affected
by high functional impact mutations and genes recurrently affected by copy number alterations. As a
result, only 32% of variants in novel driver genes were missense variants.

In order to address the reviewer's question of hetero- and homozygosity of variants, we had to
calculate the cancer cell fractions (CCF) (iv) for all variants as detailed below in response to Q2-3.

The 22 novel coding driver genes were altered by truncating mutations and also affected by CNAs
(Fig.2a, Extended Data Fig.4a-d, Supplementary table 12, Supplementary Note 1). Most mutations
occurred in protein domains, and 62% of mutations were detectable in >50% of tumour cells (median
cancer cell fraction, CCF, ≥ 0.5) and 89% in $\geq 20\%$. All novel CNAs for which we could predict a target
gene(s) were also clonal (median CCFs ≥ 0.8) (Fig. 2b, Extended Data Fig.4e). Novel candidate driver
mutations affected multiple biological pathways including RNA-ribosome processing (Fig.2c).

We added some additional details in the Methods section:

In addition to annotations from VEP, we also annotated the variant with the Prot2HG database to
determine if a variant was within a protein domain, annotated as in a "site" (short sequence with
known function) and as in a "region" (larger regions of about 100 amino acids associated with a
function of the protein; more details in the original publication²¹).

In addition, we used paired WGS and RNA-seq data of the same sample to confirm the variants in
novel coding drivers are expressed in RNA and shows when it changes gene expression level. These
new results are detailed in the text, as follow:

Performing RNA-seq on representative CLL samples from 74 patients with known and novel coding
mutations (for 40 of the 58 drivers, n variants = 118, Methods, Supplementary table 4), we validated
the expression of 73% of variants at the RNA level (Extended Data Fig.5a, Supplementary table 13). As
expected, most (29/43) mutations that were either not detectable or were seen at low expression
levels were truncating mutations consistent with nonsense-mediated decay (Supplementary table 13).
Additionally, allelic skewing and/or a reduction of mutant transcript expression compared to the mean
expression of WT transcripts across the cohort was shown, notably for specific mutations in *SPEN*,
*SETD2*, *TP53* and *IRF2BP2* (Fig. 2d). When considering all mutations, significantly reduced gene
expression was demonstrated for *TP53*, *ATM* and *SETD2*^{22,23}(Extended Data Fig.5b).

Finally, the main aim of our study was to evaluate the clinical impact of genomic aberrations, not to
define the biological function of individual gene mutations. In this section, we detailed the statistical

significance of each mutated candidate driver on the survival of patients. This analysis confirmed that
our bioinformatics pipeline was able to detect novel candidate driver mutations associated with
known risk factors and clinical outcome. We detailed these findings in the text:

When we associated the 58 drivers and regions of CNAs with other biological variables such as disease
stage, *TP53* alterations, IGHV mutation status (unmutated, u-IGHV; and hypermutated, m-IGHV) and
stereotyped B-cell receptor immunoglobulin subsets (BCR IG) including IGHV3-21 usage (Fig.2e,
Supplementary table 14, Fisher's exact test, FDR < 0.05), we found that *SETD2/del3p21.31*, *del9p21.3*
and gains of chr17q21.31 were associated with relapsed/refractory (R/R) disease and *TP53* disruption,
whereas *MED12* and *DDX3X* mutations were associated with u-IGHV CLL. BCR IG subset #2,
representing about 3% of all CLL and known to be associated with poor prognosis²⁴ was linked to the
novel driver *FAM50A*. The IGHV3-21 rearrangement was also enriched for *FAM50A* and for
*ATM/del11q22*, *SF3B1* mutations and chr21q21.3-q22.3 gains.

Restricting analysis to patients with information on long-term survival outcome (n = 243 out of 485),
13 drivers and recurrent CNAs were significantly associated with progression free survival (PFS) and
11 with overall survival (OS) (FDR<0.05) (Fig.3c, Supplementary table 15-16).

We also confirmed that the number of mutated drivers was significantly associated with PFS:
A higher number of drivers was associated with worse PFS, especially when non-coding variants were
included (Extended Data Fig.9a-b and Supplementary table 15-16).

We have now included examples of novel candidates of interest based on all additional results
supporting their role as CLL drivers in the revised manuscript. This was detailed in the text:

Twenty-one out of the 22 novel drivers were also related to disease progression (Extended data fig.5c-
f), including two of the most commonly mutated ones. *IRF2BP2* (Interferon Regulatory Factor 2 Binding
Protein 2), located in the minimally deleted region of chr1q42.3 (Fig.3d) was also affected by
deleterious mutations and CNAs (Fig.3e) (in total, n = 28, 5.8%) with high CCFs (Fig.3f left panel).
Mutations showed evidence of clonal expansion in more advanced disease (Fig.3f right panel) and
altered RNA expression (Fig.2d). This gene contributes to the differentiation of immature B-cells and
is associated with a familial form of common variable immunodeficiency disorder²⁵.

Similarly, *SMCHD1* (Structural Maintenance Of Chromosomes Flexible Hinge Domain Containing 1),
previously reported as a candidate tumour suppressor in haematopoietic cancers²⁶ was affected by
CN losses (del18p11.32-p11.31) (Fig.3g) and truncating SNVs/indels with high CCFs (Fig.3h) (n = 24,
5.0%). *SMCHD1* mutations showed clonal expansion (Fig.3i) and were associated with adverse OS
(median = 48.2 months, p-value < 1 x 10⁻⁴, log-rank test) (Fig.3j).

2.2-What are the "more permissive criteria" used for gene selection of additional hotspot genes?

As this section was unclear, we clarified our analysis by adding Figure 1a. Also, we added more details
in the result section as follow:

We also found 66 additional genes affected by recurrent CNAs using more permissive criteria
(Methods, discovery method 2). While these are potentially interesting, they were not considered to
be CLL drivers and were not taken forward for downstream analyses (Supplementary table 9).

And in the Methods section, as follow:

For discovery method 2, first we identified minimally overlapping regions altered by CNAs, defined as
the smallest genomic regions where CNAs of four samples overlap. 74 regions were defined by this
method. Second, all genes falling within these loci were analysed further to select the most probable
drivers of each region. MutComFocal v1.0²⁷ was used to rank the genes in each region, according to
the following approach:

- 1. We ranked the genes in each region, calculating focality and recurrence scores based on
SNV, indel and CNA data. The ranked genes were categorised into tiers based on the entropy
H of the posterior distribution of the scores. Genes ranked in the top 2 tiers were selected
(in our dataset, this was defined as $H(P) > 2.48 \times 10^{-4}$ for “CN gain + mutation”, and $> 1.01 \times$
10^{-4} for “CN loss + mutation”).
- 2. We refined the list of selected genes to keep only genes (a) affected by CN gains if they are
suspected or proved oncogene from the literature, or (b) affected by CN loss if they are
suspected or proved TSG from the literature.
- 3. Among these, we finally chose only genes with acquired SNVs/indels in at least 1% of the
cohorts (at least 5 samples). The genes that did not satisfy this criterion were not classified
as “candidate drivers”, but instead are listed in **Supplementary Table 9** in the “Permissive
list of genes”. We provide this gene list with combined recurrence and focality scores as
calculated by MutComFocal as a resource for the community for future research on CLL.

2.3-

(a) What is the allelic fraction of the variants identified (same question applies to the copy number
aberrations below).

Regarding small variants (SNVs/indels), the cut-off applied to allelic fraction (AF) was > 0.05 , as
detailed in the Methods section under “Variant filtering”. We did not further study any variants with
an allele fraction less than 0.05.

To add additional information of variants in coding genes and CNAs, we calculated the allele fraction
and the cancer cell fraction (CCF). The methods for this were added in the Methods section, under
“Cancer cell fraction calculation”.

We are now showing the CCF for all mutations in the 58 genes and recurrent CNAs in multiple figures.
It is also mentioned in the result section:

Most mutations occurred in protein domains, and 62% of mutations were detectable in $>50\%$ of
tumour cells (median cancer cell fraction, CCF, ≥ 0.5) and 89% in $\geq 20\%$. All novel CNAs for which we
could predict a target gene(s) were also clonal (median CCFs ≥ 0.8) (Fig. 2b, Extended Data Fig.4e).

(b) Are the authors considering subclonal events – which again would have unclear functional
significance to the pathogenesis of the disease, unless the authors can demonstrate their progressive
expansion during disease evolution.

We also explored the progressive expansion of mutation in the 58 genes and recurrent CNAs during
disease evolution. These analyses consisted in (i) comparing Frontline and R/R samples of our cohort
(unmatched samples), (ii) comparing CLL phase vs. high-grade transformation of CLL called Richter's
syndrome (RS) (previously published cohort) and (iii) performing additional WGS of samples from
patients in our initial cohort of 485 who relapsed during clinical trials. We show that numerous genes
and regions affected by CNAs were either already clonal at the time of first treatment initiation or had
an increasing CCF at the later stage.

We updated the methods section:

For a subset of 25 patients, we obtained a sample taken at relapse (**Supplementary table 4**).

To investigate findings in more advanced disease, we re-analysed WGS data coming from a cohort of
17 patients for which two concurrent samples were collected: the CLL phase and the transformed
phase (called Richter's syndrome). This cohort includes samples and data generation as described in
Klintman et al.²⁸.

And presented the findings in the result section:

Next, we examined the relationship between driver gene mutations and disease progression in three
different cohorts (**Fig.3a, Supplementary Table 4, Methods**): (i) unpaired frontline vs. R/R (main
cohort, unmatched, n = 443 vs. 30 – excluding the 12 early CLL); (ii) paired samples from the CLL and
Richter's syndrome (RS) phases of the same patient (previously published cohort, matched, n=17); (iii)
a second sample taken after relapse from patients who had already been profiled before frontline
treatment: paired frontline vs. relapsed (main cohort, matched, n=25).

There was evidence of clonal expansion with disease progression all in three cohorts including a higher
mutation frequency in the RS compared to the CLL phase ($p=2.1 \times 10^{-2}$, **Extended Data Figs.5c,d**) and
higher CCFs of mutated genes at more advanced stages for all drivers with a median CCF > 0.8 (**Fig.3b**,
**Extended Data Fig.5e-f-g**).

3. COPY NUMBER ANALYSIS:

3.1- It seems that 105 samples were analyzed by SNP genotyping and it is thus presumed that the rest
were called based on sequencing-based tools: please clarify and provide criteria for CN calls.

Clarifications were added into the methods section, as follow:

We used both SNP genotyping (n = 109 samples) and WGS (n = 485 samples) to determine CNAs and
observed high concordance between the two methods. Of 282 CNAs detected by WGS 240 (85%) were
also reported by SNP array with high confidence (**Supplementary Table S6**). In addition, we further
reduced false positive signals using a combination of intersects between several variant callers and
visual inspection as detailed below.

Samples from subset of 109 patients enrolled in ARCTIC and AdMIRe trials were genotyped using
HumanOmni2.5-8 BeadChip arrays (Illumina Inc., San Diego, CA, USA). Genotypes were called using
GenomeStudio (Illumina Inc., San Diego, CA USA). CN gains and losses > 50kb and cnLOH >5Mb were
reported using Nexus Copy Number v.10 (BioDiscovery, Inc., El Segundo, CA, USA), as previously
described^{29,30}, with the following settings (SNP Rank Segmentation): Significance Threshold 1×10^{-5} ;

Max Contiguous Probe Spacing (Kb) 1000.0; Min number of probes per segment 5; High Gain 0.6; Gain
0.2; Loss -0.2; Big Loss -1.0; 3:1 sex chromosome gain 1.2; Homozygous Frequency threshold 0.95;
Homozygous Value Threshold 0.8; Heterozygous Imbalance Threshold 0.4; Minimum LOH Length (Kb)
20; percentage outliers to remove 3%. We also inspected visually all genomes to scan for changes not
identified using these analysis settings using Nexus visualisation tool.

In the case of WGS, Canvas 1.3.1³¹ and Manta 0.28.0 were used to call CNAs, filtering out centromeric
and telomeric regions as defined in the UCSC cytoband table. Variants reported by Canvas with a
quality score < 10 were filtered out. Variants reported by Manta were filtered out as follows: (1)
variants with a normal sample depth near one or both variant break-ends three times higher than the
chromosomal mean, and (2) variants with somatic quality score < 30.

For each remaining CNA, its presence and type (gain or loss) were confirmed by visually inspecting the
genome-wide mean coverage and B-allele frequency data, derived from the aligned reads in 100 kb
windows. Calls with continuous copy number changes of length > 100 kb were kept.

We also referenced this work in the result section:
We identified 74 regions of the genome that were recurrently affected by CNAs in at least four samples
(Fig.1c, Extended Data Fig.2a, Supplementary table 6). Using SNP array data 85% of CNAs could be
validated (Supplementary table 7).

3.2-

(a) What is the basis for nominating individual genes as the targets of the aberration?

To nominate individual genes as the targets of the aberration we used previously published algorithm
MutComfocal, as detailed in the methods section and previously in question 2.2- And also in the result
section:

By combining SNVs/indels with CNAs (see Methods), we predicted the most likely target gene for nine
known regions, including *TP53/del17p13.1*, and seven novel regions including *PCM1/del8p*,
*IRF2BP2/del1q42.2q42.3* and *SMCHD1/del18p11.32-p11.31* (Fig.1d, Extended Data Fig.2b-c,
Supplementary table 8). We also found 66 additional genes affected by recurrent CNAs using more
permissive criteria (Methods, discovery method 2). While these are potentially interesting, they were
not considered to be CLL drivers and were not taken forward for downstream analyses
(Supplementary table 9).

(b) Were these in focal CN losses/gains?

The regions were defined as the minimally affected region when intersecting all CNAs in the cohort,
as described in the methods section (question 2.2-) and figure 1a - discovery method 2.

(c) Were these clonal or subclonal?

The regions were primarily clonal, as stated in the text and presented in figure 2b:
All novel CNAs for which we could predict a target gene(s) were also clonal (median CCFs ≥ 0.8) (Fig.
2b, Extended Data Fig.4e).

(d) If these alterations are heterozygous, is expression of the residual allele lost? Or do the authors
imply that these new targets of CN losses are haploinsufficient tumor suppressors?

(d) We investigated the change of expression in novel genes affected by both CNAs and mutations
*DLC1*, *IRF2BP2*, *LUC7L2*, *PCM1* and *ZFPM1*. However, the small sample size didn't allow us to show any
RNA expression differences (data not shown). Change of expression was found for *TP53*, *ATM* and
*SETD2* for samples with mutation and/or CN loss. These results are presented in Extended data figure
5b.

4. MUTATIONAL SIGNATURES: -The proposed use of SBS9 as a surrogate marker to assess IGHV
mutated and unmutated CLL is interesting but not convincing (please provide cosine similarity for all
analyses). Indeed, the % of CLL-M samples correctly identified using this signature was >80%
(supplementary methods 2). Moreover, the SBS9 signature has been associated with both AID and
pol-eta. This proposed approach would probably require a manuscript on its own.

We agree with the reviewer that the focus of this manuscript cannot be to provide sufficient evidence
that SBS9 works as a permanent replacement for assessing IGHV mutation status. We therefore
defined the IGHV mutational status Sanger sequencing or IgCaller (Nadeu et al., Nature
Communications 2020). In total, Sanger sequencing derived mutation status was obtained for 264
(54%) of our data. To define as many as possible of the remaining samples and also to derive the
stereotyping, we use the software IgCaller. This software uses WGS read level data to provide more
granularity about the rearrangements by generating a complete immunogenomic picture of each
sample by performing local alignment and assembly to Ig loci. This method has been demonstrated
to be a reliable alternative to Sanger sequencing for determining Ig rearrangements and assessing Ig loci.
In 27 cases (6%) without Sanger or IgCaller information we have used SBS9. Thus, mutation status was
defined hierarchically, where the Sanger derived status > IgCaller > SBS9.

A table of comparative results between the Sanger derived mutation status, the IgCaller results, and
the SBS9 results has been provided in the supplementary material. These data will be a useful resource
to the CLL research community.

These details were included in the methods section:

Immunoglobulin gene characterization

To determine the IGHV status of our cohort, we prioritized data from Sanger sequencing, followed by
WGS-derived data including IgCaller³² results and the presence of non-canonical AID mutational
signature (SBS9). This prioritizing scheme resulted in 54% (264/485) cases classified by Sanger
sequencing, 40% (194/485) by the IgCaller algorithm and 6% (27/485) by the mutational signature
SBS9. The correlation between these 3 methodologies was high and can be found at Supplementary
Table 26. In addition, the IgCaller algorithm was used to further characterize the IG genes, including
to define the IGHV3-21 rearrangement in 10% (47/485) of cases and CLL stereotypy in 27% (132/485).
To assign CLL stereotypes, the IgCaller output was used as input for AssignSubsets online tool³³, which
annotates the 19 major subsets, including subsets #1, #2, #4 and #8, as recommended by ERIC
guidelines³⁴. In cases more than one rearrangement were detected, we selected the rearrangement
with the highest score to define the main CLL stereotype. In cases where a rearrangement was not

assigned, but there was a proximal rearrangement reported, we included this rearrangement in our
analysis.

As part of the mutational signatures analysis, cosine similarity was derived for all signatures and has
been provided in the Supplementary Table 20.

5. DATA ORGANIZATION:

5.1- Key information and conceptual criteria allowing the reader to understand how the analyses were
performed and how conclusions were made should be provided in the main body of the manuscript.

Additional key information and conceptual criteria about coding driver detection, non-coding
regulatory element annotations and copy-number detection as detailed in questions 1 to 4, were
added into the main body of the manuscript and main figures:

Fig1a, Fig2e, Fig4a-b, Fig6b.

5.2- Some of the figures are impossible to read (eg Figure2, Figure 4, extended Figure 7)

High resolution figures are provided at resubmission.

5.3- Figure 1: there is no labeling to the y axis of Figure 1b

Labels were edited

**Reviewer #2**

Remarks to the Author:

This manuscript describes a comprehensive whole genome analysis, encompassing both coding and
non-coding variants, and genome-wide genomic lesions / CNAs in a cohort of 485 chronic lymphocytic
leukemia (CLL) patients. Several novel coding driver genes were identified, as well as non-coding
variants through the use of known regulatory elements in CLL. These non-coding variants partly
occurred in genes and pathways that have been known to be affected by coding variants. Multi-layer
genomic data integration (coding, non-coding and genome-wide lesions) allowed to distinguish CLL
into genomic subgroups. Comparison to clinical data in a sub-cohort of the CLL patients resulted in
associations between these genomic subgroups and outcome.

Building on existing genomic data sets in CLL that mostly concerned either coding variants, or -to some
extent- non-coding variants, the clear added value of this study is the integration of coding and non-
coding SNVs / indels plus genomic lesions, which allows a much more comprehensive evaluation of
the entire mutational landscape in the largest CLL cohort evaluated in this way so far. Another plus of
this study is that for interpreting non-coding variants the authors have made use of ATAC-seq and
CHIP-seq data pointing to relevant regulatory elements (promoters, enhancers, UTR) to interpret non-
coding variants. Overall, this is an interesting paper, but I have several major comments, as listed
below.

1.1- A major comment is that the whole study is very oncogenetically oriented, whilst it is obvious
from numerous papers over a period of 20 years or so, that immunogenetics is another important
factor in CLL. Admitted, the authors divide CLL into mutated and unmutated CLL, but immunogenetics
meanwhile goes far beyond this. Bifurcation into M-CLL and U-CLL is too simple. It is well known that
there is an intermediate group, with borderline mutation status and a different epigenetic profile, and

also particular stereotypic CLL subgroups are now well-established. In this respect this reviewer finds
it crucial to make further comparisons with immunogenetically defined to better understand the
biology of the here introduced CLL subgroups.

We added an immunogenetics angle as suggested by reviewer 2. To further describe the
immunogenetics picture of the patients in this cohort, we have used the IgCaller (detailed in Q4
addressing a question of Reviewer 1). It allowed us to comprehensively determine the
immunogenomic profile of the patients. In doing so, we were able to describe the enrichment of
genomic features in specific groups such IGLV3-21 rearrangement, and specific stereotypic subsets,
where possible.

Immunoglobulin gene characterization

To determine the IGHV status of our cohort, we prioritized data from Sanger sequencing, followed by
WGS-derived data including IgCaller³² results and the presence of non-canonical AID mutational
signature (SBS9). This prioritizing scheme resulted in 54% (264/485) cases classified by Sanger
sequencing, 40% (194/485) by the IgCaller algorithm and 6% (27/485) by the mutational signature
SBS9. The correlation between these 3 methodologies was high and can be found at Supplementary
Table 26. In addition, the IgCaller algorithm was used to further characterize the IG genes, including
to define the IGHV3-21 rearrangement in 10% (47/485) of cases and CLL stereotypy in 27% (132/485).
To assign CLL stereotypes, the IgCaller output was used as input for AssignSubsets online tool³³, which
annotates the 19 major subsets, including subsets #1, #2, #4 and #8, as recommended by ERIC
guidelines³⁴. In cases more than one rearrangement were detected, we selected the rearrangement
with the highest score to define the main CLL stereotype. In cases where a rearrangement was not
assigned, but there was a proximal rearrangement reported, we included this rearrangement in our
analysis.

The results were as follow:

When we associated the 58 drivers and regions of CNAs with other biological variables such as disease
stage, *TP53* alterations, IGHV mutation status (unmutated, u-IGHV; and hypermutated, m-IGHV) and
stereotyped B-cell receptor immunoglobulin subsets (BCR IG) including IGHV3-21 usage (**Fig.2e**,
**Supplementary table 14**, Fisher's exact test, FDR < 0.05), we found that *SETD2*/del3p21.31, del9p21.3
and gains of chr17q21.31 were associated with relapsed/refractory (R/R) disease and *TP53* disruption,
whereas *MED12* and *DDX3X* mutations were associated with u-IGHV CLL. BCR IG subset #2,
representing about 3% of all CLL and known to be associated with poor prognosis²⁴ was linked to the
novel driver *FAM50A*. The IGHV3-21 rearrangement was also enriched for *FAM50A* and for
*ATM*/del11q22, *SF3B1* mutations and chr21q21.3-q22.3 gains.

1. 2-

(a) This especially holds for the big division into M-CLL and U-CLL in comprehensively analyzing
genomic data, as in lines 219 and 220, where esp. a borderline category could be introduced. And it
also holds in a more specific way for the subgroups as summarized in lines 258-261. Is m-GS2 perhaps
equivalent to stereotypic subsets #4 or satellites?

For our clustering analysis, we divided the cohort into 2 groups: m-IGHV and u-IGHV. In absence of
cohorts of thousands of patients, it is not feasible to further divide the cohort before performing the

NMF clustering as numbers in each sub-group would be too small to derive robust signatures that are
not overfitted.

However, to add more granularity into our analysis, we have further defined the genomic subgroups
in light of the stereotypic subsets and IGHV3-21/IGLV3-21 rearrangements, as detailed in the result
section:

[About u-IGHV CLL groups:] All three subgroups included patients with BCR IG subset #1 and #8, which
are known to be associated with aggressive disease³⁴ (Extended Data Fig.14c).

Regarding m-IGHV CLL (Fig.8d), m-GS1 was similar to u-GS1 (cosine similarity of 0.81) and also to u-
GS2 (cosine similarity of 0.7) (Supplementary table 24). In contrast, m-GS1 was enriched for older men,
BCR IG subset #2 (FDR = 2.96×10^{-6}) and IGHV-3-21 (FDR = 7.50×10^{-9}) (Extended Data Fig.14c),
although most patients in m-GS1 did not have any defined CLL stereotype.

(b) And does the GC 7/8 subgroups harbor poor stereotypic subsets such as #1, #2, and #8?

We also compared the stereotypic subsets and IGHV3-21/IGLV3-21 rearrangements with the eight
genomic complexity groups we have defined. No enrichment was found and this is detailed in the
revised results.

None of the GC groups was significantly enriched in stereotyped subsets (Extended Data Fig.12g).

2. In comparing outcome to subgroups, 44 genomic features were found to predict PFS and/or OS.
These are all univariate analyses, but how many did survive multivariate analysis. Could the ones that
remain after multivariate analyses perhaps be the real relevant ones?

We selected these genomic features following a widely accepted type of multivariate analysis known
as Regularised Cox Regression or least absolute shrinkage and selection operator (LASSO) Cox
Regression. This is a form of multivariate survival analysis, which selects a minimal set of predictors
(among a larger set of candidates used as input) with maximal predictive power with respect to PFS
or OS observations in a given cohort of patients. From a computational perspective, this selection is
achieved through a statistical model fitting procedure that utilises cross-validation to minimise the
out-of-sample prediction error. Software used and the relevant reference are given in the methods
section of the main text.

We also performed a multivariate analysis using penalised Cox regression, as implemented in the R
package glmnet³⁵, to find a minimal set of predictors with maximal predictive power. An optimal value
of the penalisation parameter λ was selected using leave-one-out cross-validation; specifically, the
value of λ that minimises the cross-validation error.

The findings were as follow:

To understand the potential clinical relevance of the different modes of local or global genomic
aberrations, we first used penalised multivariate regression analysis for least absolute shrinkage and
selection operator. This analysis led to the identification of 56 individual genomic features that
predicted PFS and/or OS including *SMCHD1*/del18p11.32-p11.31 which retained significance as an
independent predictor of OS (Extended Data Fig.14a-b).

3. Are the identified coding mutations in known drivers in the same hotspots or are there new
hotspots. Also, with respect to new drivers, are the found mutations clustered? All this information is
very useful for further future analysis in new patient samples and should be provided somewhere.

We added the following annotations: (i) the impact of the mutation (missense / truncating), (ii) if the
gene was also recurrently affected by CNAs, (iii) if the variant was previously reported in the COSMIC
database, (iv) we inferred the cancer cell fractions (CCF), (v) if the variants were affected a protein
domain. These annotations were added in Figure 2a and Supplementary table 11. All details are
mentioned in section 2.1- addressed to reviewer 1.

In addition, to the full table listing all the mutations, we have added a supplementary note showing
the position of each mutation for the 58 drivers in respect of exons and protein domains
(SupplementaryNote1), as we agree with the reviewer that this is very useful.

4. With respect to mutations in UTRs, there are currently only associations based on identified RE. But
is there any proven effect on expression levels. For me it would be mandatory to check this based on
RNA seq data or qPCR experiments to make statements that these variants are truly pathogenic?

In this revised manuscript, we have investigated the effect of selected representative non-coding
mutations on gene expression. These mutations were not only in UTRs, but also in promoters and
enhancers. To do so we generated ATAC-seq and RNA-seq of the samples carrying the non-coding
mutations.

We detailed the CCFs of the non-coding mutations and investigated some selected mutations by
transcriptomics analysis.

Of the 29 mutated UTRs, 58% (n=17) had a median CCF ≥ 0.5 , and 83% had a CCF > 0.2 , thus indicating
their selection during CLL pathogenesis (Extended Data Fig.6c).

As detailed in question 1-1 in the document addressed to reviewer 1, we confirmed that mutations in
5'UTR of BCL2 were associated with BCL2 overexpression.

5. In line 62 I would like to see split out inversions in immunoglobulin light chain genes into kappa and
lambda, and it should also be added what partners are seen in those inversions.

The details of the inversions detected in the dataset, especially those in immunoglobulin light chain
genes into kappa and lambda and their partners were detailed further in the result section:

A major attribute of WGS is the power to identify inversions and translocations. We identified 1,248
inversions (Methods, Extended Data Fig.3a) with frequent breakpoints involving either the
Immunoglobulin light chain Kappa Locus (IGK) (n = 65, 13.4%), the Immunoglobulin Heavy Chain Locus
(IGH) (n = 65, 13.4%) or chr13q14.2 (n = 40, 8.7%) (Extended Data Fig.3b, Supplementary table 10-
11).

6. In Figure 3 genomic complexity leads to subgrouping into 8 group. Group 5 seems to be lacking in
the survival analysis in Extended data Figure 15, or is this due to the small group size? The rationale to
further lump GC groups is unclear, is this purely empirical, is it eyeballing, or is there any idea behind
it?

We defined 8 sub-groups by studying genomic complexity based on the presence/absence of (1) CN
loss(es), (2) CN gain(s) and (3) trisomy/ies.

For subsequent analyses, as sample size did not allow reliable estimation of the variance ($n < 5$), which
is required for robust statistical analysis of survival, group 5 was excluded from Extended data Figure
15a-b.

In order to ensure sufficiently large sample size for subsequent analysis (Extended data Figure 15c-f
and Figure 4a-b), the 8 groups were combined based on the two variables that allowed the strongest
clustering, namely the presence/absence of (1) CN loss(es), (2) CN gain(s). The four groups then
remaining were therefore made irrespectively of the presence of trisomy/ies:

- • groups 1 and 3 combined by the absence of CN Gains and losses
- • groups 2 and 5 combined by the presence of CN Gains only,
- • groups 4 and 6 combined by the presence of CN losses only,
- • groups 7 and 8 combined by the presence of both CN gains and losses.

These points were included in the revised version of the manuscript:

For the subset of samples with survival data ($n = 243$), we combined GC groups with CN gains only, CN
losses only and both CN gains and losses to increase statistical power. Interestingly, the 8 groups were
associated with different PFS and OS (**Extended Data Fig.13a-b**), independent of *TP53* status
(**Extended Data Fig.13c-d**). Furthermore, patients with both *TP53* mutations and GC#7/8 changes had
ultra-high-risk disease (median PFS = 8 months, median OS = 15 months) and fared worse compared
to patients with *TP53* mutations but no GC#7/8 ($P = 0.03$, **Fig.8a-b**).

The figure legend is as follow:

**Extended Data Fig. 14 : Integration of all genomic features defines genomic subgroups associated with**
**clinical outcome.**

a-b, Penalised Cox Regression performed on 186 genomic features identified independent predictors
for (a) progression-free survival and (b) overall-survival. Each panel shows the hazard ratios for those
genomic features that jointly minimise the out-of-sample prediction error (estimated through leave-
one-out cross-validation). The full list and details of each genomic feature is presented in
Supplementary Table 21. c, Proportion of each subgroup defined by NMF with stereotype clusters. d,
mclust classification of the proportion of each hypermutated genomic subgroup (m-GS1, m-GS2)
calculated in each hypermutated sample using deconstructSigs. Sample proportions were logit
transformed before clustering and the appropriate number of clusters approximated by examining
mclust BIC plots. e, ROC curve of m-GS2 derived from coding sequence data against m-GS2 derived
from whole genome sequencing data.

7. In Figure 4e the PFS curve comes with a clear p value for the different u-GS groups, yet in line 234
this is referred to as similar PFS. What is correct here?

The p-value in the figure refers to the difference between the most extreme curves, with all other
pairwise curve comparisons being non-significant. After removal of samples with *TP53/del17p*
alterations, factors already known to significantly reduce time to progression, the remaining
difference between the curves is not significant.

**Minor comments**

-Readability of many Figures is poor; the information is dense and fonts of labels, axes, tables are often
too small to read or grasp the message.

-Figure 2c: label is included for “coding” variants, but the whole point of Figure 2 is the “non-coding”
variants, so in my opinion it should be left out.

The figure was adapted to clarify the type of target genes (CLL drivers, COSMIC genes, other genes)

-Figure 2b: the legend is entirely unclear as to what is exactly shown in the figure.

This figure was omitted from the revised manuscript as more appropriate analyses were performed
instead.

-Figure 3f: the DBS seems to be gone; is this grouped with SBS or simply left out?

The figure legend was updated to clarify this point.

**f, Fraction of each mutational signature detected in each coding driver. DBS not shown as data was**
**too scarce.**

-Figure 4: the number of u-GS and m-GS groups is not consistent between main text (lines 221-222),
graph and legend. Please clarify whether it should read as 2 or 3 groups in both m-GS and u-GS.

It was clarified in the text.

-Extended data Figure 2c: lacks a color coding, either in the graph or in the legend.

This figure was removed and replaced by similar ones in Supplementary note 1. It now also contains
figure legends.

-Extended data Figure 5a: the bottom panel is a complete repeat of what is in Figure 2c, so can be left
out here.

The figure was removed from the revised manuscript.

- 1. Roberts, A. W. *et al.* Targeting BCL2 with Venetoclax in Relapsed Chronic Lymphocytic
Leukemia. *N. Engl. J. Med.* **374**, 311–322 (2016).
- 2. Ramachandreddy, H. *et al.* BCL6 promoter interacts with far upstream sequences with
greatly enhanced activating histone modifications in germinal center B cells. *Proc. Natl. Acad.*
*Sci.* **107**, 11930–11935 (2010).
- 3. Bunting, K. L. *et al.* Multi-tiered Reorganization of the Genome during B Cell Affinity
Maturation Anchored by a Germinal Center-Specific Locus Control Region. *Immunity* **45**, 497–
512 (2016).
- 4. Schwessinger, R. *et al.* DeepC: predicting 3D genome folding using megabase-scale transfer
learning. *Nat. Methods* **17**, 1118–1124 (2020).
- 5. Ciardullo, C. *et al.* Low BACH2 Expression Predicts Adverse Outcome in Chronic Lymphocytic
Leukaemia. *Cancers (Basel)*. **14**, 23 (2021).
- 6. Danecek, P. *et al.* Twelve years of SAMtools and BCftools. *Gigascience* **10**, (2021).
- 7. Buenrostro, J. D., Giresi, P. G., Zaba, L. C., Chang, H. Y. & Greenleaf, W. J. Transposition of
native chromatin for fast and sensitive epigenomic profiling of open chromatin, DNA-binding
proteins and nucleosome position. *Nat. Methods* **10**, 1213–8 (2013).
- 8. Smith, J. P. *et al.* PEPATAC: an optimized pipeline for ATAC-seq data analysis with serial
alignments. *NAR Genomics Bioinforma.* **3**, (2021).
- 9. Sanson, K. R. *et al.* Optimized libraries for CRISPR-Cas9 genetic screens with multiple
modalities. *Nat. Commun.* **9**, 5416 (2018).
- 10. Doench, J. G. *et al.* Optimized sgRNA design to maximize activity and minimize off-target
effects of CRISPR-Cas9. *Nat. Biotechnol.* **34**, 184–191 (2016).
- 11. Forrest, A. R. R. *et al.* A promoter-level mammalian expression atlas. *Nature* **507**, 462–470
(2014).
- 12. Severin, J. *et al.* Interactive visualization and analysis of large-scale sequencing datasets using

ZENBU. *Nat. Biotechnol.* **32**, 217–219 (2014).

13. Beekman, R. *et al.* The reference epigenome and regulatory chromatin landscape of chronic
lymphocytic leukemia. *Nature Medicine* **24**, 868–880 (2018).

14. Ernst, J. & Kellis, M. Chromatin-state discovery and genome annotation with ChromHMM.
*Nat. Protoc.* **12**, 2478–2492 (2017).

15. Rheinbay, E. *et al.* Analyses of non-coding somatic drivers in 2,658 cancer whole genomes.
*Nature* **578**, 102–111 (2020).

16. De Paepe, A. Elucidating regulatory elements : studies in chronic lymphocytic leukemia and
multiple myeloma. (Karolinska Institutet, 2018).

17. Fishilevich, S. *et al.* GeneHancer: genome-wide integration of enhancers and target genes in
GeneCards. *Database* **2017**, (2017).

18. Rao, S. S. P. *et al.* A 3D Map of the Human Genome at Kilobase Resolution Reveals Principles
of Chromatin Looping. *Cell* **159**, 1665–1680 (2014).

19. Duttke, S. H., Chang, M. W., Heinz, S. & Benner, C. Identification and dynamic quantification
of regulatory elements using total RNA. *Genome Res.* **29**, 1836–1846 (2019).

20. Ott, C. J. *et al.* Enhancer Architecture and Essential Core Regulatory Circuitry of Chronic
Lymphocytic Leukemia. *Cancer Cell* **34**, 982–995.e7 (2018).

21. Stanek, D. *et al.* Prot2HG: a database of protein domains mapped to the human genome.
*Database* **2020**, (2020).

22. Parker, H. *et al.* Genomic disruption of the histone methyltransferase SETD2 in chronic
lymphocytic leukaemia. *Leukemia* **30**, 2179–2186 (2016).

23. Austen, B. *et al.* Mutations in the ATM gene lead to impaired overall and treatment-free
survival that is independent of IGVH mutation status in patients with B-CLL. *Blood* **106**, 3175–
3182 (2005).

24. Sonia Jaramillo *et al.* Prognostic impact of prevalent chronic lymphocytic leukemia
stereotyped subsets: analysis within prospective clinical trials of the German CLL Study Group
(GCLLSG). *Haematologica* **105**, 2598–2607 (2019).

25. Keller, M. D. *et al.* Mutation in IRF2BP2 is responsible for a familial form of common variable
immunodeficiency disorder. *J. Allergy Clin. Immunol.* **138**, 544–550.e4 (2016).

26. Brideau, N. J. *et al.* Independent Mechanisms Target SMCHD1 to Trimethylated Histone H3
Lysine 9-Modified Chromatin and the Inactive X Chromosome. *Mol. Cell. Biol.* **35**, 4053–4068
(2015).

27. Trifonov, V., Pasqualucci, L., Favera, R. D. & Rabadan, R. MutComFocal: an integrative
approach to identifying recurrent and focal genomic alterations in tumor samples. *BMC Syst.*
*Biol.* **7**, 25 (2013).

28. Klintman, J. *et al.* Genomic and transcriptomic correlates of Richter transformation in chronic
lymphocytic leukemia. *Blood* **137**, 2800–2816 (2021).

29. Knight, S. J. L. *et al.* Quantification of subclonal distributions of recurrent genomic aberrations
in paired pre-treatment and relapse samples from patients with B-cell chronic lymphocytic
leukemia. *Leukemia* **26**, 1564–1575 (2012).

30. Klintman, J. *et al.* Clinical-grade validation of whole genome sequencing reveals robust
detection of low-frequency variants and copy number alterations in CLL. *Br. J. Haematol.* **182**,
412–417 (2018).

31. Roller, E., Ivakhno, S., Lee, S., Royce, T. & Tanner, S. Canvas: versatile and scalable detection
of copy number variants. *Bioinformatics* **32**, 2375–2377 (2016).

32. Nadeu, F. *et al.* IgCaller for reconstructing immunoglobulin gene rearrangements and
oncogenic translocations from whole-genome sequencing in lymphoid neoplasms. *Nat.*
*Commun.* **11**, 3390 (2020).

33. Bystry, V. *et al.* ARResT/AssignSubsets: a novel application for robust subclassification of
chronic lymphocytic leukemia based on B cell receptor IG stereotypy. *Bioinformatics* **btv456**
(2015). doi:10.1093/bioinformatics/btv456

- 34. Rosenquist, R. *et al.* Immunoglobulin gene sequence analysis in chronic lymphocytic
leukemia: updated ERIC recommendations. *Leukemia* **31**, 1477–1481 (2017).
35. Simon, N., Friedman, J., Hastie, T. & Tibshirani, R. Regularization Paths for Cox’s Proportional
Hazards Model via Coordinate Descent. *J. Stat. Softw.* **39**, (2011).

Decision Letter, first revision:

25th May 2022

Dear Professor Schuh,

I'm sorry it's taken so long to return this decision to you. Thank you so much for your patience.

Your Article, "Whole genome landscape of chronic lymphocytic leukaemia and its association with clinical outcome" has now been seen by your 2 referees. You will see from their comments below that while they find your work of interest, some important points are raised. We are interested in the possibility of publishing your study in Nature Genetics, but would like to consider your response to these concerns in the form of a revised manuscript before we make a final decision on publication.

To guide the scope of the revisions, the editors discuss the referee reports in detail within the team, including with the chief editor, with a view to identifying key priorities that should be addressed in revision and sometimes overruling referee requests that are deemed beyond the scope of the current study. While we agree that more thorough functional follow-up would have fortified the study, we are comfortable moving forward without it, given the value of the resource. However, we think that some of Reviewer #1's other points are pertinent and should be addressed. We agree that in the absence of functional validation and / or a robust analysis to show positive selection, we think that any mention of 'driver genes' or 'drivers' must be tempered with 'putative' or 'potential' or similar. While we absolutely don't want you to unnecessarily undermine your work, it would be helpful to position the work as a springboard for further downstream analyses, with the recognition that your putative drivers, together with the clinical utility of the classifier will need to be robustly tested in future studies.

Reviewer #1 has also pointed out that the functional work performed in non-CLL cell lines is of limited value to the paper. Having discussed this as a team, we would suggest that these data are removed from the manuscript.

Finally, at this point, we'd like to have some clarity about data availability. Please note that we would expect all genomic data to be made freely available if we were to proceed with publication.

Please do not hesitate to get in touch if you would like to discuss these issues further.

We therefore invite you to revise your manuscript taking into account all reviewer and editor comments. Please highlight all changes in the manuscript text file. At this stage we will need you to upload a copy of the manuscript in MS Word .docx or similar editable format.

We are committed to providing a fair and constructive peer-review process. Do not hesitate to contact us if there are specific requests from the reviewers that you believe are technically impossible or

7unlikely to yield a meaningful outcome.

*2) If you have not done so already please begin to revise your manuscript so that it conforms to our Article format instructions, available

[here](http://www.nature.com/ng/authors/article_types/index.html).

*3) Include a revised version of any required Reporting Summary:

[REDACTED]

We hope to receive your revised manuscript within four to eight weeks. If you cannot send it within this time, please let us know.

Sincerely,

Safia Danovi
Editor
Nature Genetics

Reviewers' Comments:

Reviewer #1:

Remarks to the Author:

The revised version of the paper by Robbe et al. contains a significant amount of additional data and information. In general, the paper provides a better documentation of the results. Nonetheless, the revision has failed to conclusively address the two major shortcomings of the original version, i.e. the unclear functional significance (see below) of most of coding and especially non-coding mutations observed in their large CLL panel. Thus, the claim that the paper reports "22 novel coding drivers and 126 candidate non-coding drivers" remains largely not justified by the data presented. A more conservative statement would state that the paper reports an expanded catalogue of genetic mutations of potential functional significance. This catalogue undoubtedly represents a resource for future studies, but its conceptual value is limited at the present time. Considering also that a presumably large fraction of the observed mutation originates from a hypermutator phenotype, an equally large fraction may represent innocent bystanders, thus making the "clinical impact" premature.

Major concerns:

1. The results confirm most known coding drivers of CLL and, thanks to WGS, provide mostly novel data on the inversions (1248) and translocations (933). However, only very few (2 translocations) have meaningful recurrence and none is characterized for functional significance. This concern is greater for missense mutations, which most likely provide a significant amount of background "noise" in the clinical classification.

2. Although prudently defined as "candidates", the non-coding drivers remain of unexplored significance. Unfortunately, the revised version does not correctly address the concerns stated during the original review. The authors have provided very preliminary data showing quite weak correlations between mutated regulatory regions (RE) and overexpression of the candidate linked genes (BCL2 and BCL6) (Fig. 5). They also show that deletion of parts of the REs (BCL6) leads to some reduction in target gene expression (35%) and cell proliferation (40%) in the Raji cell line, a Burkitt Lymphoma (BL) cell line. This result has little significance for CLL since BL is biologically quite distinct from CLL

9and RE regions are differentially activated in distinct phenotypes. Most notably, these results address the question of RE dependency, but provide no information of the significance of mutated REs or, better, which/whether mutations matter.

3. Given the above-described uncharacterized nature of the observed mutation, their “bulk” usage for a clinical classifier appears premature and cannot lead to a clinically- actionable classifier vis a vis existing approaches based on solid drivers.

Reviewer #2:

Remarks to the Author:

The authors have done a great effort in addressing my concerns and suggestions, which has led to clear clarifications, additional associations and improvement of data presentation.

I have no further comments.

Author Rebuttal, first revision:

We thank the Editor, the Editorial Board and the reviewers for their extremely useful and balanced comments. Below are our answers point by point. Throughout our documents, sentences that were deleted are formatted with “strike-through” and those newly included are written in blue. In addition, we have provided two Word documents: one containing these edits as visible and one with these edits as integrated in the rest of the text.

Comments from Editorial board

Comment 1 - While we agree that more thorough functional follow-up would have fortified the study, we are comfortable moving forward without it, given the value of the resource.

We thank the editorial board for recognising the value of our analyses so far. However, we think that some of Reviewer #1's other points are pertinent and should be addressed. We agree that in the absence of functional validation and / or a robust analysis to show positive selection, we think that any mention of 'driver genes' or 'drivers' must be tempered with 'putative' or 'potential' or similar.

We have clarified throughout the text when we talk about known drivers as opposed to the novel, putative/potential/candidate drivers.

We have removed the strong statement referring to the discovery of novel drivers from the abstract and replaced it with the wording as suggested by reviewer 1:

Line 48: ~~“We identify 22 novel coding drivers and 126 candidate non-coding drivers and~~ **We identify an extended catalogue of recurrent coding and non-coding genetic mutations that represents a source for future studies and show in a subset that these associate with chromatin accessibility and change in gene expression.** “

To emphasise the putative nature of the novel candidate coding drivers, we have added the following paragraph into the discussion:

Line 377: **“Based on a strict pipeline for discovery of coding drivers, we selected the top ranked recurrently mutated genes which comprised 36 known CLL drivers^{3,6,7}, and 22 genes as novel putative drivers. Only 32% of variants in those novel putative driver genes were missense variants with the majority being truncating and stop-gain mutations. Although these putative drivers shared characteristics of known drivers (i.e. damaging mutations in protein domains, impact on RNA expression, high CCF that further increased at disease progression, association with survival), we cannot exclude the possibility that some may simply represent passengers.”**

For novel putative non-coding drivers, we changed the heading of the results to:

Line 180: **“Non-coding putative driver mutations”.**

We also added a conclusion paragraph at the end of the results:

Line 254: “Collectively, these data suggest that a small subset of the non-coding mutations in CLL have characteristics indicative of a driver and target REs of genes that are critical for B-cell development and function as well as cancer progression. However, the effects on chromatin accessibility and gene expression levels were subtle and require in-depth functional characterization.”

Comment 2 - While we absolutely don't want you to unnecessarily undermine your work, it would be helpful to position the work as a springboard for further downstream analyses, with the recognition that your putative drivers, together with the clinical utility of the classifier will need to be robustly tested in future studies.

We have replaced the clinical conclusion in the abstract by the following:

Line 56: “Our results highlight the utility of whole genome sequencing for risk stratification in CLL. While requiring independent validation, our findings highlight the potential of WGS to inform future risk stratification in CLL.”

We have also added a sentence at the end of the introduction:

Line 77: “Finally, we integrate the different modes of genetic alterations to define five genomic subgroups of CLL and relate these to clinical outcome. with clinical impact, which provide better estimation of patient prognosis than achieved by previous single gene analyses. Our results act as a springboard to in-depth functional validation of putative drivers and raise the prospect that after independent clinical validation, this integrated genome-wide approach could refine current clinical outcome prediction.”

We have changed the heading of the result section to:

Line 313: “~~Development of~~ **Towards a new patient classifier**” to clarify that we do not propose the immediate usage of our model in clinical practice.

We have added a sentence to draw the reader’s attention to the NMF results presented in extended data figure 15 after limiting the model to known coding drivers and the four established recurrent CNAs:

Line 351: “In our analysis of patients treated with chemoimmunotherapy, NMF subgroups could not be defined without the different acquired local and global non-coding genomic changes, since combining all known coding drivers and the 4 common recurrent CNAs did not cluster patients into the genomic subgroups. thereby demonstrating the benefit of integrating different types of genomic information to precisely stratify all patients and the limitations of relying on specific single gene mutations (Extended Data Fig.14e, Extended Data Fig.15).”

In response to the reviewer’s concerns of using putative coding and non-coding drivers in the NMF analysis to define a clinical classifier for response prediction, we have added a discussion point:

Line 401: “Ideally, only genomic features experimentally validated as disease drivers should be included in any prognostic classification system, even if they were selected by very stringent criteria as those applied in this study (see above). However, it is well recognized that some genomic features are clearly not disease drivers, yet carry prognostic relevance. For example, in CLL, the IGHV mutation status representing the cell-of-origin or telomer length reflecting proliferative activity, are strongly associated with clinical outcome, but are not considered disease drivers.

In our NMF model using only the known coding drivers and recurrent CNAs did not allow us to recover the same level of discrimination as that afforded by inclusion of additional local and global non-coding information. This observation implies that the combination of coding and non-coding information in the classifier increases the precision of clinical risk prediction at least in our cohort of clinical trial patients.”

Finally, we have toned down the conclusion of the discussion to:

Line 417: “Collectively, our study provides a spring board for downstream functional analyses of novel putative coding and non-coding drivers. Further robust testing on independent cohorts of patients undergoing targeted therapy will be required to establish the clinical utility of this new WGS-based classifier further.”

Comment 3 - Reviewer #1 has also pointed out that the functional work performed in non-CLL cell lines is of limited value to the paper. Having discussed this as a team, we would suggest that these data are removed from the manuscript.

We have removed these data from the manuscript.

Comment 4 - Finally, at this point, we'd like to have some clarity about data availability. Please note that we would expect all genomic data to be made freely available if we were to proceed with publication.

We have now included the following mention in the manuscript:

Line 923: “Data from the National Genomics Research Library, which are held in a secure Research Environment, are available freely to registered users. Please see <https://www.genomicsengland.co.uk/research/academic> for more information.”

Comments from Reviewer #1

General comments:

(1) The revised version of the paper by Robbe et al. contains a significant amount of additional data and information. In general, the paper provides a better documentation of the results.

We are grateful to the reviewer for recognising our additional efforts.

(2) Nonetheless, the revision has failed to conclusively address the two major shortcomings of the original version, i.e. the unclear functional significance (see below) of most of coding and especially non-coding mutations observed in their large CLL panel. Thus, the claim that the paper reports “22 novel coding drivers and 126 candidate non-coding drivers” remains largely not justified by the data presented. A more conservative statement would state that the paper reports an expanded catalogue of genetic mutations of potential functional significance. This catalogue undoubtedly represents a resource for future studies, but its conceptual value is limited at the present time. Considering also that a presumably large fraction of the observed mutation originates from an hypermutator phenotype, an equally large fraction may represent innocent bystanders, thus making the “clinical impact” premature.

We have significantly toned down/removed the claims in question (formatted as strike-through) and clarified throughout the text whether we talk about known drivers as opposed to the novel, putative/potential/candidate drivers (in blue text).

We thank the reviewer for their suggestions for re-wording. We have removed the strong statement referring to the discovery of novel drivers from the abstract and replaced it with the wording as suggested:

Line 48: ~~“We identify 22 novel coding drivers and 126 candidate non-coding drivers and~~ **We identify an extended catalogue of recurrent coding and non-coding genetic mutations that represents a source for future studies** and ~~show in a subset that these associate with chromatin accessibility and change in gene expression. “~~

We have clarified throughout the text when we talk about known drivers as opposed to the novel, putative/potential/candidate drivers.

For example, we have amended the two following sentences **in the paragraph entitled “Coding mutations and structural variants”** :

Line 96: ~~“We identified 58 driver genes~~ **36 known and 22 novel putative driver genes (Fig.1b, Extended Data Fig.1a-f),** ~~of which 22 were not found associated with CLL in the literature and also not prevalent above 1% in two landmark genomic studies in CLL^{3,7}.”~~

Line 139: “When we associated the ~~58~~ 36 known and 22 putative drivers and regions of CNAs with other biological variables”

In the paragraph entitled “Association of coding mutations with disease evolution”, we edited three sentences in the text as follows:

Line 152: “We examined the relationship between driver recurrent gene mutations and disease evolution”

Line 162: “Restricting analysis to patients with information on long-term survival outcome (n = 243 out of 485), 13 known or putative drivers and recurrent CNAs were significantly associated with progression free survival (PFS) and 11 with overall survival (OS) (FDR<0.05)”

Line 166: “Twenty-one out of the 22 novel putative drivers were also related to disease progression”

The rest of the manuscript contains similar edits.

For novel putative non-coding drivers, we changed the heading of the results to:

Line 180: “Non-coding putative driver mutations”.

We also added a conclusion paragraph at the end of the section on non-coding mutation:

Line 254: “Collectively, these data suggest that a small subset of the non-coding mutations in CLL have characteristics indicative of a driver and target REs of genes that are critical for B-cell development and function as well as cancer progression. However, the effects on chromatin accessibility and gene expression levels were subtle and require in-depth functional characterization.”

These are also detailed below in the point by point response in question 2.

Major concerns:

(1A) The results confirm most known coding drivers of CLL and, thanks to WGS, provide mostly novel data on the inversions (1248) and translocations (933). However, only very few (2 translocations) have meaningful recurrence and none is characterized for functional significance.

We have addressed this by adding:

Line 117: “We detected 993 translocations, of which two occurred in more than 10 samples and affected known genes with no previously documented role in CLL...”

(1B) This concern is greater for missense mutations, which most likely provide a significant amount of background “noise” in the clinical classification.

To emphasise the putative nature of the novel candidate coding drivers, we have added the following paragraph into the discussion:

Line 377: “Based on a strict pipeline for discovery of coding drivers, we selected the top ranked recurrently mutated genes which comprised 36 known CLL drivers^{3,6,7}, and 22 genes as novel putative drivers. Only 32% of variants in those novel putative driver genes were missense variants with the majority being truncating and stop-gain mutations. Although these putative drivers shared characteristics of known drivers (i.e. damaging mutations in protein domains, impact on RNA expression, high CCF that further increased at disease progression, association with survival), we cannot exclude the possibility that some may simply represent passengers.”

(2A) Although prudently defined as “candidates”, the non-coding drivers remain of unexplored significance. Unfortunately, the revised version does not correctly address the concerns stated during the original review. The authors have provided very preliminary data showing quite weak correlations between mutated regulatory regions (RE) and overexpression of the candidate linked genes (BCL2 and BCL6) (Fig. 5).

We completely agree with the reviewer that -as we expected-, the effect of the individual non-coding mutations on chromatin accessibility and target gene expression is weak.

As previously state above, for novel putative non-coding drivers, we therefore changed the heading of the results to:

Line 180: “Non-coding putative driver mutations”.

And we also added a conclusion paragraph at the end of the results:

Line 254: “Collectively, these data suggest that at least a small subset of the non-coding mutations in CLL have characteristics indicative of a driver and target REs of genes that are critical for B-cell development and function as well as cancer progression. However, the effects on chromatin accessibility and gene expression levels were subtle and require in-depth functional characterization.”

We edited the discussion as follow:

Line 387: “For a small subset of selected non-coding candidate mutations, we were able to demonstrate a modest ~~We were able to demonstrate a modest, but definite~~ impact on chromatin accessibility and/or target gene expression (5’UTR of *BCL2*, enhancer of *BCL6*, promoter of *BACH2* and promoter of *ATAD1*). ~~Together, these data imply that integration of the combined small effects of non-coding mutations is needed to fully understand their role in CLL biology.~~”

(2B) They also show that deletion of parts of the REs (BCL6) leads to some reduction in target gene expression (35%) and cell proliferation (40%) in the Raji cell line, a Burkitt Lymphoma (BL)

cell line. This result has little significance for CLL since BL is biologically quite distinct from CLL and RE regions are differentially activated in distinct phenotypes. Most notably, these results address the question of RE dependency, but provide no information of the significance of mutated REs or, better, which/whether mutations matter.

We have removed the CRISPR experiment from the manuscript.

3. Given the above-described uncharacterized nature of the observed mutation, their “bulk” usage for a clinical classifier appears premature and cannot lead to a clinically- actionable classifier vis a vis existing approaches based on solid drivers.

We would like to point out that the aim of our study was never to experimentally characterize the effect of individual non-coding candidate mutations. Our aim was to demonstrate an association between global integrated genome-wide analyses and clinical outcome. We would argue that many prognostically relevant genomic markers are not experimentally identified as disease drivers. When we performed the NMF analysis based solely on the solid drivers as suggested by the reviewer (see Extended Data Figure 15) we found that we could not define our genomic subgroups robustly without the local and global non-coding genomic aberrations (AUC 0.6).

We have added a sentence at the end of the introduction:

Line 77: “Finally, we integrate the different modes of genetic alterations to define five genomic subgroups of CLL **and relate these to clinical outcome.** ~~with clinical impact, which provide better estimation of patient prognosis than achieved by previous single gene analyses.~~ **Our results act as a springboard to in-depth functional validation of putative drivers and raise the prospect that after independent clinical validation, this integrated genome-wide approach could refine current clinical outcome prediction.”**

We have replaced the clinical conclusion in the abstract by the following:

Line 56: ~~“Our results highlight the utility of whole genome sequencing for risk stratification in CLL. While requiring independent validation, our findings highlight the potential of WGS to inform future risk stratification in CLL.”~~

We have added the following discussion point into the manuscript:

Line 401: **“Ideally, only genomic features experimentally validated as disease drivers should be included in any prognostic classification system, even if they were selected by very stringent criteria as those applied in this study (see above). However, it is well recognized that some genomic features are clearly not disease drivers, yet carry prognostic relevance. For example, in CLL, the IGHV mutation status representing the cell-of-origin or telomere length reflecting**

proliferative activity, are strongly associated with clinical outcome, but are not considered disease drivers.

In our NMF model using only the known coding drivers and recurrent CNAs did not allow us to recover the same level of discrimination as that afforded by inclusion of additional local and global non-coding information. This observation implies that the combination of coding and non-coding information in the classifier increases the precision of clinical risk prediction at least in our cohort of clinical trial patients.”

Finally, we have toned down the conclusion of the discussion to:

Line 415: ~~“Hence our analysis serves to support the value of WGS for more precisely defining patient outcome.”~~

Collectively, our study provides a spring board for downstream functional analyses of novel putative coding and non-coding drivers. Further robust testing on independent cohorts of patients undergoing targeted therapy will be required to establish the clinical utility of this new WGS-based classifier further.

Comments from Reviewer #2

(1) The authors have done a great effort in addressing my concerns and suggestions, which has led to clear clarifications, additional associations and improvement of data presentation. I have no further comments.

We thank reviewer 2 for their positive feedback on our revision, in particular as the two points raised by reviewer 1 concerning further characterisation of putative coding and non-coding drivers had also been raised by reviewer 2.

Decision Letter, second revision:

Our ref: NG-A57494R1

9th Jun 2022

Dear Dr. Schuh,

Thank you for submitting your revised manuscript "Whole genome landscape of chronic lymphocytic leukaemia and its association with clinical outcome" (NG-A57494R1). We assesses your revision in-house and I'm delighted to inform you that we'll be happy in principle to publish it in Nature Genetics, pending minor revisions to satisfy our editorial and formatting guidelines.

Regarding co-ordinated publication with the paper from Drs Wu and Getz, please discuss your plans with them for co-ordinated publication and let me know what your preferences are. You can opt to publish online and in print together, or just appear in the same print issue together. In the case of the latter, it is likely that your partner paper would be published online first.

Sincerely,

Safia Danovi
Editor
Nature Genetics

Final Decision Letter:

In reply please quote: NG-A57494R2 Schuh

16th Sep 2022

Dear Dr. Schuh,

I am delighted to say that your manuscript "Whole-genome sequencing of chronic lymphocytic leukaemia identifies subgroups with distinct biological and clinical features" has been accepted for publication in an upcoming issue of Nature Genetics.

Your paper will be published online after we receive your corrections and will appear in print in the next available issue. You can find out your date of online publication by contacting the Nature Press Office (press@nature.com) after sending your e-proof corrections. Now is the time to inform your Public Relations or Press Office about your paper, as they might be interested in promoting its publication. This will allow them time to prepare an accurate and satisfactory press release. Include your manuscript tracking number (NG-A57494R2) and the name of the journal, which they will need when they contact our Press Office.

Please note that *Nature Genetics* is a Transformative Journal (TJ). Authors may publish their research with us through the traditional subscription access route or make their paper immediately open access through payment of an article-processing charge (APC). Authors will not be required to make a final decision about access to their article until it has been accepted. [Find out more about Transformative Journals](https://www.springernature.com/gp/open-research/transformative-journals)

Authors may need to take specific actions to achieve [compliance with funder and institutional open access mandates](https://www.springernature.com/gp/open-research/funding/policy-compliance-faqs). If your research is supported by a funder that requires immediate open access (e.g. according to [Plan S principles](https://www.springernature.com/gp/open-research/plan-s-compliance)) then you should select the gold OA route, and we will direct you to the compliant route where possible. For authors selecting the subscription publication route, the journal's standard licensing terms will need to be accepted, including [self-archiving-and-license-to-publish](https://www.nature.com/nature-portfolio/editorial-policies/self-archiving-and-license-to-publish). Those licensing terms will supersede any other terms that the author or any third party may assert apply to any version of the manuscript.

Please note that Nature Portfolio offers an immediate open access option only for papers that were first submitted after 1 January, 2021.

An online order form for reprints of your paper is available at <https://www.nature.com/reprints/author-reprints.html>. Please let your coauthors and your institutions' public affairs office know that they are also welcome to order reprints by this

method.

If you have not already done so, we invite you to upload the step-by-step protocols used in this manuscript to the Protocols Exchange, part of our on-line web resource, natureprotocols.com. If you complete the upload by the time you receive your manuscript proofs, we can insert links in your article that lead directly to the protocol details. Your protocol will be made freely available upon publication of your paper. By participating in natureprotocols.com, you are enabling researchers to more readily reproduce or adapt the methodology you use. [Natureprotocols.com](http://natureprotocols.com) is fully searchable, providing your protocols and paper with increased utility and visibility. Please submit your protocol to <https://protocolexchange.researchsquare.com/>. After entering your nature.com username and password you will need to enter your manuscript number (NG-A57494R2). Further information can be found at <https://www.nature.com/nature-portfolio/editorial-policies/reporting-standards#protocols>

Sincerely,

Safia Danovi
Editor
Nature Genetics